# Last Millennium Reanalysis with an expanded proxy database and seasonal proxy modeling

Robert Tardif[1], Gregory J. Hakim[1], Walter A. Perkins[1], Kaleb A. Horlick[2], Michael P. Erb[3], Julien Emile-Geay[4], David M. Anderson[5], Eric J. Steig[6,1], and David Noone[2]

[1]Department of Atmospheric Sciences, University of Washington, Seattle, Washington, USA
[2]College of Earth, Ocean, and Atmospheric Sciences, Oregon State University, Corvallis, Oregon, USA
[3]School of Earth Sciences and Environmental Sustainability, Northern Arizona University, Flagstaff, AZ, USA
[4]Department of Earth Sciences, University of Southern California, Los Angeles, California, USA
[5]Retired, NOAA Paleoclimatology Program, Boulder, CO, USA
[6]Department of Earth and Space Sciences, University of Washington, Seattle, Washington, USA

**Correspondence:** R. Tardif (rtardif@atmos.washington.edu)

**Abstract.**

The Last Millennium Reanalysis (LMR) utilizes an ensemble methodology to assimilate paleoclimate data for the production of annually resolved climate field reconstructions of the Common Era. Two key elements are the focus of this work: the set of assimilated proxy records, and the forward models that map climate variables to proxy measurements. Results based on an updated proxy database and seasonal regression-based forward models are compared to the LMR prototype, which was based on a smaller set of proxy records and simpler proxy models formulated as univariate linear regressions against annual temperature. Validation against various instrumental–era gridded analyses shows that the new reconstructions of surface air temperature, 500 hPa geopotential height are significantly improved (from 10% to more than 100%), while improvements in reconstruction of the Palmer Drought Severity Index are more modest. Additional experiments designed to isolate the sources of improvement reveal the importance of the updated proxy records, including coral records for improving tropical reconstructions; and tree-ring density records for temperature reconstructions, particularly in high northern latitudes. Proxy forward models that account for seasonal responses, and dependence on both temperature and moisture for tree-ring width, also contribute to improvements in reconstructed thermodynamic and hydroclimate variables in mid-latitudes. The variability of temperature at multi-decadal to centennial scales is also shown to be sensitive to the set of assimilated proxies, especially to the inclusion of primarily moisture-sensitive tree-ring width records.

## 1 Introduction

Reconstructions of Earth's past climate, particularly covering periods prior to instrumental data sets, are key to understanding the causes of natural climate variability. For example, understanding natural variability provides the basis for improving pre-

dictions of climate variability in the coming decades. Information on past climates has traditionally been derived either from climate proxy data (e.g., tree rings, ice cores, etc.) or from Earth system model simulations, and synthesizing information from these two sources is one of the challenges of paleoclimate science. Paleoclimate data assimilation (PDA) has emerged as a powerful framework for such synthesis because it provides the optimal combination of climate signals recorded by proxies as constrained by the dynamics of Earth system models. PDA-generated climate field reconstructions have been used to investigate climate variability prior to the instrumental era (Goosse et al., 2006, 2010; Widmann et al., 2010; Bhend et al., 2012; Steiger et al., 2014; Matsikaris et al., 2015; Franke et al., 2017; Okazaki and Yoshimura, 2017; Steiger et al., 2018). Within this general PDA framework, a flexible PDA system is being developed for the Last Millennium climate Reanalysis (LMR) project for the production of annually resolved reconstructions of the Common Era. Hakim et al. (2016) describe a prototype configuration of the LMR and show results in good agreement with previous reconstructions of Northern Hemisphere mean near-surface air temperature. Detailed comparisons with several gridded instrumental temperature data products revealed significant skill over tropical regions but less skillful reconstructions over Northern Hemisphere continental areas, where a large proportion of proxy data are located.

As with any data assimilation system, two of three important components impacting the quality of the resulting analyses are the set of assimilated observations (here, proxy records) and the forward models that map variables from climate model output to proxy measurements ("proxy system models"; hereafter, PSMs). The third component is the model providing the prior state, although it is not the focus of this work. Hakim et al. (2016) assimilated proxy records from the first compilation of the PAGES 2k Consortium (PAGES 2k Consortium, 2013), and modeled the proxies through univariate linear regressions calibrated against annually-averaged instrumental temperature data. Here we examine the impact on LMR reconstructions of improvements to these two key components: (1) an updated and expanded proxy database, primarily composed of records from PAGES 2k Consortium (2017), and assessment of the additional records described in Anderson et al. (2019); (2) more realistic PSMs in which seasonality and, for tree-ring-width proxies, temperature and moisture sensitivity are taken into account. Motivation for expanding the proxy database derives from evidence that climate reconstructions are generally sensitive to the set of proxy records used as input (e.g., Wang et al., 2015), while the introduction of more sophisticated PSMs is motivated in part by the fact that comprehensive reconstructions of temperature and hydroclimate variables depend on properly treating temperature-sensitive and moisture-sensitive tree ring proxies (e.g., Steiger and Smerdon, 2017).

The focus of improvements in PSMs here is on regression-based (i.e. statistical) forward models, in contrast to recent efforts focusing on process-based PSMs (see e.g., Breitenmoser et al., 2014; Dee et al., 2016; Acevedo et al., 2017). Our objective is to establish baseline skill of PDA reconstructions using statistical PSMs, to serve as a benchmark for evaluating possible improvements associated with process-based PSMs (e.g., Dee et al., 2016). Here we develop a hierarchy of statistical PSMs to identify aspects that contribute increased skill to reconstructed temperature and hydroclimate states compared to the prototype LMR.

The remainder of the paper is organized as follows. Section 2 outlines the LMR PDA-based framework and describes the proxy database and PSMs. Reconstructions based on this configuration are presented in section 3, with comparisons to the

prototype described in Hakim et al. (2016). Section 4 explores the contributions to improvements in the new reconstructions. A concluding summary is given in section 5.

## 2  Methods

Paleoclimate data assimilation has three main components: proxy records, providing an indirect record of past climatic conditions; climate models, providing prior estimates of the climate; and proxy system models, providing the connection between the model prior and the proxy values. The method for each component is now described.

### 2.1  Data assimilation framework

LMR employs ensemble data assimilation (DA) to blend information from proxies and climate model data. DA is performed using a variant of the ensemble Kalman filter, which for our application appears to perform well compared to alternative PDA methods such as particle filters (Liu et al., 2017). The update equation is given by

$$\mathbf{x}_a = \mathbf{x}_b + \mathbf{K}[\mathbf{y} - \mathbf{y}_e]. \tag{1}$$

Here, $\mathbf{x}_b$ is the prior state vector, which contains the climate variables to be reconstructed, averaged over an appropriate timescale (here, annual), and $\mathbf{x}_a$ is the posterior state vector (i.e. the reanalysis, or reconstruction). The state vector may include scalars, such as climate indices, and/or grid-point data for spatial fields. Vector $\mathbf{y}$ contains the assimilated proxy data (i.e. observations) and $\mathbf{y}_e$ is a vector containing estimates of the proxies derived from the prior by

$$\mathbf{y}_e = \mathcal{H}(\mathbf{x}_b), \tag{2}$$

where $\mathcal{H}$ is the forward model mapping the prior $\mathbf{x}_b$ to proxy space (i.e. the PSM, see section 2.4). The innovation, $[\mathbf{y} - \mathbf{y}_e]$, is the new information from the proxies not already contained in the prior. This new information is weighted against the prior through the Kalman gain matrix

$$\mathbf{K} = \mathbf{B}\mathbf{H}^{\mathrm{T}}\left[\mathbf{H}\mathbf{B}\mathbf{H}^{\mathrm{T}} + \mathbf{R}\right]^{-1} \tag{3}$$

where $\mathbf{B}$ is the prior covariance matrix, $\mathbf{R}$ is the error covariance matrix for the proxy data, and $\mathbf{H}$ is the linearization of $\mathcal{H}$ about the mean value of the prior. Here, Eq. (1) is solved using the ensemble square-root filter (EnSRF) approach of Whitaker and Hamill (2002), in which the ensemble-mean and perturbations about the ensemble mean are solved separately. Moreover, $\mathbf{R}$ is taken as a diagonal matrix (uncorrelated observation errors) where the diagonal elements represent the error variance for each assimilated proxy record; details on how these are estimated are provided in Section 2.4. This allows for serial processing of observations, in which observations are assimilated one at a time. This greatly simplifies the implementation of covariance localization, which is used to control sampling error in the prior covariance. Solutions for the ensemble mean, $\overline{\mathbf{x}}_a$,

and perturbations, $\mathbf{x}'_a$, for the single $k^{th}$ proxy $y_k$, are obtained from:

$$\overline{\mathbf{x}}_a = \overline{\mathbf{x}}_b + \frac{w_{loc} \circ cov(\mathbf{x}_b, y_{e,k})}{var(y_{e,k}) + R_k} \left(y_k - \overline{y}_{e,k}\right) \tag{4a}$$

$$\mathbf{x}'_a = \mathbf{x}'_b - \left[1 + \sqrt{\frac{R_k}{var(y_{e,k}) + R_k}}\right]^{-1} \frac{w_{loc} \circ cov(\mathbf{x}_b, y_{e,k})}{var(y_{e,k}) + R_k} \left(y'_{e,k}\right) \tag{4b}$$

where $y_{e,k}$ is the prior estimate of the proxy from (2) and $R_k$ is the diagonal element of $\mathbf{R}$ corresponding to proxy $y_k$. The

5 ensemble of updated states is then recovered by combining the posterior ensemble mean and perturbations

$$\mathbf{x}_a = \overline{\mathbf{x}}_a + \mathbf{x}'_a. \tag{5}$$

Covariance localization, given by a Schur product denoted by $\circ$ in Eqs. 4 (i.e. element-wise multiplication), is a distance-weighted filter $w_{loc}$ on the prior covariance matrix (see e.g. Hamill et al., 2001). Sections 4.2, and S5 in the supplementary material, provide more details on localization.

We also use an "appended state vector" approach that avoids the need to recompute (2) after each proxy is assimilated. The $\mathbf{y}_e$ proxy estimates from each record are appended to the state vector $\mathbf{x}_b$:

$$\mathbf{x}_b = \begin{bmatrix} x_1 \\ \vdots \\ x_N \\ y_e^1 \\ \vdots \\ y_e^P \end{bmatrix}, \tag{6}$$

where the $x_1 \ldots x_N$ elements contain the ensemble grid point data from model variables included in the state (e.g. temperature, precipitation etc.), with $N$ the sum of the number of variables times the number of grid points, and the $y_e^1 \ldots y_e^P$ are the

15 ensemble proxy estimates for each of the $P$ proxy records considered. Each of the $x_1 \ldots x_N$ and $y_e^1 \ldots y_e^P$ elements are of dimensions $1 \times N_{ens}$, where $N_{ens}$ is the specified size of the ensemble. Hence, $\mathbf{x}_b$ is a matrix of dimension $(N + P) \times N_{ens}$. With such an appended state, the $y_e$ elements in Eq. 6 are updated through Eqs. 4 as any other state variables, eliminating the need to re-evaluate $\mathbf{y}_e$ with Eq. 2 once the state has been updated. This simplification is particularly attractive in the context of LMR updates discussed herein as it enables a straightforward implementation of seasonal PSMs (i.e. forward models more

accurately representing the seasonal responses of individual proxy records) as discussed in section 2.4. In our implementation for the reconstruction of annually-averaged states, the data assimilation procedure follows this general algorithm.

1. The proxy estimates ($\mathbf{y}_e$) are pre-calculated using Eq. 2 with either annually- or seasonally-averaged model data as input (i.e. the $\mathbf{x}_b$ in Eq. 2).

2. A sample ensemble of annually-averaged model states is randomly drawn from a pre-existing simulation to form the

25 main part of the prior state vector (i.e. the $x_1 \ldots x_N$ elements in Eq. 6).

3. The pre-calculated $y_e^1 \ldots y_e^P$ proxy estimates are added on to form the appended state as shown in Eq. 6. This appended state becomes the $\mathbf{x}_b$ in Eq. 1, which is decomposed in an ensemble-mean ($\overline{\mathbf{x}}_b$) and perturbations about the mean ($\mathbf{x}_b'$) as shown in Eqs. 4.

4. Proxies forming the $\mathbf{y}$ vector are then serially processed, with the updated state, including the proxy estimates, obtained from Eqs. 4. The reanalysis is completed for one year once all proxies have been assimilated.

We note here that with a configuration involving seasonal PSMs without the use of an appended state, the vector $\mathbf{x}_b$ has to include states with sufficient temporal resolution to allow the calculation of the updated seasonal $y_e^1 \ldots y_e^P$ proxy estimates. In this scenario, an additional step to the ones listed above is required, involving Eq. 2 using the appropriate seasonally-averaged updated states as input. With proxies characterized by a wide range of seasonal responses, this requirement would impose an $\mathbf{x}_b$ composed of monthly data which would greatly increase the computational cost of the reanalysis. Reanalysis results would also likely be adversely affected by the larger noise level characterizing data at shorter (i.e. monthly) timescales through its impact on ensemble estimates of prior covariances (see, e.g. Tardif et al., 2016).

As in Hakim et al. (2016), an "offline" DA approach is used, where the prior ensemble is formed by random draws of time–averaged states from a pre-existing millennial-long model simulation, with the same randomly drawn ensemble members used for every year in the reconstruction of a given reanalysis realization (see Section 2.3 below). We note that in the limit of no proxy information, this approach leads to a posterior that reverts to the prior ensemble (see Eq. 1), which randomly samples the model climate and therefore has no skill over the model climatology. This is in contrast to online DA (e.g., Matsikaris et al., 2015; Perkins and Hakim, 2017), where a numerical model is used to dynamically forecast the evolution of climate states from the latest proxy-informed analysis to the following year, when new proxy observations are assimilated. The "offline" approach, introduced by Oke et al. (2002) and Evensen (2003) and used in an ocean DA system by Oke et al. (2005), offers several practical advantages, particularly from a computational cost perspective (Oke et al., 2007). Its use is further justified when model forecasts have limited skill over timescales corresponding to the time interval between updates, as is the case here with global climate models and proxies assimilated on an annual basis. This scenario is further supported by the PDA results of Matsikaris et al. (2015) who show similar performance is achieved with online and offline approaches. From a cost-benefit perspective, the high cost of running ensembles of comprehensive global climate model simulations does not appear justified. However, ongoing research suggests cost-effective online PDA may be achieved by using simplified climate models (Perkins and Hakim, 2017).

## 2.2 Climate proxies

Our proxy database is updated to the latest PAGES 2k collection (PAGES 2k Consortium, 2017, hereafter PAGES2k-2017). This dataset represents the community standard in global proxy observations covering the Common Era (CE) and serves as the core source of proxy information used in our updated reanalysis. PAGES2k-2017 proxies were screened to retain temperature-sensitive records, extensively quality controlled, and described by more metadata compared to previous collections. The additional records assembled by Anderson et al. (2019)[1], consisting in large part of the tree ring width records from Breitenmoser

et al. (2014) (hereafter B14), are later considered as potential enhancement to proxy information used in our paleo-reanalyses (see section 4.3).

As in the LMR prototype (Hakim et al., 2016, hereafter H16), only records with sub-annual to annual resolutions are considered; sub-annual records are averaged to annual. Figure 1 compares the PAGES 2k Consortium (2013) (hereafter PAGES2k-2013) dataset used in H16, and the PAGES2k-2017 update. Only records for which a PSM can be established are shown in Fig. 1, defined by proxy records with at least 25 years of (non-contiguous) overlap with calibration data (see section 2.4). Compared to the proxies assimilated in H16, PAGES2k-2017 data provide enhanced spatial coverage in the tropics with additional coral $\delta^{18}$O and Sr/Ca records. Additional tree ring wood density records from Europe and western North America are also included. The temporal distribution of the total number of records remains similar, except for significant increases in the number of tree ring width and coral proxies during 1800–2000CE, and tree ring wood density records during 1500–2000CE.

## 2.3 Climate model prior information

For all reconstruction experiments reported in this paper, the prior state vector is formed with data from the CMIP5 (Taylor et al., 2012) Last Millennium simulation from the Community Climate System Model version 4 (CCSM4) coupled atmosphere-ocean-sea ice model. The simulation covers years 850 to 1850 CE and includes incoming solar variability, variable greenhouse gases as well as stratospheric aerosols from volcanic eruptions known to have occurred during the simulation period (see Landrum et al., 2013). The same "offline" DA methodology as in H16 is used, where the prior ensemble is a random sample of annual averages, with the same sample used for all years of the reconstruction. The sampled states are deviations (i.e. anomalies) from the temporal mean taken over the entire length of the simulation. Therefore, the prior ensemble–mean does not contain time-specific information about climate events (e.g. a volcanic eruption) or trends characterizing specific periods (e.g. twentieth century warming). Consequently, all trends and temporal structure in reconstructed fields result from information provided by the proxies. Finally, the spatial resolution of prior state variables is reduced from $0.95^o \times 1.25^o$ of the Last Millennium simulation to a of $4.3^o \times 5.7^o$ Gaussian grid as in H16.

All reconstruction experiments are composed of 51 Monte-Carlo assimilation realizations, each using a different randomly chosen 100–member ensemble and 75% of available proxy records for assimilation. This Monte-Carlo sampling over subsets of prior states and proxy records is designed to incorporate uncertainties in covariance estimates derived from model states, and uncertainties associated with proxy error estimates. Moreover, we have found that averaging over ensembles from Monte-Carlo realizations leads to more accurate results. This is likely the result of averaging over random errors introduced into the reanalysis from few randomly chosen proxy records with underestimated observation errors. Little sensitivity to the use of 75% of the proxies for each realization has been found (not shown), while 100 members have been chosen to maintain consistency with H16. In the following, climate reanalyses are taken as the mean over the 100–member DA ensembles and 51 Monte-Carlo realizations (i.e., a 5100–member "grand ensemble").

---

[1]An exception is the use of the Palmyra coral record from Cobb et al. (2003) rather than the Emile-Geay et al. (2013) update, as described in Anderson et al. (2019).

## 2.4 Proxy modeling

A critical component of PDA is the mapping of prior climate state variables (e.g., temperature, precipitation from a climate model) to the assimilated proxies (e.g., tree ring width). This is expressed mathematically by Eq. 2, section 2.1, where the operator $\mathcal{H}$ (i.e. the forward model) ideally represents the complete set of processes associated with proxy values, i.e. a comprehensive physically-based PSM. This remains a major challenge as the information archive is often complex, involving physical, biological and chemical processes (Evans et al., 2013). Despite recent progress in the development and use of process-based PSMs (e.g., Dee et al., 2015, 2016; Goosse, 2016; Steiger et al., 2017; Acevedo et al., 2017), the focus here is on statistical PSMs, which offer distinct advantages: 1) ease of implementation and flexibility with respect to forward modeling of multiple proxies, regardless of archive types, measurements, units etc.; 2) observation error statistics for each assimilated record are well-defined from the regression (see below); and 3) regressions are formulated on the basis of deviations from the mean over a reference period (e.g. 1951–1980) of the driving climate variable(s), therefore avoiding issues with absolute calibration where climate model bias is problematic, particularly for PSMs having threshold transitions (see e.g., Dee et al., 2016). Statistical PSMs also have distinct disadvantages: 1) PSMs cannot be calibrated without sufficient overlap with calibration data (a threshold of at least 25 overlapping data is imposed); 2) the accuracy of the models depends on the limitations of the calibration datasets (e.g. less reliable analysis over the Southern Ocean and over high latitude continental areas due to a lack of observations); 3) possible lack of stationarity of the derived relationships established with instrumental–era data; and 4) lack of representation of nonlinear and/or multivariate influences when PSMs are formulated as linear univariate models. Despite these limitations, statistical PSMs provide advantageous capabilities within the context of the LMR and, moreover, define a baseline to measure future progress with the development of process-based PSMs.

Here, univariate and bivariate statistical PSMs are considered,

$$y_k = \beta_{0k} + \beta_{1k}\overline{X_1'} + \epsilon_k \tag{7}$$

and

$$y_k = \beta_{0k} + \beta_{1k}\overline{X_1'} + \beta_{2k}\overline{X_2'} + \epsilon_k \tag{8}$$

where $y_k$ are annualized observations from the $k^{th}$ proxy time series, $X_1', X_2'$ are anomalies of key climate variables (e.g. near-surface air temperature and precipitation) from calibration instrumental–era datasets, $\beta_0$ is the intercept and $\beta_1, \beta_2$ are the slopes with respect to the $X_1'$ and $X_2'$ independent variables respectively, and $\epsilon$ is a Gaussian random variable with zero mean and variance $\sigma^2$. The overbar in Eqs. 7 and 8 denotes time averages over annual periods as in H16, or over appropriate seasonal intervals for the seasonal PSMs. Calibration data concurrent with available proxy observations are taken at the grid point nearest the proxy location and the appropriate least-squares solution determines regression parameters $(\beta_0, \beta_1, \beta_2, \sigma)$. In this version of LMR, PSM configuration is the same for each proxy category (e.g., univariate for all coral $\delta^{18}O$, bivariate for all tree ring widths records, etc.).

With the framework described above, the regression–based approach measures the diagonal elements in matrix $\mathbf{R}$ through the variance of regression residuals, i.e. $R_k = \sigma^2$. This is a key parameter in PDA as it determines the extent to which the

information provided by the proxy is weighted against prior information in the resulting reanalysis. This method provides a sound basis through which assimilated proxy records influence the reanalysis depending on the strength of their relationship to the dependent climate variables. For example, a record with a poor fit to calibration data will be characterized by larger residuals, hence larger observation error variance, and less weight in the reanalysis relative to a record that has a stronger correlation with climate variables. We note that modestly different results are obtained with different observational calibration datasets (see H16).

The calibration datasets used in this study are the NASA Goddard Institute for Space Studies (GISS) Surface Temperature Analysis (GISTEMP) (Hansen et al., 2010) version 4 for temperature, and the gridded precipitation dataset from the Global Precipitation Climatology Centre (GPCC) (Schneider et al., 2014) version 6 as the source of monthly information on moisture input over land surfaces. The use of precipitation instead of the more traditional Palmer Drought Severity Index (PDSI) to account for moisture is described in more detail in section S4 of the supplementary material.

### 2.4.1 Seasonality

Here we take advantage of the availability of expert information about the seasonal response to temperature for each proxy record included in the PAGES2k-2017 metadata. This information is not available in PAGES2k-2013, hence the use of PSMs calibrated on annual averages for all records in H16. Seasonality information is provided for each record as a numerical representation of a sequence of consecutive months (e.g. JJA as [6,7,8]). Seasonal PSMs are derived by using this sequence as the averaging period defining $\overline{X'_1}$ and $\overline{X'_2}$ in Eqs. 7 and 8.

Precise information on proxy seasonality is however not available for all records in the updated LMR proxy database. The proxies from Anderson et al. (2019), considered as a possible additional expansion of the proxy database, have not been subjected to extensive community–wide screening and vetting as with the PAGES2k-2017 proxies. In particular, seasonality information for the large number of additional tree ring records from B14 has been encoded using a simple latitudinal dependence which does not attempt to represent possible record-by-record diversity (see Anderson et al., 2019). This lack of expert-informed seasonality motivates an objective alternative to the metadata seasonality information for calibrating tree ring width (TRW) forward models. We consider several potential seasonal periods, perform a regression over each possible season, and identify the linear relationship providing the best fit to proxy values, as defined by the maximum value of the adjusted $R^2$, a goodness-of-fit measure defined as (Goldberger, 1964, p. 217):

$$R^2_{adj} = 1 - \left[ \frac{(1 - R^2)(N - 1)}{N - M - 1} \right]. \tag{9}$$

Here, $R^2$ is the variance explained by the linear model, $N$ is the sample size and $M$ is the number of predictors in the model. The adjusted $R^2$ penalizes complexity (i.e. the number of predictors) of the model in such a way that values characterizing a more complex model will increase only if the additional predictors improve the fit more than would be expected by chance. Test periods considered include, in addition to the seasonal response in the proxy metadata (if available), the calendar year, boreal summer (JJA) and boreal winter (DJF), and extended Spring and Fall growing seasons (MAMJJA, JJASON for NH trees, SONDJF, DJFMAM for SH trees) to account for ecosystem–dependent variations in tree growth shifted toward the earlier or

later parts of the warm season (see, e.g., Sano et al., 2009; D'Arrigo et al., 2005). With this test set of seasonal responses, the dominant sensitivity of some TRW chronologies to winter temperature (D'Arrigo et al., 2012) is included, as well as the winter and spring precipitation sensitivities characterizing some tree species (see, e.g., Stahle et al., 2009; Touchan et al., 2003). The latter point is germane to the calibration of seasonal TRW models using precipitation as a predictor (see next section).

### 2.4.2 Tree ring width sensitivity to temperature and moisture

Proxy number is strongly dominated by TRW records in the LMR proxy database, particularly with the addition of chronologies from B14. Furthermore, these records have not been screened on temperature, which opens the opportunity to measure moisture sensitivity through the regression framework. The addition of an explanatory variable increases the potential for overfitting, and our framework is designed to measure that using the 25% of proxies withheld from assimilation, for which we can measure reconstruction errors and compare results with proxies that were assimilated (see discussion of proxy verification results in section 3).

Two methods are considered, both adding a dependence to moisture input (as represented here with precipitation). The first maintains the univariate approach (Eq. 7) but considers linear PSMs calibrated against either temperature or precipitation. For each TRW record, distinct regressions with either variable are established and the model providing the best fit to proxy data is selected. Following a common practice in dendroclimatology, this approach determines whether the record is predominantly temperature or moisture limited (see, e.g., St. George, 2014). Similar univariate "temperature or moisture" models (abbreviated as "TorM" hereafter) are successfully used in Steiger et al. (2018). The second method consists of simultaneously factoring both temperature and moisture sensitivities through the bivariate relationship expressed in Eq. 8.

Seasonal univariate TorM and bivariate TRW models are considered, with distinct sets of models calibrated using proxy seasonality either from the proxy metadata or objectively-derived during calibration. This selection has important implications for the representation of the proxy seasonal response to moisture in particular. For the proxy metadata, seasonality for moisture is assumed to be identical to temperature as this is the only information available, whereas the objective approach allows for independent encoding of seasonal responses to temperature and moisture. For TorM models, the objective seasonality for univariate moisture models is independent of temperature as it is determined solely from the fit to precipitation data. For bivariate PSMs, all possible combinations of seasonal responses specified independently for temperature and moisture are considered and the combination providing the best fit is selected. With such flexibility, TRW models with objectively-derived seasonality are expected to provide a more realistic representation of the significant variability in seasonal responses to moisture characterizing TRW records (see, e.g., St. George et al., 2010). We note that this approach is similar to the methodology used to calibrate the VS-Lite model (Tolwinski-Ward et al., 2011) in that grid cell temperature and precipitation data are used to determine site-specific growth seasons and seasonally-dependent temperature and moisture growth parameters.

An examination of PSM characteristics, summarized here, with more detail provided in appendix A, confirms that proxies are represented more accurately by seasonal models, particularly for tree-ring wood density and width records (see Table A1). Moreover, more-accurate fits to TRW data are obtained when proxy seasonal responses are determined objectively during model calibration. Finally, the addition of moisture input as a climate driver in TRW modeling proves most beneficial when

implemented in bivariate models (see Table A2). These findings serve as the basis for defining a PDA configuration used for the reconstruction described in the next section.

## 3   The updated reanalysis

We present a comparison between the updated reanalysis described by the method in the previous section with the LMR prototype described in H16[2]. Specifically, the updated reanalysis consists of all proxy records in the expanded database, using objectively-derived seasonal PSMs, with a bivariate formulation for all TRW proxies and univariate for all other proxy types. Covariance localization is applied with a 25000 km cut-off radius (see section 4.2 for more details). In the next section we identify the sources of improvement that contribute to the increase in skill of the updated reconstruction. Results are evaluated against various twentieth century instrumental data and reanalyses, as well as verification performed in proxy space, using the Pearson correlation coefficient and the coefficient of efficiency (CE) (Nash and Sutcliffe, 1970). These skill scores are complementary since correlation measures signal timing while CE, based on mean square error with climatology as a reference, is sensitive to bias and errors in signal amplitude.

Figure 2a shows a comparison of reconstructed global-mean temperature (GMT) between the prototype and updated reanalyses over the entire Common Era. Similar features are observed in the ensemble mean from both reanalyses, namely the cooling trend over most of the Common Era, followed by the industrial-era warming. Superimposed on these main trends, significant multidecadal to multicentennial variability characterize both reanalyses, including a cool period prior to the industrial warming, consistent with the Little Ice Age (LIA). Noticeable differences also exist between the reanalyses, most noticeably the absence in the updated LMR of the relatively warm period during 870–1000CE, representing the Medieval Climate anomaly (MCA). Also, warmer conditions prevail in the prototype during the second half of the fifteenth century, while cooler conditions occur during the early part of the instrumental period in the prototype compared to the updated reanalysis. We note however that verification against instrumental–era temperature analyses (discussed later in the section) provides evidence that the prototype reanalysis is too cold during that period.

Ensembles provide access to useful diagnostics regarding reconstruction uncertainty. It can be shown mathematically that the assimilation of observations monotonically reduces the variance of the posterior ensemble compared to the prior. The ratio of ensemble variance of the posterior (reanalysis) to the prior is a measure of the information provided by the assimilated proxies. Figure 2b shows the temporal evolution of $1 - \mathrm{Var}[\mathbf{x}_a]/\mathrm{Var}[\mathbf{x}_b]$, so that a value of zero indicates no influence from proxies and one implies that all error has been removed. In the early part of the Common Era, when few proxy data are available, a variance decreases of only 10-15% occur in the prototype compared to 15-20% for the updated reanalysis. The influence of proxies gradually increases after 450CE, at similar rates in both reanalyses. The reductions in variance are roughly similar in both reanalyses until 1700CE, corresponding to the period with a significantly larger number of proxies in the updated database

---

[2]We use the experiment included in Figure 12 of H16, with PSMs calibrated using GISTEMP. Moreover, we use this configuration to generate a reconstruction of the Palmer Drought Severity Index (PDSI), which was not included in H16.

(see Fig. 1). The largest reduction, 68% in the prototype compared to 78% in the updated reanalysis, is found during the 20th century when the most proxies are available, which underscores the importance of the expanded proxy database in LMR.

To gain further perspective on our results, we compare the reconstructed Northern Hemispheric average 2m air temperature from the prototype and updated reanalyses with other reconstructions quoted in the Intergovernmental Panel on Climate Change Fourth and Fifth Assessment Reports (IPCC AR4 and AR5) (Fig. 2c). Here we restrict the comparison to reconstructions covering the entire hemisphere and with a temporal coverage extending to at least 1980. A 30-year low-pass Butterworth filter is applied on all results to highlight variability at the lower frequencies. The comparison shows that most reconstructions from other studies are within the bounds of the LMR ensemble most of the time, indicating a general agreement between the different products, at least within the bounds of uncertainty as defined from LMR. As with GMT, periods with the largest differences correspond to the MCA (870–1000CE), the late fifteenth and early sixteenth centuries and the latter part of the nineteenth century. First, the reconstructed colder temperatures during the medieval period are in contrast with the prototype LMR and other reconstructions. However, this period is one where the various reconstructions exhibit significant disagreement. This sensitivity to the proxy network and reconstruction method underscores the inherent ambiguities in defining this feature, as discussed in Diaz et al. (2011). With respect to LMR, differences between the update and prototype are primarily rooted in the change from PAGES 2k Consortium (2013) to the more recent PAGES 2k Consortium (2017) proxy data. A distinctly warmer medieval period is not a prominent feature of the new collection, as indicated by the global temperature composites presented in PAGES 2k Consortium (2017). Second, the colder temperatures in the updated reanalysis during the late fifteenth and early sixteenth centuries is in better agreement with the majority of reconstructions in other studies, with respect to both the magnitude and trend of temperature anomalies. The LMR prototype appears as a warm outlier for this 100-year period. In contrast, the prototype LMR appears as a cold outlier during the latter part of the nineteenth and early twentieth centuries. During that period, the updated reanalysis is in better agreement with results from other authors, in particular with the borehole temperature reconstruction by Pollack and Smerdon (2004).

GMT verification results of the LMR ensemble mean against various instrumental temperature products are shown in Figs. 3a and b for the prototype and updated reanalyses respectively. Noticeably higher verification scores characterize the updated LMR, including a 9% increase in CE relative to the average of observations-based temperature analyses ("consensus"), and an increase in CE in the verification of the detrended GMT (over 1880-2000CE) from 0.32 in the prototype to 0.59 in the updated reanalysis (see Table 1). Spatial verification is provided by comparing the LMR gridded 2m air temperature field against the Berkeley Earth instrumental–era temperature analysis (Rohde et al., 2013) (Fig. 4). Berkeley Earth is chosen as the verification reference as it is not used to calibrate the PSMs, and provides the most complete spatial coverage compared to other instrumental products. The updated temperature reconstruction is largely improved compared to the prototype over large areas, including the tropical Pacific, northern Atlantic, western North America, northern Europe, central Asia, Oceania, and over portions of the Pacific sector of the Southern Ocean. The improvement is reflected in both correlation and CE scores, indicating improved timing and amplitude in reconstructed temperature variability. Exceptions are found over parts of the southern Atlantic and Indian oceans, although the decrease in skill is generally more modest compared to the magnitude of improvements elsewhere.

Next we verify a climate variable away from the surface, the 500 hPa geopotential height field, against the corresponding field from NOAA's twentieth century reanalysis (20CR-v2, Compo et al., 2011) (Fig. 5). Once again we find the largest improvements over extratropical continental locations, and over the Arctic. We note similar improvements are found over the Northern Hemisphere mid-latitudes when verified against the ERA-20C reanalysis (Poli et al., 2016) (not shown); however, over the Northern Hemisphere high latitudes verification against ERA-20C is worse, which underscores significant differences between twentieth-century reanalyses in these data-sparse regions.

Table 1 summarizes the verification results discussed above through globally averaged verification scores. The table also includes verification results of reconstructed PDSI, not discussed above. A more detailed analysis for this variable is reserved for section 4.3, where the role of additional proxy records is discussed. Improvements in the updated reanalyses are evident for all reconstructed variables, particularly with respect to the CE score, which is sensitive to bias and amplitude in interannual variability. These skill improvements suggest significant positive impact from the updated tropical coral proxies and tree-ring proxies at higher latitudes. Furthermore, we anticipate that generalizing PSMs to accounting for seasonality and moisture sensitivity for TRW proxies also contribute to the improvements.

We consider now an independent evaluation of the reconstructions in proxy-space using proxies withheld from assimilation. Proxy time series estimated (forward-modeled) from the posterior (i.e. the reconstructions) are compared to the actual proxy observations and various skill metrics are evaluated. Verification of proxy estimates obtained from the uninformed climate-model prior serve as a reference for comparison. Specifically, we use the change in CE between the posterior proxy estimates and estimates obtained from the prior, $\Delta$CE = (CE$_{posterior}$ - CE$_{prior}$). Values are compiled from all proxy records withheld from assimilation, and the following summary scores are considered: the fraction of all proxy records which are characterized by a positive $\Delta$CE (i.e. proxy records more accurately represented in the posterior than in the prior), and the median of the $\Delta$CE distribution compiled over all proxy time series. These provide global summary measures of how reanalyses skill differs from the prior. An additional discriminating factor on the quality of the reanalysis is "ensemble calibration" as defined by (Murphy, 1988),

$$ECR = \left[ \frac{1}{N-1} \sum_{n=1}^{N} (v_n - \overline{x_n})^2 \right] \left[ \frac{1}{N-1} \sum_{n=1}^{N} (\sigma_{x,n}^2 + \sigma_{v,n}^2) \right]^{-1}, \tag{10}$$

where the numerator is the mean square error (MSE) of the analysis ensemble mean with respect to verification data $v$ (i.e. the proxies), and the denominator is the innovation variance: the sum of the analysis ensemble variance $\sigma_x^2$ and the error variance $\sigma_v^2$ characterizing the verification data. Here we apply Eq. 10 to proxy time series so the error variance $\sigma_v^2$ corresponds to the $R_k$ terms in Eqs. 4. The ECR ratio expresses the degree to which the ensemble predicts the distribution of observations. A well-calibrated ensemble exhibits an approximate agreement between the ensemble variance and the ensemble–mean MSE, i.e. ECR $\approx$ 1.0, while an overdispersive ensemble has variance larger than the ensemble-mean MSE (ECR < 1.0), and an under-dispersive ensemble is diagnosed when its variance is smaller than the ensemble–mean MSE (ECR >1.0). Proxy verification results are shown in Table 2, over different periods of the Common Era. Significantly reduced skill characterizes the earliest period of the Common Era, followed by a continuous increase over time in all verification metrics considered, for both LMR reanalyses. We also note that reanalysis ensembles are generally well-calibrated throughout the Common Era, indicating that

respective uncertainties remain consistent with mean errors (i.e. reliable ensembles). Although verification data is not identical between prototype and updated reanalyses, we also note that the increase in skill is more pronounced in the updated reanalysis, particularly from 1000CE onward. These results provide further evidence of a more skillful updated LMR. In the following section we systematically evaluate improvements from various sources.

## 4   Sources of improvement

In this section, we identify the sources of reanalysis improvement. Results from multiple reconstruction experiments are presented, designed to quantify the impact of PSM formulation, the role of covariance localization and the assimilation of additional proxies.

### 4.1   Proxy system models

The different PSM configurations described in section 2.4 are used in a series of reconstruction experiments using PAGES2k-2017 proxies exclusively. We note that these records have well-defined seasonal metadata.

The impact of seasonal PSMs is first considered with three experiments performed using univariate temperature regression models for: (1) annual-mean calibration; (2) seasonality defined by expert metadata; and (3) objectively determined seasonality. Performance is again measured by correlation and CE scores with verification against the Berkeley Earth analysis. Relative to reconstructions with annual-mean PSMs (Figs. 6a and b), the reconstructions with seasonal PSMs (Figs. 6c–f) show improvements in both measures over nearly the entire globe (Figs. 6g–j). Results show a larger improvement for CE (Figs. 6h and j) compared to correlation (Figs. 6g and i), reflecting improvement in both the amplitude of temperature variability and bias. Noteworthy improvements are found in regions with large numbers of tree-ring proxies, such as the western United States, the region around and including Alaska, Northern Canada and western Arctic ocean, over Scandinavia and Norwegian Sea, central Asia and over the Southern Pacific west of the Antarctic Peninsula (see Fig. 6h). Comparing the differences of correlations and CE in Figs. 6i and j to those shown in Figs. 6g and h reveals that PSMs with objectively-derived seasonality contribute positively to skill for the aforementioned regions, especially where tree ring width records are most abundant (e.g. North America and Asia).

We turn now to the impact of moisture on seasonal TRW PSMs on the reconstructions. Since objectively defined seasonality performs best (i.e., Figs. 6e and f), reconstructions generated with univariate PSMs are used as the reference for measuring skill improvements for modeling TRW records as univariate in either temperature or moisture (abbreviated as "TorM") (Figs. 7c and d) and for bivariate "temperature and moisture" PSMs (Figs. 7e and f). Improvement over univariate PSMs is apparent for the bivariate approach compared with the univariate "TorM" approach (cf. Fig. 7 panels g,h with i,j, respectively). In the bivariate approach regions such as western North America and central Asia, where most of the TRW records are found, improve the most in CE, but also over Australia, likely in response to the improved modeling of TRW records in New Zealand and Tasmania. Improvements are also noticeable, through teleconnections with proxy locations in the central Atlantic and southern India

Oceans, and over the eastern North Pacific Ocean. A decrease in skill is present over the mid-latitude Pacific ocean, but this is smaller in magnitude compared with skill enhancements elsewhere.

Verification of GMT for reconstructions using seasonal PSMs (Table 3) yields a similar interpretation to the spatial verification results. Compared to the consensus of instrumental-era products, we find that the 20th century trend in GMT is overestimated with the PAGES2k-2017 proxy data set if univariate PSMs are used. This is particularly the case with annual PSMs. Better agreement is obtained when seasonal bivariate PSMs are used to model TRW proxies. The representation of GMT interannual variability as measured by verification of the detrended GMT is also improved with seasonal PSMs, particularly for the CE metric. Similar to spatial verification results, PSMs with objectively-derived seasonality and bivariate TRW modeling have GMT reconstructions with consistently higher skill scores.

We recognize that the previous evaluation relies on comparisons with observation-based products covering the same time period as the data used to calibrate the statistical PSMs. To test the sensitivity of the results to the calibration period, we conduct additional independent instrumental–era calibration–validation experiments where PSMs are calibrated over a subset of the instrumental-era period and reconstructions are evaluated with data not used in calibration. Results from these experiments, described in section S3 in the supplementary material, confirm the main results and conclusions drawn here on the superiority of seasonal PSMs relative to those calibrated with annual averages, and the use of bivariate models for TRW proxies.

We now examine results from an evaluation performed in proxy-space using proxies withheld from assimilation as in section 3. Results for both the PSM calibration and pre-calibration periods are shown in Table 4. Differences among the various experiments suggest the superiority of the seasonal (with objective seasonality) PSMs as skill scores consistently rank among the highest among all experiments, for both calibration and pre-calibration periods. The reconstruction using univariate annual PSMs show the weakest verification statistics, confirming the verification based on instrumental-era analyses. Finally, use of bivariate seasonal PSMs for TRW records is also suggested from proxy validation results, as larger correlations and $\Delta$CE are obtained with this configuration.

## 4.2 Covariance localization

One approach to managing sampling error in ensemble data assimilation is through spatial covariance localization. Localization is applied to minimize the adverse impact of spurious covariances at large distances from a proxy location, which results from sample error in finite ensembles (Hamill et al., 2001). If localization is not applied, spurious covariances allow proxies to affect remote locations, which adversely affects the quality of the analysis. On the other hand, too-short localization length scales reduces the useful information that can be derived from the proxies. Therefore a balance is sought between minimizing sampling noise versus retaining useful proxy information.

We use the Gaspari-Cohn (Gaspari and Cohn, 1999) fifth-order polynomial with a specified cut-off radius for the localization function ($w_{loc}$ in (4)). See section S5 in the supplementary material for information on the characteristics of $w_{loc}$. A series of reconstructions are performed with a wide range of localization length scales. As with previous experiments, 51 Monte-Carlo realizations are carried out, each with 100 ensemble members assimilating 75% of proxy records. Results from the instrumental-era verification scores previously described are summarized in Table 5. We observe that the GMT trend is

underestimated and verification scores are significantly reduced when "too-small" localization radii are used, indicating the information on temperature provided by some proxy records is not properly incorporated in the reanalysis. In contrast, the trend is overestimated and verification scores are generally reduced without covariance localization. This is particularly the case for the CE score for the detrended GMT, sensitive to the amplitude in interannual variability. This skill measure is maximized for a localization radii within the 15000 to 25000 km range. A localization radius at the upper end of this range (25000 km) is preferable, as results from the other verification scores suggest that a skillful reconstruction is obtained with this covariance localization configuration. See Figure S4 in the supplementary material for an example where the 25000 km localization function is applied to a proxy record located in California, United States. We note that the optimal localization radius depends on a number of factors, such as ensemble size, the observation network and observation error characteristics.

## 4.3 Proxy data sets

Here we explore the impact of adding the large number of proxies from Anderson et al. (2019) (hereafter A19), which include the tree ring width chronologies from Breitenmoser et al. (2014) (hereafter B14), not strictly screened for climate sensitivity in contrast to the PAGES2k collection. Duplicate records between datasets are identified (based on correlation between co-located records and cross-referencing metadata) and eliminated. Priority is given to records found in the PAGES2k collection (see A19 for more details). Figure 8 shows the spatial and temporal distributions of the B14 records, which reveals enhanced coverage over eastern North America, southern Europe, boreal Eurasia and southern South America. Other additions, totaling 94 records, provide additional records in the Tropics (23 coral records), and an enhanced number of ice core records concentrated over Greenland and eastern Canadian Arctic (37 records) and Antarctica (26 records in West Antarctica and Drönning Maud Land). A few lower latitude ice core records (6 records) are also added in the Peruvian Andes and Tibetan Plateau, along with two higher latitude lake core records. From a temporal perspective, the addition of the B14 tree ring width records contributes a notable number of additional proxies back to 1000CE, more than double the number of records available for assimilation from 1500CE onward, up to a fourfold increase during the nineteenth and twentieth centuries.

In order to measure the impact with the best configuration, the reconstruction experiments reported in this section are carried out using seasonal PSMs with objectively derived seasonality for all records, with a bivariate formulation on temperature and precipitation for all TRW proxies, and univariate on temperature for all other proxies. The baseline reconstruction uses the PAGES2k-2017 proxies (as in section 3), which we compare to results first obtained with the addition of the B14 TRW records, and finally with the further addition of the coral, ice and lake core records from A19 (i.e. the full proxy database). Other trial reconstructions performed with the vastly expanded proxy network, not reported here, have shown that a well-calibrated GMT ensemble is obtained with a covariance localization cut-off radius of 25000 km. Next, we compare reconstruction results from this configuration to the baseline reanalysis.

Differences in correlation and CE associated with the addition of the B14 collection over the PAGES2k-2017 proxies show skill improvements in temperature reconstructions over the continental United States and Mexico, Europe, and the southern edge of the Tibetan Plateau (cf. Figs. 9g and h). Through the influence of significant spatial covariances with the added records, assimilation of the additional TRW records also leads to improved temperature skill over remote areas of the mid-latitude

Pacific and northern Atlantic oceans. The addition of records described in A19 has minimal additional impact overall, with the exception of modest increases in correlation and CE over Greenland (see Figs. 9i and j).

Hydroclimate verification is defined by a comparison of the reconstructed Palmer Drought Severity Index (PDSI) with the Dai (2011) product. We note here that the reconstruction is not directly related to the PDSI product used for verification as TRW forward models were calibrated on precipitation and not on PDSI as in Steiger et al. (2018). A comparison of the reconstructed PDSI between the prototype[3], the updated reanalysis of section 3 and a reconstruction carried out with the B14 TRW records and the additional coral, ice and lake core records (i.e. the full database) is shown in Figure 10. The PDSI is slightly improved in the updated reanalysis compared to the prototype (Figs.10g and h). Enhanced skill is noticeable over western North America, and over eastern Europe and Asia to a lesser degree. Decreased skill is found over the central plains of North America and along a narrow band along the Siberian Taiga. The impact of adding the Anderson et al. (2019) records is mostly found over the eastern part of the United states and over western Europe (Figs.10i and j). Finally, we note that this impact is due entirely to the B14 TRW records, as the additional coral, ice and lake core records from A19 do not significantly affect the PDSI reconstruction skill (from results of additional reconstruction experiments carried out to isolate this impact, not shown).

Examining the differences between reconstructions over the entire Common Era (Fig. 11), we see a significantly modified Northern Hemisphere temperature (NHMT) resulting from the assimilation of the additional proxies. A generally warmer NHMT is obtained throughout the Common Era, but most significantly during the LIA, worsening the agreement with reconstructions from other studies shown in Figure 2. A noticeable loss of variability is observed, confirmed by comparing spectra from both experiments (Fig. 11c). This loss of variability in the reconstruction using all proxies occurs at nearly all scales, underlining an adverse impact from assimilating B14 tree ring width proxies.

We now turn to verification in proxy space, which is the only source available prior to the instrumental period. Proxy estimates from reanalyses (estimated using the appropriate PSM) are compared directly to proxy observations. Here, reanalysis skill is assessed using independent (the 25% withheld from assimilation) proxies. We further restrict our analysis to verification against tree ring wood density proxies, as they are among the most reliable recorders of temperature in our database, as evidenced by the generally better fits to calibration temperature data obtained when calibrating the univariate PSMs. Also, these proxies provide good temporal coverage of the latter portion of the LIA into the industrial period as shown in Figure 1. The results, presented in Table 6, show distinctly larger skill scores for the experiment using PAGES2k-2017 proxies only compared to when all proxies are assimilated. Improved skill is observed for both periods of interest. Results from a third reconstruction experiment are also presented, where only a small fraction of B14 records are assimilated (B14 subset experiment in Table 6). A total of 188 records (out of the 2156 available) have been selected on the basis of their strong relationship to calibration temperature and precipitation data as determined from the correlation coefficient characterizing bivariate PSMs. Records with a calibration correlation above 0.6 are found to be located for the most part over the United States. Proxy verification results indicate an increase in skill in the representation of tree ring wood density proxies, as indicated by skill metric values only slightly lower than in the PAGES2k-2017 experiment. Spatial verification of temperature and PDSI (not shown) also suggest that some of the skill enhancements shown in Figure 10i and j are retained even when this small fraction of the B14 records

---

[3]The LMR prototype configuration has been used to reconstruct PDSI, a variable not included in H16, for the purpose of this comparison.

is considered. This suggests that the issues with the assimilation of the B14 records identified above can possibly be mitigated while maintaining some of the skill they provide toward enhanced temperature and hydroclimate reconstructions in local regions. Optimal selection of these records requires further careful attention, and could serve as the basis for future efforts.

## 5  Concluding summary

A paleoclimate reanalysis of the Common Era has been developed using an updated data assimilation framework. Results show significant improvement over the prototype Last Millennium Reanalysis presented in Hakim et al. (2016). An updated proxy database and implementation of proxy system models (PSMs) with improved realism are shown to be key contributors to the enhanced reanalysis. The main upgrade to the proxy database consist of a change from the community-standard of PAGES 2k Consortium (2013) to the more recent PAGES 2k Consortium (2017) data set, while the records described in Anderson et al. (2019) remain available for possible future enhancements to the proxy information used in the reanalysis. Moreover, new methods to map state variables to observations extend the prototype's linear univariate models calibrated on annual-mean temperature in two key aspects: accounting for seasonal dependencies of individual proxy records, and the modeling of tree-ring-width proxies using temperature and moisture as predictors. The encoding of proxy seasonality information within PSMs has also been refined by objectively determining the characteristic seasonal response of individual records, and by decoupling the seasonality for temperature and precipitation sensitivity for tree-ring-width.

Climate field reconstructions from a series of assimilation experiments carried out with various proxy and PSM configurations have been compared to available instrumental–era observation-based analyses, revealing notable improvements not only in the reconstructed global mean temperature in general, but also in reconstructed spatial fields. More skillful tropical Pacific temperatures are obtained primarily due to the updated set of coral records in the PAGES 2k Consortium (2017) collection. Improved temperature reconstructions over continental extratropical regions are the result of the newly implemented seasonal PSMs, combined with the forward modeling of tree-ring-width chronologies using a bivariate temperature-moisture formulation. Improvements are reflected not only in temperature reconstructions, but also in 500 hPa geopotential height and to some extent in hydroclimate variables such as the PDSI. Lastly, the introduction of the large collection of Breitenmoser et al. (2014) tree-ring-width chronologies, not screened for temperature sensitivity, appears to provide local skill enhancements in hydroclimate variables (e. g. PDSI over the eastern United States). However this is achieved at the expense of accuracy in the reconstruction of important features of pre-industrial climate such as the colder temperatures during the Little Ice Age. However, the generally positive impact of a simple *ad hoc* screening of the Breitenmoser et al. (2014) suggest that further improvements may be possible with a careful selection of tree-ring chronologies.

Results presented here, based upon regression PSMs, may serve as a reference for future efforts designed to assess the value of more comprehensive process-based PSMs in paleoclimate data assimilation research. Finally, we note that the version of the PDA system described here corresponds to the configuration used in the production release of the NOAA Last Millennium Reanalysis, available at https://atmos.washington.edu/~hakim/LMR/.

*Code availability.* The code used in the production of reanalyses is publicly available at https://github.com/modons/LMR.

*Data availability.* The output from the reanalysis and the required input data are available from https://atmos.washington.edu/~hakim/LMR/.

## Appendix A: Proxy system model characteristics

Features introduced in the updated LMR proxy modeling capabilities include a representation of the seasonal response to
climate drivers characterizing individual proxy records (i.e. proxy seasonality), as well as proxy system models (PSMs) that
include precipitation and temperature as driving variables for modeling tree-ring-width (TRW) records.

The first approach is to use univariate PSMs calibrated against temperature data, with proxy seasonality either defined from
the available proxy metadata or derived objectively using the method described in section 2.4.1. PSM performance is compared
using the Bayesian Information Criterion (BIC), defined as (Schwarz, 1978),

$$BIC = -2\ln(\hat{L}) + k\ln(n) \tag{A1}$$

where $\hat{L}$ is the maximized value of the likelihood function of the model, $n$ is the sample size and $k$ is the number of estimated
parameters in the model. We note that the second term in Eq. A1 represents a penalty for models with a larger number of
explanatory variables, i.e. a more complex model. This feature is particularly useful when comparing univariate and bivariate
models. Here we use the difference in BIC values between two models $\Delta BIC = (BIC_M - BIC_{ref})$, to determine the relative
accuracy of model $M$ over a reference. The model with the lowest BIC is preferred (i.e. a better fit to the data), hence a negative
$\Delta BIC$ indicates the superiority of the test model over its reference. Here, the seasonal PSMs are tested against the univariate
PSMs calibrated with annually-averaged temperatures as the reference. Significant evidence of the superiority of the test model
over its reference is obtained when $\Delta BIC <$ -2.0.

Table A1 presents a summary of $\Delta BIC$ results for records in each proxy category considered in LMR. The advantage
of seasonal PSMs is particularly significant for tree-ring wood-density chronologies, a proxy known for its strong seasonal
response (Briffa et al., 2004). Seasonal PSMs also provide improved fits to tree ring width data, although to a lesser extent
compared to density records. As indicated by the larger negative $\Delta BIC$ values, models based on objectively-derived seasonal
responses lead to more accurate descriptions of proxy data compared to those calibrated using metadata seasonality, even for
tree ring chronologies within the community-curated PAGES2k-2017 data set. These results suggest that the objectively-derived
seasonality information is noticeably different than in the metadata, particularly for tree ring records in the Breitenmoser et al.
(2014) (i.e. B14) data set, but also for those in PAGES 2k Consortium (2017) (i.e. PAGES2k-2017). More details on this aspect
are provided in the supplementary material. The use of objectively-defined seasonality improves upon the simple latitude-
dependent relationship described in Anderson et al. (2019), more consistent with records from the PAGES2k-2017 data set.
Apart from lake sediment records, which are also more accurately modeled with seasonal PSMs, Table A1 shows that PSMs for
other proxy types are not as sensitive to seasonality. In fact, the majority of the (tropical) coral records included in the current

database have metadata seasonality defined as annual already, as do the high-latitude ice core records. Note that some of these records originate from the collection described by Anderson et al. (2019), where seasonal metadata information is generally not available. As a result, these records are assumed to be annual.

In addition to seasonal models, other improvements involve the development of PSMs that add precipitation as an input variable for the modeling of TRW proxies as outlined in section 2.4.2. One approach consists of selecting the univariate models, either calibrated on temperature or moisture input, which best describe the proxy data. This "temperature or moisture" selection (abbreviated as "TorM") is performed on individual TRW records, and the resulting proportion of TRW proxies identified as temperature-sensitive is 56.4% versus 43.6% for moisture when metadata seasonality information is considered. This is compared to 36.8% temperature-sensitive versus 63.2% moisture-sensitive trees when seasonal responses are determined objectively. The latter option, leading to a larger proportion of moisture-sensitive records, is in better agreement with a comparable characterization performed by Steiger et al. (2018) on a similar set of TRW records.

A second approach consists of bivariate PSM formulation, where TRW depends on both temperature and precipitation (see Eq. 8. The $\Delta BIC$ results characterizing the univariate "TorM" and bivariate PSMs against their univariate temperature-only counterparts (as the reference) are summarized in Table A2. The negative mean $\Delta BIC$ values confirm the advantage of including moisture in TRW linear models. The evidence is more pronounced for the B14 records, perhaps not surprisingly given the larger proportion of moisture-sensitive records included in this data set. Nonetheless, the prevalent reduction in $BIC$ for models of PAGES2k-2017 trees suggests a non-negligible response to moisture despite the screening of records for temperature. The mean positive $\Delta BIC$ characterizing the bivariate models calibrated using metadata seasonality confirm that the assumption of identical seasonal responses for temperature and moisture is problematic for modeling tree ring growth, at least with these more complex models. On the other hand, allowing distinct representations of temperature and moisture seasonal responses in bivariate PSMs, as enabled by the goodness-of-fit objective determination of these responses, leads to significantly more accurate TRW modeling compared to univariate temperature PSMs.

*Author contributions.* R.T. developed the code related to all the updates discussed in the paper (generation of expanded proxy database, calibration and application of the new proxy forward models), configured and performed all the reanalysis experiments, analyzed the experimental results and wrote the manuscript with G.J.H. G.J.H. proposed the general design of the updated tree-ring proxy models, contributed the Kalman update and original versions of the verification codes. W.A.P. developed several key parts of the reanalysis code. K.A.H. collected and formatted the proxy records taken from sources other than the PAGES 2k collection, re-processed the tree ring data taken from Breitenmoser et al. M.P.E. contributed to the identification of proxy records to be included/excluded from the updated proxy database and suggested key code modifications. J.E.G. provided the PAGES 2k proxy data and advice on proxy system modeling. D.M.A. supervised the collection of proxy records and helped define the content of the proxy database. E.J.S. contributed advice on proxy system modeling, statistical measures, and helped define the content of the proxy database. D.N. contributed advice on proxy system modeling. All authors contributed to the final editing of the manuscript.

*Competing interests.* The authors declare that they have no conflict of interest.

*Acknowledgements.* The authors thank Nathan Steiger of the Lamont-Doherty Earth Observatory, Columbia University, for providing the PDSI data from the CCSM4 Last Millennium simulation, and Jörg Franke of the University of Bern, Switzerland, for providing the tree-ring data from the Breitenmoser et al. (2014) study. This research was supported by grants from the National Oceanic and Atmospheric Administration (grant NA14OAR4310176) and the National Science Foundation (grants AGS-1304263 and AGS-1702423 to the University of Washington). We acknowledge the World Climate Research Programme's Working Group on Coupled Modeling, which is responsible for CMIP. For CMIP the U.S. Department of Energy's Program for Climate Model Diagnosis and Intercomparison provides coordinating support and led development of software infrastructure in partnership with the Global Organization for Earth System Science Portals. CMIP5 data used in this paper may be obtained from the Earth System Grid Federation at http://esgf.llnl.gov/. Some calibration and verification data sets were provided by the NOAA/OAR/ESRL PSD, Boulder, Colorado, USA, from their website at https://www.esrl.noaa.gov/psd/.

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

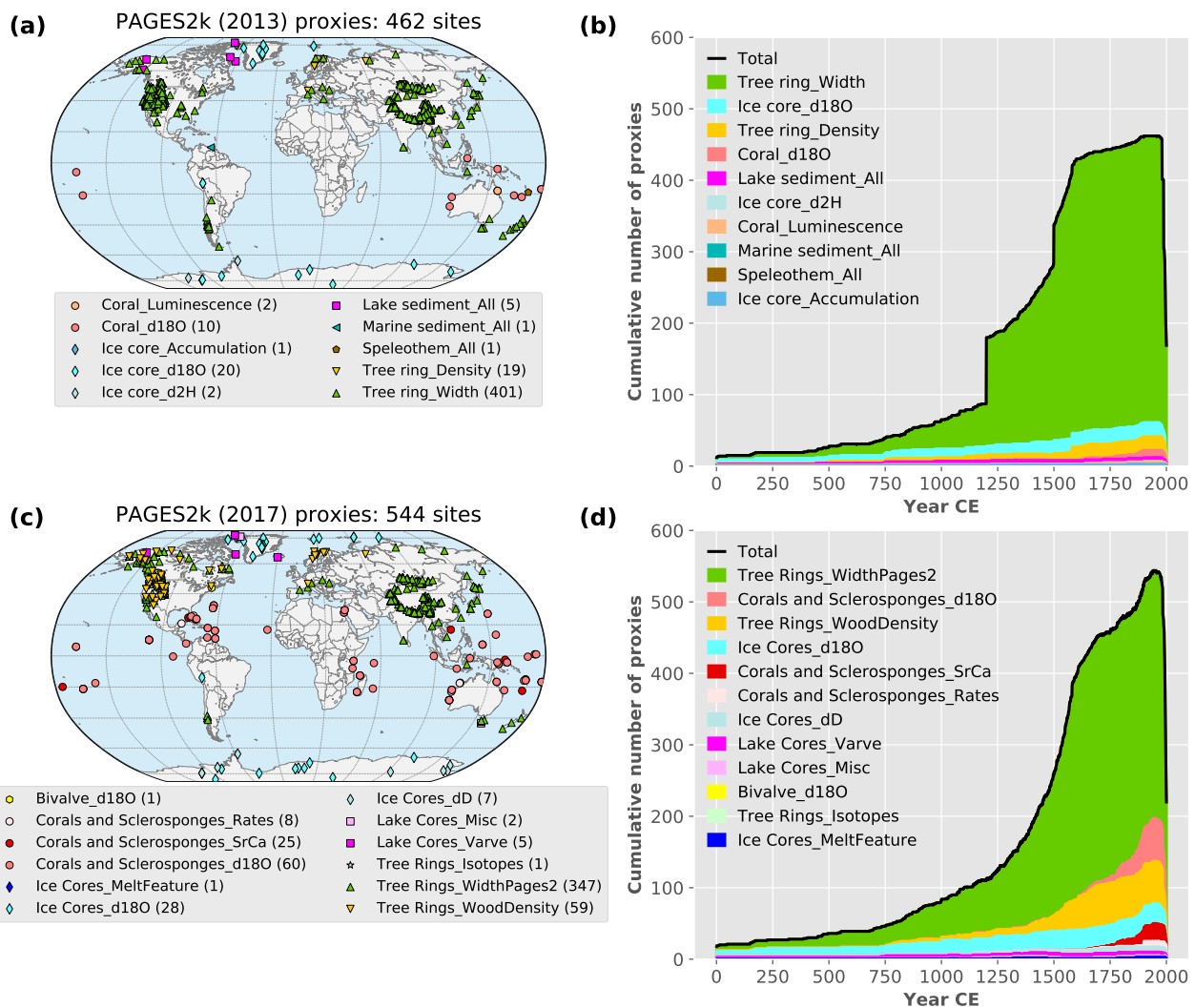

**Figure 1.** Locations (left column) and temporal (right column) distributions of proxy records available for assimilation (proxies for which linear PSMs calibrated with GISTEMP version 4 are available), (a) and (b) used in the prototype version, (c) and (d) LMR proxy database updated to PAGES 2k Consortium (2017) proxies.

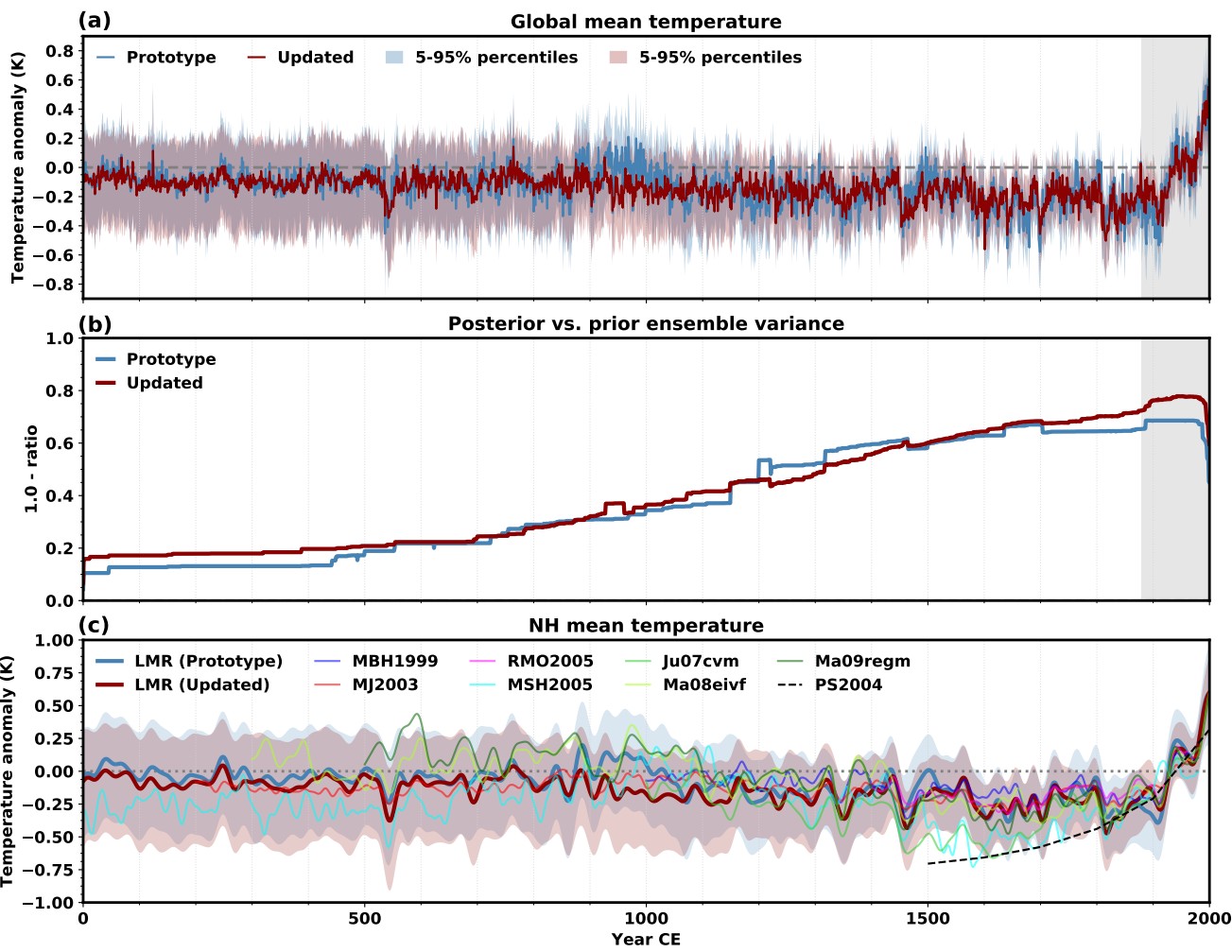

**Figure 2.** Comparison of the LMR global-mean 2 m air temperature (GMT) (a) grand ensemble mean (solid lines) and 5–95% percentile range (shading) from the prototype (blue) and updated (red) reanalyses over the Common Era, and (b) one minus the mean (across Monte-Carlo realizations) ratio of the posterior and prior GMT ensemble variance. (c) Comparison of the LMR Northern Hemisphere 2 m air temperature grand ensemble mean (solid lines) and 5–95% percentile range (shading) from the prototype and updated reanalyses with reconstructions from other authors: MBH1999: Mann et al. (1999), MJ2003: Mann and Jones (2003), RMO2005: Rutherford et al. (2005), MSH2005: Moberg et al. (2005), Ju07cvm: Juckes et al. (2007), Ma08eivf: Mann et al. (2008), Ma09regm: Mann et al. (2009), PS2004: Pollack and Smerdon (2004). All series in (c) represent anomalies (K) from the 1900–1980 mean, and have been smoothed with a 30-year low-pass Butterworth filter. The light gray shading in (a) and (b) indicate the verification period discussed in Fig. 3.

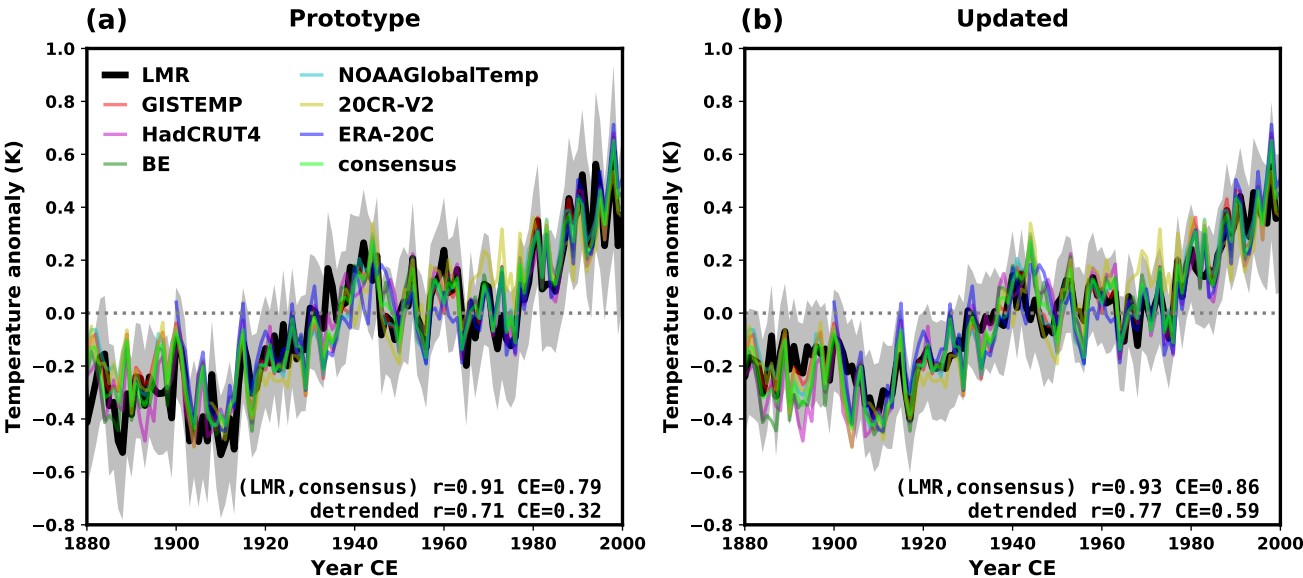

**Figure 3.** Comparison of LMR global-mean 2 m air temperature (GMT) (a) prototype and (b) updated reanalyses, against instrumental–era analyses (GISTEMP: NASA GISS surface temperature (Hansen et al., 2010); HadCRUT4: Hadley Center/Climate Research Unit at the University of East Anglia temperature data set version 4 (Morice et al., 2012); BE: Berkeley Earth surface temperature (Rohde et al., 2013); NOAAGlobalTemp:NOAA merged land-ocean surface temperature version 3.5.4 (Smith et al., 2008); 20CR-V2: NOAA twentieth century reanalysis version 2 (Compo et al., 2011); ERA-20C: ECMWF reanalysis of the twentieth century (Poli et al., 2016); Consensus: average of all but LMR). The gray bands show the LMR 5–95% percentile range. Verification correlation (r) and coefficient of efficiency (CE) values are shown at the bottom of each panel, for the original and detrended time series.

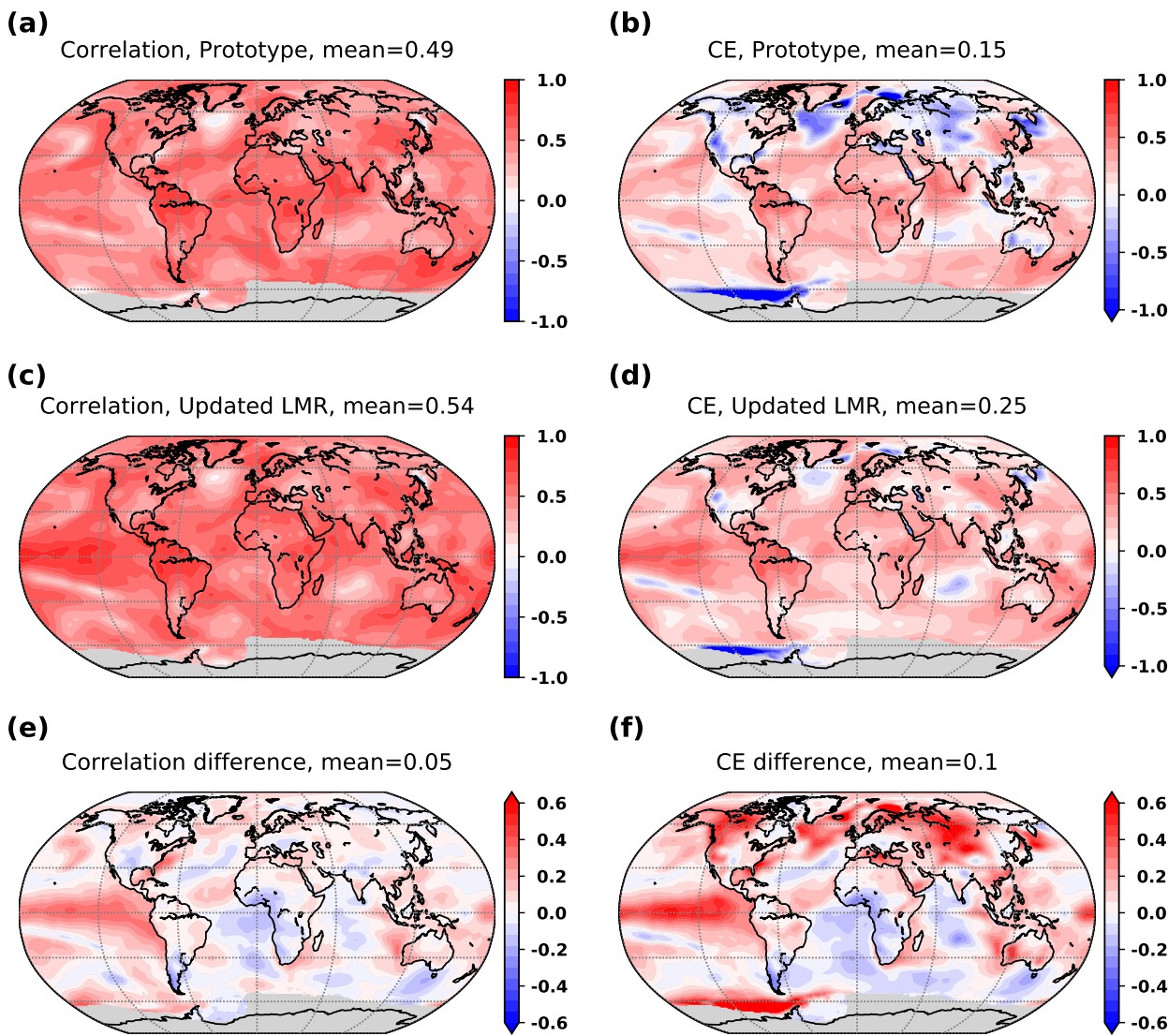

**Figure 4.** Verification of LMR 2m air temperature against the Berkeley Earth instrumental–era analysis over the 1880–2000 period. Shown are time series correlation (left column) and coefficient of efficiency (CE, right column), for (a) and (b) the prototype and, (c) and (d) the updated reanalysis. Differences in correlations and CE between the two experiments are shown in (e) and (f) respectively. Gray shading indicate regions with insufficient valid data for meaningful verification statistics.

**Instrumental-era verification for Z500**
**LMR vs. 20CRv2**

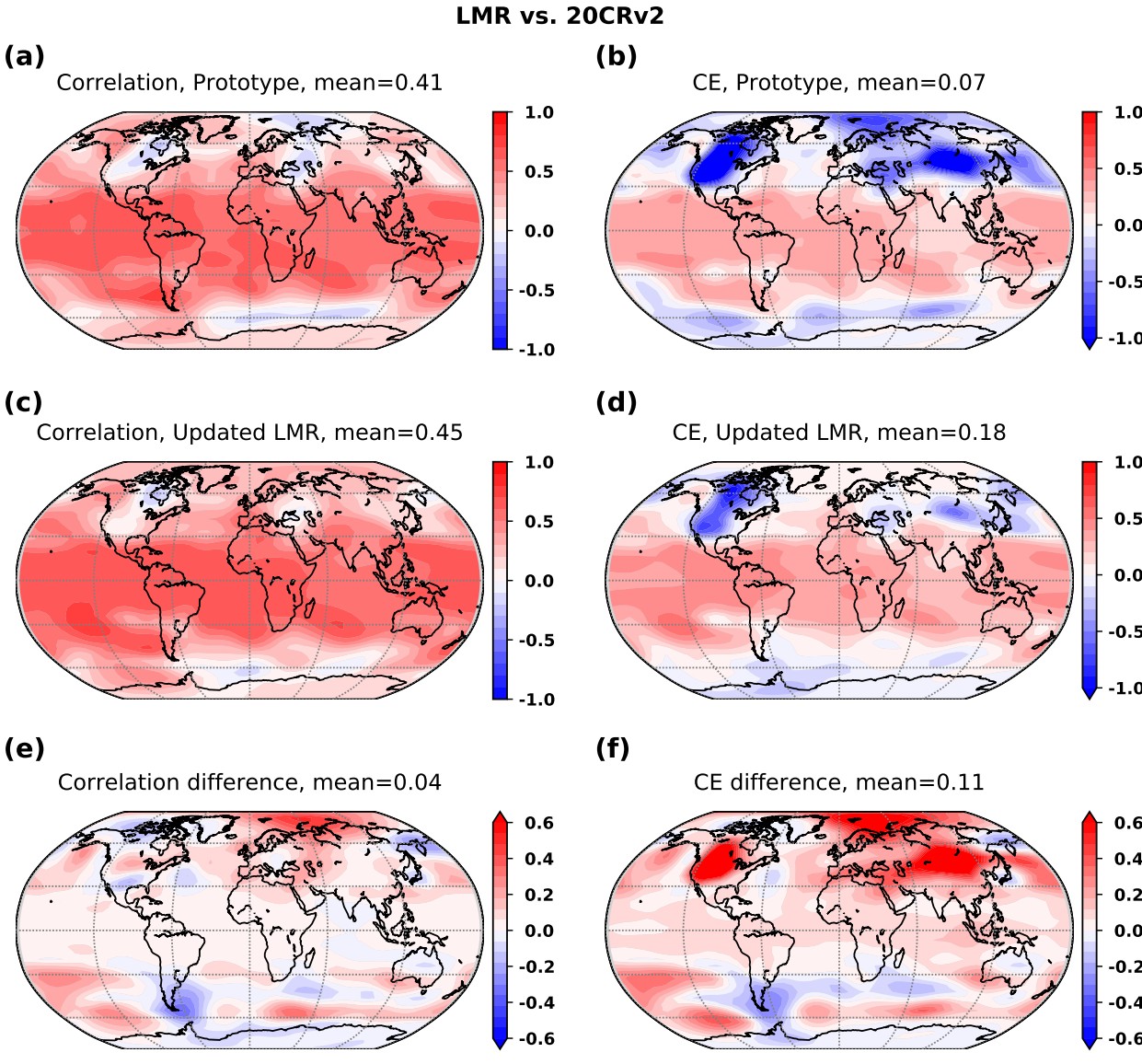

**Figure 5.** As in Fig. 4 except for the verification of LMR 500 hPa geopotential height anomalies against the 20CR-v2 reanalysis.

**Figure 6.** Verification of LMR temperature anomalies against the Berkeley Earth instrumental–era analysis, for experiments using PAGES2k-2017 proxies and univariate PSMs, with contrasting seasonalities. Shown are time series correlation (r) and coefficient of efficiency (CE), for (a) and (b) experiment 1: annual, (c) and (d) experiment 2: seasonality from the proxy metadata, and (e) and (f) experiment 3: objectively-derived seasonality. Differences in skill metrics are also shown, (g) and (h) between experiments 2 and 1, (i) and (j) between experiments 3 and 1.

## Instrumental-era verification for temperature
## LMR vs. Berkeley Earth

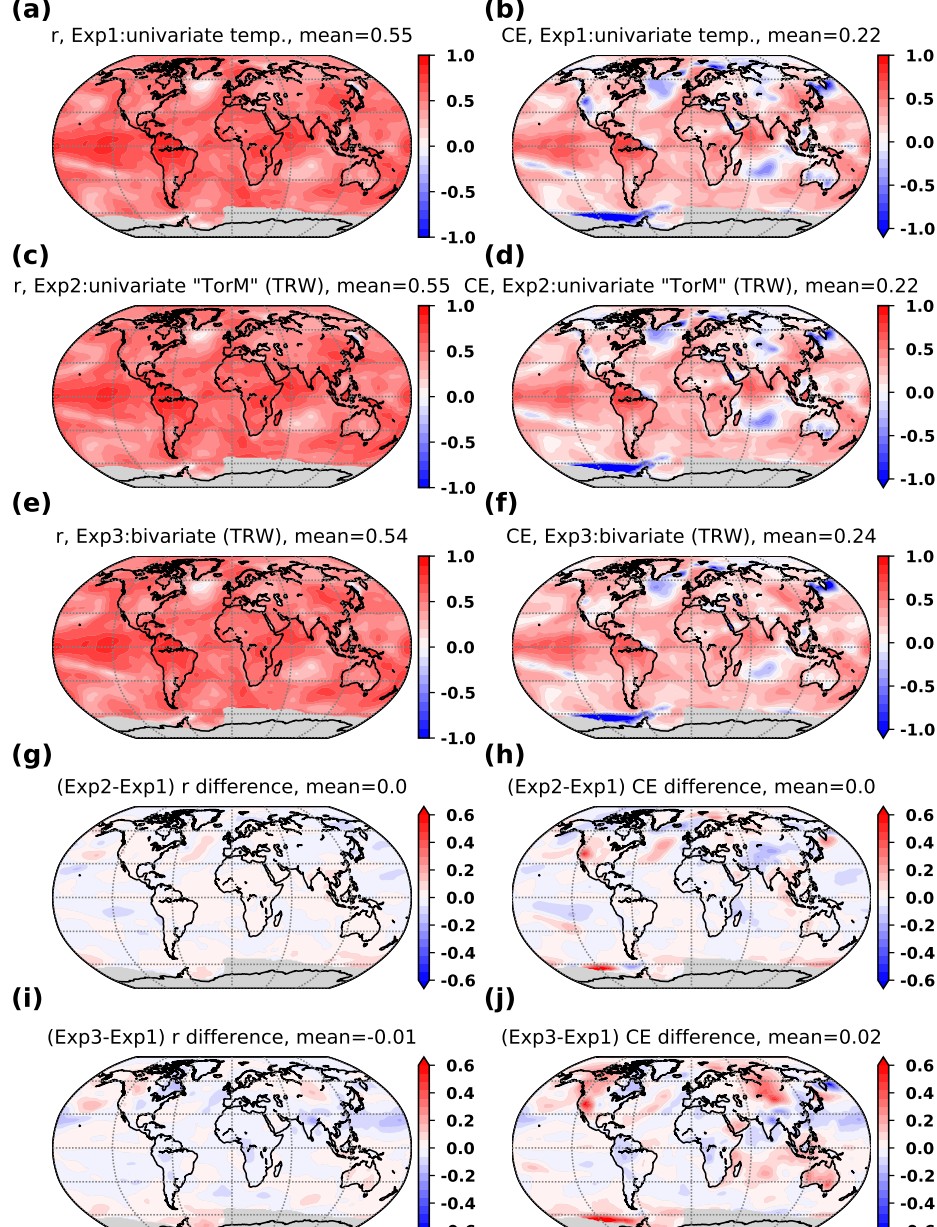

**Figure 7.** As in Figure 6, but comparing experiments performed using PAGES2k-2017 proxies with different PSM configurations for tree ring width proxies. (a) and (b) experiment 1: univariate on temperature for all proxies, (c) and (d) experiment 2: univariate with respect to temperature or moisture for TRWs, and (e) and (f) experiment 3: bivariate on temperature and moisture for tree ring widths. Differences in skill metrics are shown, (g) and (h) between experiments 2 and 1, and (i) and (j) between experiments 3 and 1. All reconstructions are based on objectively-derived seasonal PSMs.

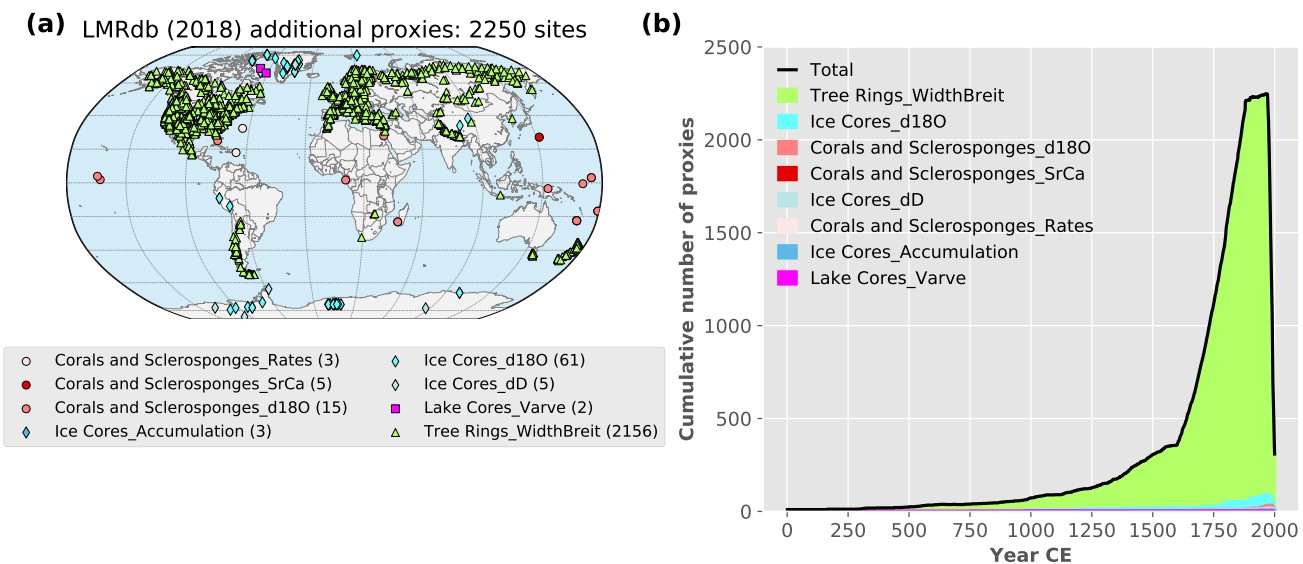

**Figure 8.** Locations (a) and temporal distributions (b) of the additional proxies from Anderson et al. (2019) considered for assimilation, including the tree ring chronologies from Breitenmoser et al. (2014). As in Fig. 1 only records available for assimilation (proxies for which regression based PSMs can be calibrated) are shown.

## Instrumental-era verification for temperature
## LMR vs. Berkeley Earth

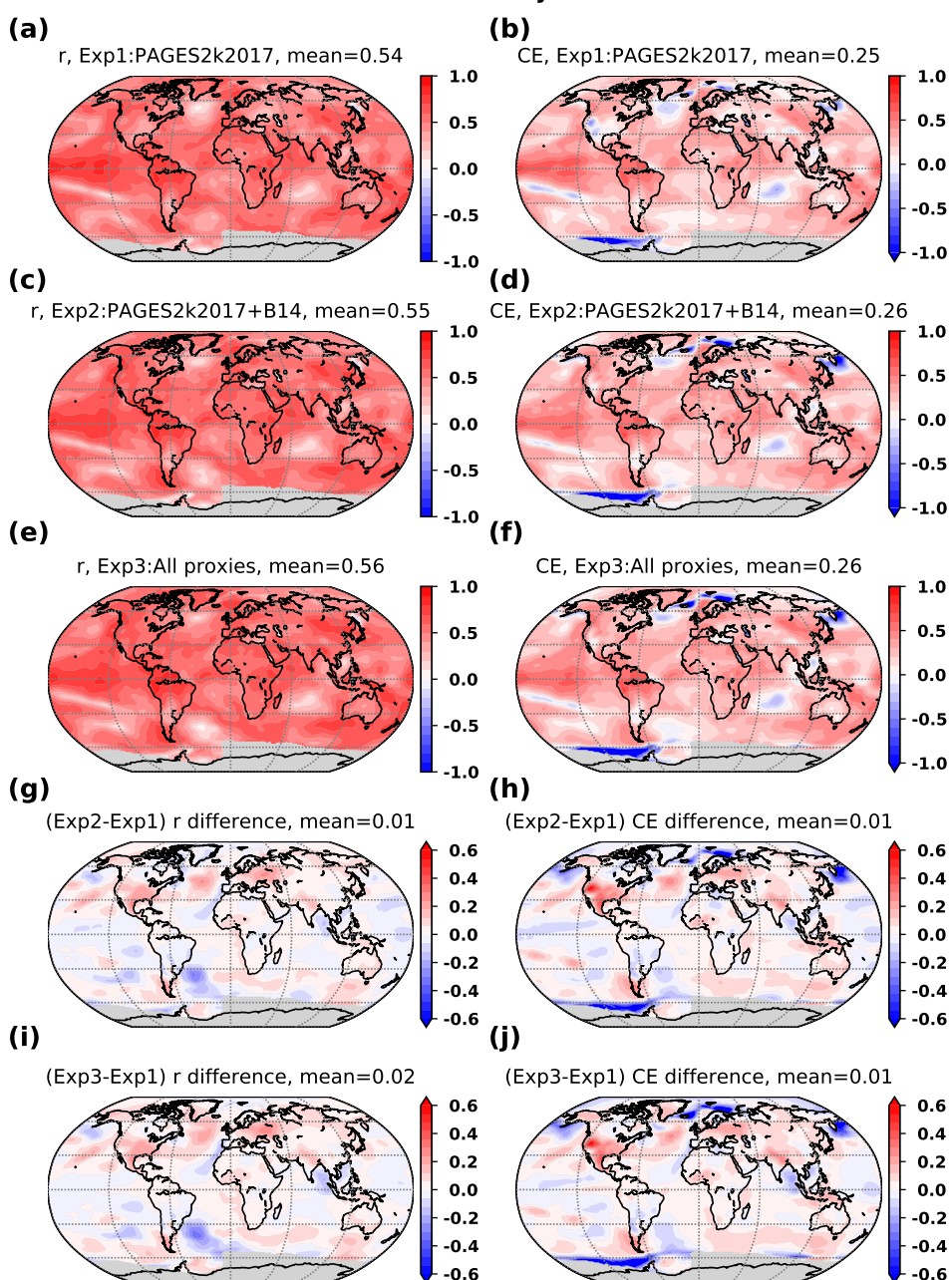

**Figure 9.** As in Figure 6, but comparing experiments performed with different proxy networks: (a) correlation (r) and (b) CE for experiment 1: PAGES 2k Consortium (2017) proxies only, (c) and (d) experiment 2: with the addition of tree ring chronologies from Breitenmoser et al. (2014), and (e) and (f) experiment 3: with all proxies in the updated LMR database. The differences in correlation and CE between experiments 2 and 1 are shown in (g) and (h) respectively, and between experiments 3 and 2 in (i) and (j). Notice the latter is different to Figure 6, where differences between experiments 3 and 1 are shown.

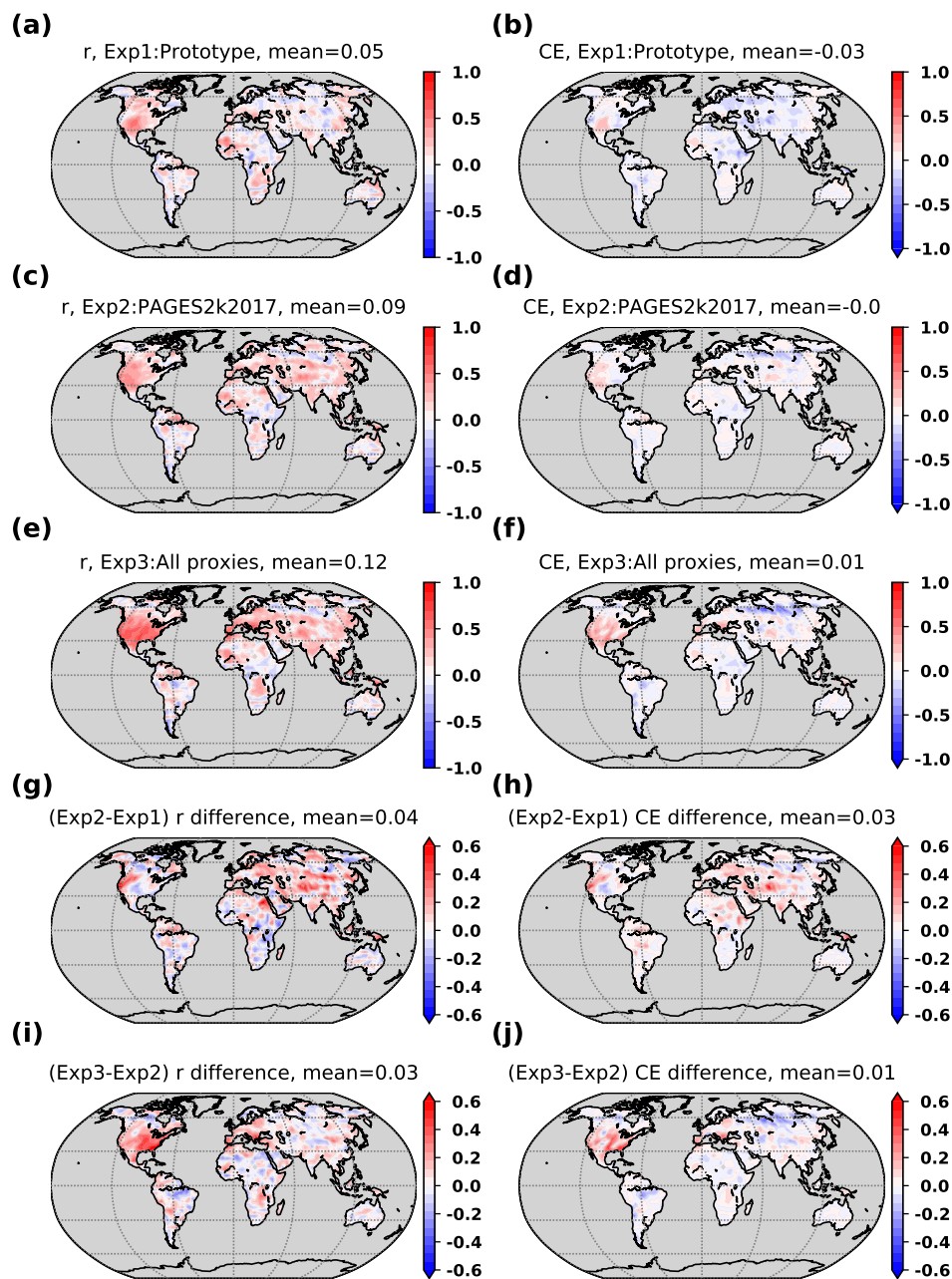

**Figure 10.** Similar to Figure 9, but comparing PDSI reconstructions against the Dai (2011) analysis for experiments performed with different proxy networks: (a) correlation and (b) CE for experiment 1: prototype reanalysis from H16, experiment 2: PAGES 2k Consortium (2017) proxies, (e) and (f) experiment 3: with further the addition of tree ring chronologies from Breitenmoser et al. (2014) and the coral, ice and lake core records from Anderson et al. (2019) (i.e. the full proxy database). The differences in correlation and CE between experiments 2 and 1 are shown in (g) and (h) respectively, and between experiments 3 and 2 in (i) and (j).

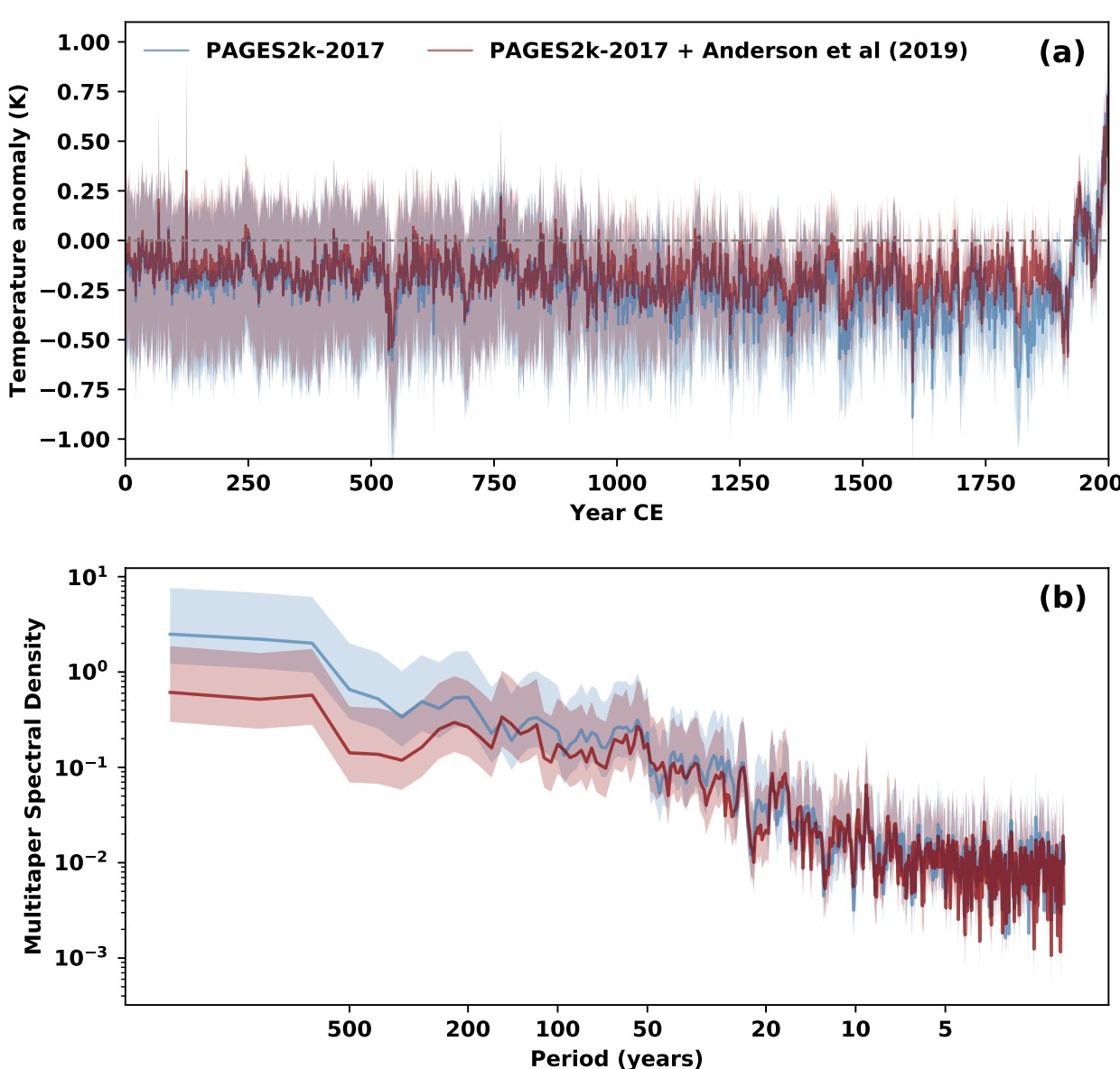

**Figure 11.** (a) Northern Hemisphere temperature (NHMT) grand ensemble mean (solid lines) and 5–95% percentile range (shading) from experiments performed with PAGES2k-2017 proxies (in blue), and with the addition of proxies from Anderson et al. (2019) (in red), (b) Spectra of NHMT grand ensemble mean from both experiments (solid lines), along with the $\chi^2$ 95% highest density regions (shading).

**Table 1.** Summary of instrumental–era verification results for the prototype and updated reanalyses. Verification scores shown are correlation (r) and coefficient of efficiency (CE), for the annual global mean temperature (GMT) and detrended GMT verified against the consensus of instrumental–era analyses, the global mean of gridpoint r and CE characterizing the spatially reconstructed temperature, 500 hPa geopotential height (Z500) and Palmer Drought Severity Index (PDSI). LMR spatial temperature is verified against the Berkeley Earth analysis (Rohde et al., 2013), Z500 is verified against the 20CR-V2 reanalysis (Compo et al., 2011) and PDSI is verified against the Dai (2011) analysis.

| Reanalysis | Annual GMT | | Detrended GMT | | Spatial temperature | | Spatial Z500 | | Spatial PDSI | |
|---|---|---|---|---|---|---|---|---|---|---|
| | r | CE | r | CE | r | CE | r | CE | r | CE |
| Prototype | 0.91 | 0.79 | 0.71 | 0.32 | 0.47 | 0.10 | 0.41 | 0.07 | 0.05 | -0.03 |
| Updated | 0.93 | 0.86 | 0.77 | 0.59 | 0.52 | 0.22 | 0.45 | 0.18 | 0.09 | 0.00 |

**Table 2.** Verification of LMR prototype and updated reanalyses against independent (withheld from assimilation) proxies. Skill scores shown are the median of distributions for correlation (r), the fraction of proxy records characterized by a positive $\Delta CE$ (%+CE), and the median of the $\Delta CE$ distribution, where $\Delta CE$ is the difference in the coefficient of efficiency (CE) between the posterior (reanalysis) and the prior. The median of the ensemble calibration ratio (ECR) distribution is also shown. Statistics are compiled over 51 Monte-Carlo realizations, and cover different time periods, including the 1880–2000 PSM calibration period.

| Verification period (years of Common Era) | Prototype | | | | Updated reanalysis | | | |
|---|---|---|---|---|---|---|---|---|
| | r | %+CE | $\Delta CE$ | ECR | r | %+CE | $\Delta CE$ | ECR |
| 1–499 | 0.00 | 56.0 | 0.00 | 0.78 | 0.03 | 55.9 | 0.00 | 0.96 |
| 500–999 | 0.08 | 62.1 | 0.01 | 1.00 | 0.13 | 65.3 | 0.02 | 1.00 |
| 1000–1499 | 0.11 | 63.0 | 0.01 | 1.10 | 0.16 | 67.3 | 0.05 | 1.06 |
| 1500–1879 | 0.14 | 64.1 | 0.02 | 1.06 | 0.28 | 72.7 | 0.10 | 1.02 |
| 1880–2000 | 0.23 | 72.6 | 0.03 | 0.97 | 0.40 | 82.7 | 0.13 | 0.89 |

**Table 3.** Summary of instrumental-era verification results for reconstruction experiments performed with various PSM configurations. Verification scores shown are the trend over the twentieth century (in K/100yrs), correlation (r) and coefficient of efficiency (CE), for the annual global mean temperature (GMT) and detrended GMT verified against the consensus of instrumental–era analyses. The GMT trend in the consensus of instrumental-era analyses is 0.56 K/100 yrs.

| PSM configuration | GMT trend | Annual GMT | | Detrended GMT | |
|---|---|---|---|---|---|
| | | r | CE | r | CE |
| Prototype | 0.61 | 0.91 | 0.79 | 0.71 | 0.32 |
| Univariate - temperature (annual) | 0.85 | 0.93 | 0.61 | 0.74 | 0.39 |
| Univariate - temperature (seasonal meta.) | 0.72 | 0.93 | 0.77 | 0.73 | 0.43 |
| Univariate - temperature (seasonal obj.) | 0.72 | 0.93 | 0.80 | 0.75 | 0.51 |
| Univariate - temperature or moisture (TRW) (seasonal meta.) | 0.71 | 0.92 | 0.78 | 0.72 | 0.44 |
| Univariate - temperature or moisture (TRW) (seasonal obj.) | 0.74 | 0.93 | 0.77 | 0.74 | 0.48 |
| Bivariate - temperature and moisture (TRW) (seasonal meta.) | 0.62 | 0.93 | 0.84 | 0.76 | 0.50 |
| Bivariate - temperature and moisture (TRW) (seasonal obj.) | 0.60 | 0.93 | 0.86 | 0.77 | 0.54 |

**Table 4.** Verification of LMR reconstructions against independent (withheld from assimilation) proxies, for experiments using various PSM configurations. Skill scores shown are the median of distributions for correlation (r), the fraction of proxy records characterized by a positive $\Delta$CE (%+CE), and the median of the $\Delta$CE distribution. Statistics are compiled over 51 Monte-Carlo realizations, for two distinct periods: 1880–2000 (PSM calibration period) and 0–1879 (pre-calibration period).

| PSM configuration | 1880–2000 | | | 1–1879 | | |
|---|---|---|---|---|---|---|
| | r | %+CE | $\Delta$CE | r | %+CE | $\Delta$CE |
| Univariate - temperature (annual) | 0.28 | 75.2 | 0.05 | 0.17 | 66.0 | 0.03 |
| Univariate - temperature (seasonal meta.) | 0.32 | 78.7 | 0.06 | 0.21 | 69.6 | 0.04 |
| Univariate - temperature (seasonal obj.) | 0.34 | 80.6 | 0.09 | 0.21 | 69.4 | 0.06 |
| Univariate - temperature or moisture (TRW) (seasonal meta.) | 0.30 | 76.1 | 0.06 | 0.19 | 67.7 | 0.04 |
| Univariate - temperature or moisture (TRW) (seasonal obj.) | 0.33 | 77.6 | 0.08 | 0.19 | 66.3 | 0.04 |
| Bivariate - temperature and moisture (TRW) (seasonal meta.) | 0.32 | 77.9 | 0.07 | 0.20 | 68.1 | 0.04 |
| Bivariate - temperature and moisture (TRW) (seasonal obj.) | 0.36 | 78.9 | 0.11 | 0.22 | 66.0 | 0.06 |

**Table 5.** Twentieth century trend of global-mean temperature (GMT), correlation (r) and coefficient of efficiency (CE), for the annual and detrended GMT, as well as the global mean of the spatial (i.e. gridpoint) r and CE of reconstructed temperature verified against the consensus of instrumental-era analyses, for reconstruction experiments performed with covariance localization using various localization cut-off radii $L_R$. Verification statistics for an experiment without covariance localization are also shown for comparison. Results from the prototype are shown for reference. The GMT trend in the consensus of instrumental-era analyses is 0.56 K/100 yrs.

|  | $L_R$ 5000 km | $L_R$ 10000 km | $L_R$ 15000 km | $L_R$ 25000 km | $L_R$ 35000 km | $L_R$ 45000 km | No localization | Prototype (no localization) |
|---|---|---|---|---|---|---|---|---|
| Trend (K/100 yrs) | 0.17 | 0.31 | 0.40 | 0.49 | 0.51 | 0.56 | 0.60 | 0.61 |
| Annual GMT r | 0.92 | 0.93 | 0.93 | 0.93 | 0.93 | 0.93 | 0.93 | 0.91 |
| Annual GMT CE | 0.46 | 0.71 | 0.82 | 0.86 | 0.87 | 0.87 | 0.86 | 0.79 |
| Detrended GMT r | 0.74 | 0.77 | 0.77 | 0.77 | 0.77 | 0.76 | 0.77 | 0.71 |
| Detrended GMT CE | 0.35 | 0.53 | 0.59 | 0.59 | 0.58 | 0.56 | 0.54 | 0.32 |
| Mean spatial r | 0.36 | 0.46 | 0.50 | 0.52 | 0.52 | 0.53 | 0.53 | 0.47 |
| Mean spatial CE | 0.11 | 0.17 | 0.19 | 0.22 | 0.21 | 0.21 | 0.20 | 0.10 |

**Table 6.** As in Table 4, but statistics compiled for tree-ring wood-density (MXD) proxies only, and for experiments using the PAGES2k-2017 proxies only, PAGES2k-2017 with the addition of all proxies from A19 (All proxies), and PAGES2k-2017 plus only a subset of A19 records obtained after removing all but 188 TRW records from B14 (B14 subset). See text for selection details. Skill scores are the median of the correlation (r) distributions, the fraction of proxy records characterized by a positive $\Delta$CE (%+CE), and the median of $\Delta$CE distributions. Statistics are compiled over the 51 Monte-Carlo realizations, for the following periods: 1880–2000 (PSM calibration period) and 1600–1879 (pre-calibration period with a significant number of MXD records and covering a significant portion of the Little Ice Age).

| PSM configuration | 1880–2000 | | | 1600–1879 | | |
|---|---|---|---|---|---|---|
| | r | %+CE | $\Delta$CE | r | %+CE | $\Delta$CE |
| PAGES2k-2017 | 0.62 | 93.2 | 0.37 | 0.58 | 91.4 | 0.39 |
| All proxies | 0.43 | 88.0 | 0.18 | 0.46 | 95.4 | 0.26 |
| B14 subset | 0.56 | 92.5 | 0.30 | 0.53 | 93.8 | 0.34 |

**Table A1.** Mean differences in Bayesian Information Criterion ($\Delta$BIC) corresponding to PSMs for records within the proxy categories considered in LMR, between models calibrated using proxy seasonal responses from the metadata or derived objectively during calibration, with respect to the reference of annual seasonality. Calibration dataset: GISTEMP v4.

| Proxy types | Number of records | Seasonal (metadata) | Seasonal (objective) |
|---|---|---|---|
| Tree ring width (PAGES2k-2017) | 347 | -1.34 | -4.84 |
| Tree ring width (Breitenmoser et al.) | 2156 | -1.72 | -5.24 |
| Tree ring wood density | 59 | -23.28 | n/a |
| Coral $\delta^{18}O$ | 75 | +0.02 | n/a |
| Coral Sr/Ca | 30 | -0.01 | n/a |
| Coral Rates | 11 | +0.03 | n/a |
| Ice core $\delta^{18}O$ | 89 | +0.02 | n/a |
| Ice core $\delta D$ | 12 | 0.00 | n/a |
| Ice core accumulation | 3 | 0.00 | n/a |
| Ice core melt | 1 | 0.00 | n/a |
| Lake core varve | 7 | -0.52 | n/a |
| Lake core misc. | 2 | -2.32 | n/a |
| Bivalve $\delta^{18}O$ | 1 | 0.00 | n/a |
| Tree ring $\delta^{18}O$ | 1 | +11.81 | n/a |

**Table A2.** Mean differences in Bayesian Information Criterion ($\Delta$BIC) for tree ring width univariate "temperature or moisture" and bivariate PSMs, calibrated using metadata seasonality or derived objectively during calibration, against their respective univariate temperature-only PSMs as reference. Calibration datasets: GISTEMP v4 and GPCC v6.

| PSM formulation | Seasonal (metadata) | | Seasonal (objective) | |
|---|---|---|---|---|
| | PAGES 2k trees | Breitenmoser trees | PAGES 2k trees | Breitenmoser trees |
| Univariate - temperature or moisture | -0.86 | -1.41 | -2.59 | -6.65 |
| Bivariate | +2.63 | +1.73 | -2.35 | -6.88 |