# Peer review of "Last Millennium Reanalysis with an expanded proxy database and seasonal proxy modeling"

_Climate of the Past, 2018_

## Referee Comment (RC1) · Anonymous Referee #1 · 19 Oct 2018

General comments

This manuscript describes a new annually resolved 2000-year reanalysis, for which paleodata are assimilated into a climate simulation ensemble, using a Kalman filtering method. The paper is well written and especially interesting from the methodological points of view. First, the way seasonal information is assimilated into annual averaged and second, because the performance of multiple proxy data sets is compared. Probably, data-assimilation techniques are the future in the field of climate reconstruction and hence the work is well suited for Climate of the Past. The use of the data assimilation method with an appended state to account for seasonality is innovative. Thus, it should not be hidden in the appendix but appear more prominent in the main text, written in a way that is understandable to the average climate of the past reader and not

just data assimilation specialists. Otherwise, it is not clear to the reader how seasonal information can be used if the state vector just contains annual averages.

In contrast to the methodological part and sound statistical validation, the paleoclimatic discussion needs to go more into depth and the authors need to be more critical about their own results. The authors compare their new data set to the previous version (Hakim et al. 2016) and conclude this new version would be an overall improvement. This conclusion is based on validation statistics in the 20th century. However, the new data set lost all multi-decadal to centennial variability in the global mean temperature and does not show a warmer medieval period nor a cooler "little ice age" anymore. The authors do not discuss this issue at all. The paper suggests that this new reanalysis version would present the more likely global mean temperature evolution of the past 2000 years although it is in contrast to what most other reconstructions and paleodata records suggest. In the introduction of this study, the authors write: "Hakim et al. 2016 [is] . . . in good agreement with previous reconstructions of northern hemisphere mean temperature" and this first version had similar low frequency variability as previous reconstructions. The loss of low-frequency variability is most likely a consequence of the proxy data sets used, because this is the major change in the new version. Many of the tree-ring chronologies in Breitenmoser et al. 2014 are not climate sensitive at all or moisture sensitive. As precipitation does not show any low-frequency variability in contrast to temperature, it is a logical consequence that using covariance information from moisture sensitive trees to correct temperature data leads to a loss of low frequency variability. A second reason may be the use of proxy data with dating uncertainties, such as ice cores, in an annual reconstruction. These proxies probably do not have age errors in the 20th century validation period but become just noise if they have an age offset of one or a few years further back in the past. The authors just conclude that using moisture sensitive data leads to improved reconstruction skill, although this is only true in the 20th century validation period but not in the pre-instrumental period, most user of this data set will be interested in.

In the current version, the global mean temperature evolution is the reappearance of the famous "hockey stick" in climate science. After all the discussion, the hockey stick was rising 20 years ago, I would not publish this as a state-of-the-art temperature reconstruction, especially not without a discussion and not if it is an artefact of un-screened input data.

I see two options, the first would involve minor revisions and the second major revisions:

1. It must be stated prominently (already in the abstract) that this reanalysis should not be used or considered to have the correct multi-decadal and centennial variability and that the global mean time series over the last 2000 year potentially has serious issues. The discussion needs to include all problems of data set, too and ideas how to overcome them in the future. In general, the paper should be put more into a context of methodological improvements to achieve better products in the future instead of claiming this would be nearly the prefect reanalysis for the past 2000 years.

2. A proper screening of the data needs to be introduced that prohibits the assimilation of non-climatic information, which has just spurious correction with observations in the short window of overlapping instrumental and proxy data (a minimum of 25 data points has been used in this study, page 5, line 2). These records will have little weight in the assimilation procedure due to large residual variance, but hundreds of little errors probably produce significant noise. Probably, precipitation limited proxies and proxies with age errors have to be removed or treated specifically, too. These are just some ideas and it will need many improvements and new experiments to find and solve the problem.

A difficulty in the review process is that the input data has not been published, yet. Hence, it is not possible to properly judge the input data base. However, it appears to be basically the Breitenmoser et al. 2014 data set with a few coral and ice core records added. Why do you not simply refer to this first publication and give citations for the

additional records or wait until the Anderson et al. paper is published?

In general, the decrease of skill further back in time is not discussed sufficiently. It should also be discussed why forcings are not important and what the consequences of unforced simulations ensembles are for the final product, especially further back in the past when the proxy network is sparse. I suggest to evaluate the spatial skill of the reanalysis in the 20th century but with the spatial proxy network at multiple time slices, e.g. 0 AD, 500 AD, 1000 AD, 1500 AD. Additionally, not using forced simulation offers the potential to use them in the validation procedure. It could be checked if temporal and spatial patterns of known past events or periods are well represented in the reanalysis, e.g. spatial moisture distribution after eruptions (Iles and Hegerl, 2015).

Finally, it would be interesting to see a map of the regression residuals to get an idea how many paleodata records have significant influence in the assimilation procedure and which are basically ignored because they have no climate information. Additionally, I would like to know how many records in the PAGES2Kv2 data base have expert information on seasonality? I would be interested to read how well the expert-based seasonality in the PAGES data base agrees with the objective assessment in this study. Probably, the experts did a similar search for highest correlation, maybe just including more possible combinations of growing season months. I was surprised to read that the authors use an extended fall period (JJASON)? Is there any reference for trees which are limited by climatic conditions in these autumn months.

It would be favorable to store the data at a world data center and not at a personal homepage.

Technical corrections

Abstract:

- skill score increase in percent is misleading. It is easy to have a large relative increase if scores were very low in the comparison data set, e.g. in Z500 where CE improves

from very negative to less negative the increase in percent is large but the skill is still negative!

- be more precise what is meant with "ensemble characteristics"

Introduction:

- Line 17: apart from paleoclimate with annual observations, the forecast model is a third important component (this is even written in the Methods section)

Methods:

- Page 7, line 6ff: I do not understand why the calibration is done with a different gridded data set than the validation. Both data sets a based on largely overlapping instrumental observations and therefore clearly not independent.

- On Page 4 line 14: How many ensemble members?

- On Page 4 line 14 it is written that Hakim et al. 2016 worked "with the same randomly drawn ensemble members used for every year in the reconstruction, whereas on page page 5, line 28 it is written for this study: "each using a different randomly chosen 100–member ensemble". Are both studies consistent and is each year build on different randomly chosen ensemble members?

- Page 5, line 1: "Only records for which a PSM can be established are shown . . .". What do you mean by "shown"? There is no reference to a figure. Do you want to say that only records meeting these criteria are assimilated?

- Page 5, line 8: Breitenmoser et al. 2014 is not screened for any climate sensitivity.

- Page 6, Proxy modeling: It should be repeated here over which period the regression coefficients are calculated. As many data points as available in the overlapping period with instrumental data but minimum 25 pairs of x and y?

- Page 7, line 30: Is there a reference that any tree-ring proxy responds to an extended

fall period (JJASON)? The given references point to common growing seasons from May to August in the northern hemisphere. Why are not all combinations of growing season length tested and the optimum is chosen? In the PAGES data base there are also various different length of growing seasons defined.

- Page 8, line 26: "local" should better be "grid box"

The updated reanalysis:

- Page 9, line 5: Can you explain the localization better? Does a cut-off radius of 25000 km mean that each proxy influences basically the entire globe?

- Page 9, line 7: Mention that the reference for the skill score is climatology

Figures:

- some text is too small that it cannot be read in a print version

- figures should have consistent font types

---

## Referee Comment (RC2) · Anonymous Referee #2 · 18 Nov 2018

General comments: The study seeks to perform the climate reanalysis for the past millennium. Although the topic is of great interest to a broad readership of this journal, this reviewer believes the methodology, analysis and the final LMR product presented herein are too premature to be acceptable for formal publication, let alone for its stated purpose to serve as the basis for the first publicly released NOAA last millennium re-analysis. My specific comments are given below.

Major comments: 1. It is misleading for this study (and its prototype in H16) to call the DA method used in this as an ensemble Kalman filter (EnKF). As in Evensen (1994) and subsequent studies, the primary promise of the EnKF is the use of flow dependent background error covariance represented by the forecasting ensemble. The current so-called "offline" DA method has none of that: the ensemble perturbations are randomly

[Figure]

sampled from a past-millennium climate simulation that has no relation to the prior estimate, and the same set of sampled perturbations were used at all analysis times. This method used in this study is similar to the commonly used 3D-Var method for numerical weather prediction with static background error covariance, and is arguably less advanced than 3D-Var since 3D-Var in NWP used the dynamic model to propagate the previous cycle's analysis as the prior before the analysis. The current so-called "offline" DA method neither cycles the analysis nor the ensemble perturbations, with the stated reason that the forecast model is not good enough to do either.

2. If the forecast model is not good enough to cycle the mean analysis or the analysis uncertainties to provide the best estimate of the prior estimate and related prior uncertainties, why would this model(s) be good at all for use as the prior estimate that the LMR reanalysis depends critically on? In this regards, it is premature to state (line 10) that the "LMR employs the ensemble data assimilation to optimally blend the information from the proxies and the climate model data". The current method is more like an objective analysis method.

3. It is not clear whether the authors are aware that the traditional static 3D-Var methods also derive the background covariance from an ensemble of perturbations, as is traditionally called "the NMC method" using the sampled forecast divergence between different lead times from many realizations. The Kalman filter update in this case is equivalent to the variational update using the 3D-Var algorithm, though again the 3D-Var in NWP cycles the analysis and forecast during data assimilation, which is the most basic function in combining the model and data.

4. The validation performed in this study for the prototype and updated LMR "reanalysis" with several existing 20th-century reanalysis is misleading at best. The quality of the LMR reanalysis for the 20th century is the least issue given the availability of the modern much more advanced reanalysis and given the exponentially increased number of proxies or model instrumental observations. The validation currently focuses exclusively on the 20th century says little on the quality and performance of the LMR

products, in particular over the early period when the proxy data are scarce. A more appropriate validation can potentially be done in two objective methods: (1) perform the 20th century "reanalysis" through thinning the observation density and maybe also degrading the observation accuracy to those representation of different periods of the past millennium; and/or (2) performing observing system experiments in which a certain number of observations are not assimilated but reserved for independent validation (or all of them in cross validation).

5. The use of a 2,5000-km covariance localization is highly questionable for the use of a 100 sets of fixed ensemble perturbations. At midlatitudes, this is amount to the observation impacts across the entire global latitude belt. The use of a fixed set of 100 sample perturbations also means a high rank deficiency over such a large area with this large localization distance.

6. On a related note, the current final LMR reanalysis derives from the mean of 51 such 100-member analyses, should it be the same if the 5100 samples of perturbations are used simultaneously in the Kalman filter update given the Kalman filter used is largely a linear operation? How much is the result sensitive to the choice of this arbitrary number of sample perturbations? It is also worth noting the the NMC method used for 3D-Var uses singular value decomposition to make it full rank. Such a approach is different from (and likely more advantageous over) the current Kalman filter update using purely non-envolving static ensemble covariances.

7. More generally, it is unclear what is the purpose of such as hastily done LMR reanalysis products with such ad-hoc DA approaches and the not-good-enough forecast models? The so-derived climate trend is almost certainly depending too much on the climate models used as a prior and ensemble sampled perturbations (and maybe the assumed climate forcings used in these models), as well as the density of observations over different periods. It could do more harm if such a premature reanalysis product is used or misused and if it were publicly released through NOAA, unfortunately. A more careful vetting of the products, and a more concerned effort in refined DA methodology

are warranted before NOAA sanctioned such a product as reanalysis, in this reviewer's opinion.

---

## Referee Comment (RC3) · Anonymous Referee #3 · 27 Nov 2018

Summary: This paper presents various sensitivity and verification tests of an updated LMR product targeting multiple climate state variables over the Common Era. The updates to the product include both methodological choices (seasonal assessments of the proxy sensitivities and tests of multiple statistical PSM formulations for univariate and bivariate dependencies) and increases in the proxies included in the network. While the abstract gives the impression that the paper serves as a the basis for the public release of a new LMR product, the paper is generally as I have stated: a large number of sensitivity and verification tests that feel more exploratory than definitive.

General remarks: I have two main concerns about the paper, which together require that the manuscript undergo major revisions before it is acceptable for publication. The

first is the character of the derived reconstruction and the unsatisfactory verification of the product using only observational data. The second is the use of multiple ad hoc methodological choices, none of which are reasonably justified or widely tested (which strikes me as strange given the extensive number of sensitivity tests that the authors have performed). I expand on each of these points below.

1. I am struck by the comparison in Figure 2a and the little attention the authors give to the differences between the previous LMR product and the newer version (not to mention the complete lack of comparison between either of these results and other temperature reconstructions). The GMT from the newer product looks almost like white noise and has lost not only the multi-decadal to centennial variability in the first product, it is also likely at odds with the now large collection of global and hemispheric temperature reconstructions spanning the last millennium or more. The authors not only need to spend more time discussing this issue, they also need to compare their results to the collection of large-scale temperature and hydroclimate reconstructions currently available. This is not only to place their results within the now lengthy body of work in this area, but it is essential for them to do more to verify their results beyond the comparisons they make to observational data. While the latter is important and useful, it is not enough. Incidentally, the authors do perform validation exercises on a withheld period of observational data and using withheld proxy series, but that work is buried in the supplemental and not adequately discussed in this context. More should be made of those efforts, which strengthen the authors results with truly out-of-sample validation experiments. Incidentally, I do not think the use of CE is the same as it is traditionally used in the paleo literature, given that the latter approach requires a true cross-validation period. The authors should clarify this point.

Similar to the above suggestions, I think it is further important for the authors to derive validation experiments for the sparsely sampled periods early in the proxy network (e.g. deriving reconstructions using only subsets of the proxies that extend back to specific time intervals). This would go a long way toward helping to better understand the loss

of proxy information back in time. This is partially addressed by the variance exercise the authors perform, but more can be done. Recons for temporal subsets of the proxy network would in fact be more useful than the MC sampling of the proxy network that the authors perform, given that it would be systematic and inform a direct question about the influence of the declining proxy network.

2. Here is a quick list of choices the authors have adopted that are not accompanied by any justifications or sensitivity discussion:

a. Use of 100-member ensemble

b. Use of the CCSM4 last millennium simulation as the prior

c. Use of 51 MC realizations

d. Use of a proxy sampling scheme based on 75% of the proxy records

e. Degradation of the model resolution to a ∼5x5 grid

All of these choices undoubtedly influence the derived LMR product. Some of them can be justified based on discussions in the literature. Some of them require empirical demonstrations. All of them come across as ad hoc. I would also venture to guess that the LMR results are more dependent on a couple of these choices than the other dependencies that the authors more systematically test. It is therefore essential that the authors do a better job of justifying these choices and convincing the reader that they are either reasonable choices or chosen based on some methodological/logistical rationale.

I should specifically mention the use of the CCSM4 as the prior. The authors say nothing about how their results might depend on the model prior and whether they have tested alternative last-millennium simulations in their analysis. This is an obvious question and the authors need to address it.

A few small details are also worth noting:

Pg. 2, Ln. 5-6: What does "synthesizing information" mean? This is vague and I am not even sure the statement is true. There are lots of central challenges of paleoclimate science, and it is arguable that what the authors are alluding to is one of them. This strikes me as an unsupported justification for what the authors subsequently say they are attempting to do.

Pg. 2, Ln. 30-32: This is a much more mundane objective than the sense given in the abstract. Are the authors attempting to release a shiny new LMR product or should this be seen as an iterative verification step toward some improved effort down the line?

Pg. 8, Ln. 9: The use of precipitation is not justified and concerning. First, precipitation is almost never the variable associated with moisture sensitivity in trees - some measure of soil moisture is. It is therefore not clear why the authors used precipitation and how it influences their results. Why not use a more conventional variable like PDSI? Secondly, how do the characteristics of precipitation influence the results? Does it matter that precip is likely not Gaussian and that it has limited spatial and temporal covariance structure? Is the use of precip perhaps adding to the loss of low-frequency variance in this new LMR product? My guess is that this specific choice has a large impact on the derived reconstruction and the use of precip is not justified in any way.

Figures: In general, there is a lot of small text in the figures that is hard to read and also rather confusing and messy. This could be cleaned up a lot and the digestibility of the figures could be improved. The many colorbars are also unnecessary in many plots when one would do.

---

## Author Comment (AC1) · 18 Jan 2019

**Authors' Responses to Anonymous Reviewer 1**

We thank the referee for thorough and insightful comments on the manuscript. The comments have challenged us to take a more complete look at the numerous reconstruction experiments we performed, and as a result, we have gained a more comprehensive perspective on the results.

**The paleoclimatic discussion needs to go more into depth and the authors need**

[Figure]

**to be more critical about their own results. The authors compare their new data set to the previous version (Hakim et al. 2016) and conclude this new version would be an overall improvement. This conclusion is based on validation statistics in the 20th century. However, the new data set lost all multi-decadal to centennial variability in the global mean temperature and does not show a warmer medieval period nor a cooler "little ice age" anymore. The authors do not discuss this issue at all. The paper suggests that this new reanalysis version would present the more likely global mean temperature evolution of the past 2000 years although it is in contrast to what most other reconstructions and paleodata records suggest.**

We agree with the referee's suggestion that results should be framed in the context of other reconstructions, and a discussion focused on the long-term perspective (not limited to the 20th century) of the updated reanalysis be included in the manuscript. To address this issue, we have prepared figures showing comparisons over the entire Common Era of LMR results (updated reanalyses and prototype) and other available reconstructions of the Northern Hemisphere (NH) temperature. Figure 1 in this document shows a comparison of LMR results (prototype and updated reanalysis) with other published reconstructions. These chosen reconstructions are presented in the IPCC AR4 and AR5 reports, restricted to those representing the entire hemisphere and extending in time well into the 20th century. Also, a low-pass filter is applied on all results to better highlight low frequencies. First, this comparison shows that most other reconstructions are found within the bounds of the LMR ensemble, indicating a general agreement between the different products, at least within the bounds of uncertainty as defined from LMR. However, as pointed out by the referee about the reconstructed global-mean temperature (GMST) presented in the submitted paper,and also reflected in the NH-mean temperature results shown here, three periods with differences between the LMR reconstructions are highlighted. These are a colder medieval period in the updated reanalysis, most notably during the 875–1050CE period, and warmer

temperatures during the 1600–1700CE and 1810–1920CE periods. We also note that the updated LMR is among the warm outliers during these cold periods compared to the prototype and most other reconstructions. Additional work has been undertaken to identify the reasons behind the above-mentioned differences. Initial findings are twofold.

First, we have concluded that the reconstructed colder temperatures during the medieval period, compared to the prototype LMR, are primarily rooted in the change from PAGES 2k Consortium (2013) to the more recent PAGES 2k Consortium (2017) proxy data. Indeed, it appears that a distinctly warmer medieval period isn't a prominent feature of the new collection, as indicated by the global temperature composites presented in PAGES 2k Consortium (2017). As this dataset reflects the community's most stringent evaluation of proxy records suitable for temperature reconstructions, we believe that the lack of a "classic" medieval warm period in our updated reconstructions of GMST and NH-mean temperature is not necessarily a glaring deficiency, but rather reflects the inherent ambiguities in defining this feature, as discussed in Diaz et al (2011).

Second, the referee's comment has motivated us to revisit our numerous experiments, as well as to perform additional test reconstructions, with a focus on results during the Little Ice Age (LIA). This exercise has allowed the identification of a notable sensitivity of reconstructed NH-mean temperatures to the set of assimilated tree ring width records (see the response to the next comment).

Following the referee's suggestion, a revised Section 3 will include a more complete discussion of these results, supported by a figure similar to Fig. 1 to better contextualize the updated LMR results against previous reconstructions.

**The loss of low-frequency variability is most likely a consequence of the proxy data sets used, because this is the major change in the new version. Many of**

**the tree-ring chronologies in Breitenmoser et al. 2014 are not climate sensitive at all or moisture sensitive. As precipitation does not show any low-frequency variability in contrast to temperature, it is a logical consequence that using co-variance information from moisture sensitive trees to correct temperature data leads to a loss of low frequency variability.**

The referee raises some valid points here. Key elements within the data assimilation (DA) framework related to the possible issues raised here are the forward models (here the proxy system models, or PSMs) used to estimate proxy observations from a model prior, and the observation error variance assigned to the proxies, i.e. the $R_k$ terms in equations 4 in the submitted manuscript. Within a DA framework, there are no fundamental reasons why the inclusion of records with sensitivities other than to temperature would lead to a deterioration to reconstructions of temperature, provided that PSMs properly account for the proxy sensitivities and observation error variances representative of the observation uncertainty are specified. However, it is acknowledged that these required conditions may not always be trivial to achieve. Nonetheless, initial LMR results (i.e the prototype) suggested that our approach has the ability to delineate proxy records with weaker sensitivity to climate by assigning relatively larger observation error variances $(R_k)$, resulting in such records only weakly influencing the reanalysis results even though they are assimilated. Also, a motivation for developing the bilinear approach to model tree-ring width (TRW) data was to gain an ability to seamlessly handle the more complex sensitivities to temperature and/or moisture of these chronologies. With this approach, in principle, TRW records can be assimilated without having to make a binary decision whether each record is dominantly sensitive to temperature or moisture, or having to screen records out a priori. However, we acknowledge that relying on simple regression-based PSMs opens up the process to the influence of spurious correlations between noisy data. With a large number of proxies considered, some records will invariably be characterized by somewhat overestimated confidence, i.e. too-small error variance, and therefore overly weighted in the update.

Additional test reconstructions, performed in response to the referee's comment, suggest that this issue is partly responsible for the warmer conditions during the LIA. Figure 2 presents results from a new experiment where a reduced number of Breitenmoser TRW records have been assimilated (updated LMR in the figure). Only the records with a bilinear calibration correlation above 0.4 have been considered. A significant sensitivity of the NH-mean temperature during the LIA is observed, primarily resulting in cooler temperatures during the 1600–1700CE and 1810–1920CE periods compared to results in Fig. 2. As a result, this new reanalysis exhibits a greater consistency with the LMR prototype and other reconstructions during the 1500–1900 CE period. The absence of a prominent Medieval Warm Period (MWP) remains however, indicating that the inclusion of unscreened records is not the sole reason for the reduced low frequency variability in the updated LMR.

**A second reason may be the use of proxy data with dating uncertainties, such as ice cores, in an annual reconstruction. These proxies probably do not have age errors in the 20th century validation period but become just noise if they have an age offset of one or a few years further back in the past. The authors just conclude that using moisture sensitive data leads to improved reconstruction skill, although this is only true in the 20th century validation period but not in the pre-instrumental period, most user of this data set will be interested in.**

This is an interesting suggestion. We tested this hypothesis by performing an additional experiment in which all ice cores records were withheld from assimilation. The results do not support the hypothesis put forward by the referee however. GMST and NH-mean temperatures exhibit similar multi-decadal variability compared to reconstructions which include ice core information. The main difference consists of a modified long-term trend, showing a flatter temporal evolution over most of the Common Era prior to the 20th century warming, worsening the agreement with the other reconstructions. The primary role of ice core proxy data is to rather anchor the millennial-scale cooling characterizing the pre-industrial era.

**In the current version, the global mean temperature evolution is the reappearance of the famous "hockey stick" in climate science. After all the discussion, the hockey stick was rising 20 years ago, I would not publish this as a state-of-the-art temperature reconstruction, especially not without a discussion and not if it is an artefact of unscreened input data.**

We agree. See responses to the comments above. In the light of results outlined above, the revised manuscript will present an alternate updated LMR, showing a greater level of low frequency variability which exhibit a better agreement with other reconstructions. The final selection of this updated reanalysis will be informed not only by the level of agreement with other reconstructions, but on the basis of objective validation of results involving verification in proxy space using independent (i.e. withheld from assimilation) records. Progress has been made in the identification of an alternate configuration of the reanalysis, and results will be described in the revised manuscript.

**I see two options, the first would involve minor revisions and the second major revisions: 1. It must be stated prominently (already in the abstract) that this reanalysis should not be used or considered to have the correct multi-decadal and centennial variability and that the global mean time series over the last 2000 year potentially has serious issues. The discussion needs to include all problems of data set, too and ideas how to overcome them in the future. In general, the paper should be put more into a context of methodological improvements to achieve better products in the future instead of claiming this would be nearly the prefect reanalysis for the past 2000 years. 2. A proper screening of the data needs to be introduced that prohibits the assimilation of non-climatic information, which has**

**just spurious correction with observations in the short window of overlapping instrumental and proxy data (a minimum of 25 data points has been used in this study, page 5, line 2). These records will have little weight in the assimilation procedure due to large residual variance, but hundreds of little errors probably produce significant noise. Probably, precipitation limited proxies and proxies with age errors have to be removed or treated specifically, too. These are just some ideas and it will need many improvements and new experiments to find and solve the problem.**

We appreciate these comments and suggestions. As outlined above, we chose to follow a path inspired by the second suggestion. A complete review of results was undertaken and new reconstructions experiments have been carried out. The newly gained perspective has led us to consider alternate reanalysis configurations which address the issues identified by the referee. Suggestions by the referee were carefully considered and integrated in the design of our latest experiments. However, we wish to underline our firm belief that an approach that seeks to simply "prohibit the assimilation of non-climatic information" and remove information from precipitation limited proxies, as suggested by the referee, is not the preferred framework in which to seek improvements in LMR reconstructions. Rather, improved forward models (PSMs), describing more accurately the relationships between climate variables and proxies, in addition to improved characterization of observation errors, are the key aspects where improvements can be achieved. We hope to convince the referee that our work on the seasonal PSMs and the development of bilinear PSMs for tree-ring proxies are worthwhile contributions to the former, while further refinements to the latter are still needed. We now believe that an approach involving some screening of the proxies available in our database, preferably less stringent than suggested by the referee, represents a compromise between the need to incorporate useful proxy information while minimizing the adverse effects of noise in the data. Our efforts will be reported in the revised version of our manuscript.

[Figure]

**A difficulty in the review process is that the input data has not been published, yet. Hence, it is not possible to properly judge the input data base. However, it appears to be basically the Breitenmoser et al. 2014 data set with a few coral and ice core records added. Why do you not simply refer to this first publication and give citations for the additional records or wait until the Anderson et al. paper is published?**

The manuscript is now available online at:

https://datascience.codata.org/article/10.5334/dsj-2019-002/

**In general, the decrease of skill further back in time is not discussed sufficiently.**

This is a great comment. We will address this using our framework enabling an assessment of reanalysis performance in proxy space. The revised manuscript will include a more prominent presentation and discussion of proxy verification results in the main text. This characterization should also be framed in the context of uncertainties in reconstructions. This capability is enabled by LMR through the availability of ensemble member information. This complementary perspective will also be incorporated in the revised version of the paper.

**It should also be discussed why forcings are not important and what the consequences of unforced simulations ensembles are for the final product, especially further back in the past when the proxy network is sparse.**

We are not sure how the referee has come to believe that forcings are not important. In fact the model simulations from which we draw prior information do include

forcings, such as pre-industrial greenhouse gas and aerosol variability, including the effects of volcanic eruptions. This point is mentioned in the manuscript. However, to bring the context into greater focus, we have found that an important characteristic of prior information within our DA framework is the amount of variance characterizing the simulations. We have found that the greater variance generally characterizing the "Last Millennium"-type simulations (which include the forcings listed above) provide for more accurate reconstructions, compared to using simulations performed without the influence of external forcings (as in the "pre-industrial" or piControl CMIP5 protocol). We also have generally refrained from using simulations which cover the 20th century warming to dispel the notion that we are "cooking the books" when reconstructing temperature trends. In our framework, temporal information (trends) come entirely from weighted information from the proxies.

**I suggest to evaluate the spatial skill of the reanalysis in the 20th century but with the spatial proxy network at multiple time slices, e.g. 0 AD, 500 AD, 1000 AD, 1500 AD.**

This is a good suggestion. We intend to move, and expand, the proxy verification results that are currently in the supplementary material into section 3 of the main body of the paper.

**Additionally, not using forced simulation offers the potential to use them in the validation procedure. It could be checked if temporal and spatial patterns of known past events or periods are well represented in the reanalysis, e.g. spatial moisture distribution after eruptions (Iles and Hegerl, 2015).**

This is also a good suggestion, however we believe that such efforts are outside the

scope of the current work. The suggestion will be considered in future efforts.

**Finally, it would be interesting to see a map of the regression residuals to get an idea how many paleodata records have significant influence in the assimilation procedure and which are basically ignored because they have no climate information.**

We agree, however a concise presentation of this is challenging, due to the varied nature of the proxy data. We will attempt to present this information as clearly as possible in a revised supplemental material. However we believe this would be addressed in a more informative way by a formal proxy impact study, which is intended to be the subject of another paper.

**Additionally, I would like to know how many records in the PAGES2Kv2 data base have expert information on seasonality? I would be interested to read how well the expert-based seasonality in the PAGES data base agrees with the objective assessment in this study. Probably, the experts did a similar search for highest correlation, maybe just including more possible combinations of growing season months.**

For the 2017 publication, the PAGES 2k consortium requested that each data certifier assess the seasonality of the temperature response and report its basis. In the LiPD format McKay and Emile-Geay (2016), this information is encoded in the `climateInterpretation_seasonality` and `climateInterpretation_basis` metadata properties. All records in the database include seasonality information, either as letters (JJA) or numbers ([6 7 8]) indexing calendar months. When the basis is reported, it is either from "first principles" (e.g.

trees are known to grow in local summer, which in most cases is synonymous with June July August), or from a search for the highest correlation. When the basis is not reported, the reader is referred to the publication documenting each record, which in most cases uses a mix of first principles and search for highest correlation, similar to the approach used in this paper. A key difference between the expert assessment of seasonality and the one done in our paper lies in the choice of target datasets. Studies focusing on individual series tend to be more careful about selecting an instrumental dataset appropriate for calibration (e.g. local GHCN station, rather than GISTEMP grid box average). These choices of target datasets are likely contributing to differences in the seasonal window determined via this process.

The choice of calibration datasets in LMR is driven by the need to uniformly process a large number of globally distributed records, hence the more general, perhaps not optimal selection as is possible when one has to consider a single or few records.

We also wish to point out that the information requested by the referee on the differences between the expert-based seasonality in the PAGES data base and objective assessment in this study is already provided in the supplemental information accompanying the main manuscript (Figure S1).

**I was surprised to read that the authors use an extended fall period (JJASON)? Is there any reference for trees which are limited by climatic conditions in these autumn months.**

Consideration of this period has been motivated following D'Arrigo et al (2005), and the fact that some seasonal responses found in the PAGES2k metadata extend to fall months, as suggested by the results shown in Figure S1a in the supplemental material accompanying our submission. It is interesting to note however that the objectively-derived seasonal responses determined by the approach described in our paper leads

to less emphasis on those fall months compared to the PAGES2k expert data.

**It would be favorable to store the data at a world data center and not at a personal homepage.**

We completely agree with the referee on this point. The LMR team has engaged interactions with the project's sponsor (NOAA) to identify a suitable storage location and access point, but these have yet to be identified, hence the reference to a personal homepage at this point.

Technical corrections

**Abstract: skill score increase in percent is misleading. It is easy to have a large relative increase if scores were very low in the comparison data set, e.g. in Z500 where CE improves from very negative to less negative the increase in percent is large but the skill is still negative!**

Point well taken. However, our emphasis has been about quantifying differences with respect to our main benchmark, the LMR prototype, to highlight improvements. Therefore we remain convinced that the formulation used is appropriate. The fact that some skill scores remain characterized by negative values is not hidden and becomes quite clear in the core of the text.

**Abstract: be more precise what is meant with "ensemble characteristics".**

We agree that this concept should be more accurately defined. It will be rectified in the

revised manuscript.

**Introduction, Line 17: apart from paleoclimate with annual observations, the forecast model is a third important component (this is even written in the Methods section).**

We agree that this statement in the Introduction could be interpreted to mean that the prior data has less importance in the DA context. A statement more accurately conveying the importance of the various components will be included in the revision.

**Methods, Page 7, line 6ff: I do not understand why the calibration is done with a different gridded data set than the validation. Both data sets a based on largely overlapping instrumental observations and therefore clearly not independent.**

The use of GISTEMP as the calibration dataset has largely been motivated by the fact that a larger number of proxy records could be calibrated (larger number of records with sufficient overap with valid calibration data) compared to other datasets. Therrefore, a larger number of proxies may participate to the reanalysis. We in fact perform validation using all datasets at our disposition, including the calibration dataset. Skill metrics show small differences among the various results, but remain in general agreement. Here we have chosen the Berkeley Earth dataset because it provides a greater spatial coverage, comparable to the calibration dataset, therefore providing a larger sample of spatial verification results.

**Methods, Page 4 line 14: How many ensemble members?**

The information is provided later in the paper, in Section 2.3, where details of the configuration are listed.

**Methods, Page 4 line 14 it is written that Hakim et al. 2016 worked "with the same randomly drawn ensemble members used for every year in the reconstruction", whereas on page page 5, line 28 it is written for this study: "each using a different randomly chosen 100–member ensemble". Are both studies consistent and is each year build on different randomly chosen ensemble members?**

Both statements are consistent, in that the first statement describes how the ensemble members for a given reanalysis realization are used to populate prior information in the time domain, whereas the second statement points out that different sets of states are used to populate the prior ensemble for different Monte-Carlo realizations of reconstructions. We will attempt to clarify this in the revised manuscript.

**Page 5, line 1: "Only records for which a PSM can be established are shown ...". What do you mean by "shown"? There is no reference to a figure. Do you want to say that only records meeting these criteria are assimilated?**

We wish to point out that a reference to Figure 1 is included in the previous sentence. We will modify the text to make sure the reader is not confused about what we are refering to in this sentence.

**Methods, Page 5, line 8: Breitenmoser et al. 2014 is not screened for any climate sensitivity.**

Point taken. We will modify the text accordingly.

**Methods, Page 6, Proxy modeling: It should be repeated here over which period the regression coefficients are calculated. As many data points as available in the overlapping period with instrumental data but minimum 25 pairs of x and y?**

This information will be clarified in the revised manuscript.

**Methods, Page 7, line 30: Is there a reference that any tree-ring proxy responds to an extended fall period (JJASON)? The given references point to common growing seasons from May to August in the northern hemisphere. Why are not all combinations of growing season length tested and the optimum is chosen? In the PAGES data base there are also various different length of growing seasons defined.**

For the first part of the question, see the reply provided earlier in this document. To address the second part, we concede that our objective seasonal PSM calibration methodology is a compromise between comprehensiveness and efficiency in the calculations, performed over more than 2000 proxy records. Nonetheless, we believe that while perhaps non-optimal, the resulting characterization provides realistic results, as suggested by the fact that more accurate reconstructions are obtained when this information is used in forward modeling tree-ring records. This outcome will be further supported in the revised manuscript by presenting verification results performed in proxy space using independent records.

**Methods, Page 8, line 26: "local" should better be "grid box".**

The text will be modified accordingly.

**Updated reanalysis, Page 9, line 5: Can you explain the localization better? Does a cut-off radius of 25000 km mean that each proxy influences basically the entire globe?**

We will add additional clarifying information about the covariance localization configuration used to generate reconstructions and its influence on how proxies are informing the reanalysis.

**Updated reanalysis, Page 9, line 7: Mention that the reference for the skill score is climatology.**

We agree. The text of the revised manuscript will be modified accordingly.

**Figures: some text is too small that it cannot be read in a print version.**

We will revise the final figures to only present the most useful information on the figures.

**Figures: figures should have consistent font types.**

Efforts will be undertaken to use uniform font types.

**References**

PAGES 2k Consortium, Continental-scale temperature variability during the past two millennia, *Nature Geoscience*, 6(5), doi:10.1038/ngeo1797, 2013

PAGES 2k Consortium, A global multiproxy database for temperature reconstructions of the Common Era, *Scientific Data*, 4(170088), 1–33, doi:10.1038/sdata.2017.88, 2017

McKay, N. P. and Emile-Geay, J., Technical note: The Linked Paleo Data framework – a common tongue for paleoclimatology, *Clim. Past*, 12, 1093–1100, doi:10.5194/cp-12-1093-2016, 2016.

Diaz, H. F., Trigo, R., Hughes, M. K., Mann, M. E., Xoplaki, E. and Barriopedro, D., Spatial and temporal characteristics of the climate in medieval times revisited, *Bull. Amer. Meteorol. Soc.*,99(11), 1487–1500, doi:10.1175/BAMS-D-10-05003.1, 2011

D'Arrigo, R., Mashig, E., Frank, D., Wilson, R. and Jacoby, G., Temperature variability over the past millennium inferred from Northwestern Alaska tree rings, *Clim. Dyn.*, 44, 227–236, doi:10.1007/s00382-004-0502-1, 2005

[Figure]

**Fig. 1.** Reconstructed Northern Hemisphere temperatures during the Common Era from LMR (prototype and updated reanalysis discussed in the original paper submission) and other reconstructions.

[Figure]

**Fig. 2.** As in Fig. 1, but with updated LMR results generated by applying some screening of the Breitenmoser et al. (2014) tree-ring chronologies.

---

## Author Comment (AC2) · 18 Jan 2019

**Authors' Responses to Anonymous Reviewer 2**

We thank referee 2 for comments on the manuscript.

**This reviewer believes the methodology, analysis and the final LMR product presented herein are too premature to be acceptable for formal publication, let alone for its stated purpose to serve as the basis for the first publicly released NOAA last millennium reanalysis.**

[Figure]

The data from the first release described in Hakim et al (2016) has been publically available for over two years. The basic method on which Hakim et al (2016) and the present paper are based has been evaluated and tested extensively in the literature (e.g. Bhend et al, 2012; Steiger et al , 2014; Matsikaris et al, 2015; Acevedo et al, 2017; Franke et al, 2017; Okazaki and Yoshimura, 2017; Steiger et al , 2018).

**It is misleading for this study (and its prototype in H16) to call the DA method used ... as an ensemble Kalman filter (EnKF). As in Evensen (1994) and subsequent studies, the primary promise of the EnKF is the use of flow dependent background error covariance represented by the forecasting ensemble. The current socalled "offline" DA method has none of that: the ensemble perturbations are randomly sampled from a past-millennium climate simulation that has no relation to the prior estimate, and the same set of sampled perturbations were used at all analysis times.**

The reviewer appears to have a narrow view of data assimilation limited to operational weather forecasting. In fact, the prior may come from a wide variety of sources, and Monte Carlo sampling of that distribution using ensembles has proved to be a powerful solution method. The "offline" EnKF approach was originally described by Oke et al (2002, 2005, 2007), and Evensen himself described it in his 2003 review of the ensemble Kalman filter (Evensen, 2003). In revision we plan to add a few more references to basic data assimilation theory for readers not familiar with the applicability of the technique in general, and to the aforementioned literature for the offline method in particular.

**This method used in this study is similar to the commonly used 3D-Var method for numerical weather prediction with static background error covariance, and is arguably less advanced than 3D-Var since 3D-Var in NWP used the dynamic**

**model to propagate the previous cycle's analysis as the prior before the analysis. The current so-called "offline" DA method neither cycles the analysis nor the ensemble perturbations, with the stated reason that the forecast model is not good enough to do either.**

We regret the reviewer's interpretation that the motivation for offline DA is because "the forecast model is not good enough," which is not the case. The choice is a result of a cost–benefit analysis: predictive skill of Earth System models on proxy timescales is small, but the cost of ensemble forecasts with these models is high. We plan to make that point clearer in the revised manuscript. For an extension of the LMR method to online DA, and comparison to the offline method, please see Perkins and Hakim (2017).

**If the forecast model is not good enough to cycle the mean analysis or the analysis uncertainties to provide the best estimate of the prior estimate and related prior uncertainties, why would this model(s) be good at all for use as the prior estimate that the LMR reanalysis depends critically on? In this regards, it is premature to state (line 10) that the "LMR employs the ensemble data assimilation to optimally blend the information from the proxies and the climate model data". The current method is more like an objective analysis method.**

Yes, the method is a form of "OI," although we believe that using such jargon is not helpful to the readership of this journal.

**It is not clear whether the authors are aware that the traditional static 3D-Var methods also derive the background covariance from an ensemble of perturbations, as is traditionally called "the NMC method" using the sampled forecast**

**divergence between different lead times from many realizations. The Kalman filter update in this case is equivalent to the variational update using the 3D-Var algorithm, though again the 3DVar in NWP cycles the analysis and forecast during data assimilation, which is the most basic function in combining the model and data.**

Yes, we are aware of the NMC method, which samples forecast differences on the timescale of the DA cycle. In our case that is one year, and the random sampling method we employ assumes that forecast differences on that timescale have converged on the climatological distribution; we lack analyses and forecasts over the Common Era to formally apply the NMC method.

**The validation performed in this study for the prototype and updated LMR "reanalysis" with several existing 20th-century reanalysis is misleading at best. The quality of the LMR reanalysis for the 20th century is the least issue given the availability of the modern much more advanced reanalysis and given the exponentially increased number of proxies or model instrumental observations. The validation currently focuses exclusively on the 20th century says little on the quality and performance of the LMR products, in particular over the early period when the proxy data are scarce. A more appropriate validation can potentially be done in two objective methods: (1) perform the 20th century "reanalysis" through thinning the observation density and maybe also degrading the observation accuracy to those representation of different periods of the past millennium; and/or (2) performing observing system experiments in which a certain number of observations are not assimilated but reserved for independent validation (or all of them in cross validation).**

Part of the method described in this paper involves withholding 25% of the proxies for

independent validation, which we do both before and during the instrumental period. Furthermore, we perform 51 realizations over each experiment, randomly sampling the proxies, so that all proxies participate in validation. In revision, we plan to move those results from the supplementary material into the main body of the paper. Although not described here, we have done experiments consistent with suggestion (1), where we vary the percentage of withheld proxies and there is little sensitivity to the 25% value.

**The use of a 2,5000-km covariance localization is highly questionable for the use of a 100 sets of fixed ensemble perturbations. At midlatitudes, this is amount to the observation impacts across the entire global latitude belt. The use of a fixed set of 100 sample perturbations also means a high rank deficiency over such a large area with this large localization distance.**

This reasoning is consistent with covariance lengthscales on weather timescales. Covariance lengthscales on annual timescales are much longer, and the effective dimension of the covariance matrix is comparatively smaller.

**The current final LMR reanalysis derives from the mean of 51 such 100-member analyses, should it be the same if the 5100 samples of perturbations are used simultaneously in the Kalman filter update given the Kalman filter used is largely a linear operation?**

The fact that we get better results by averaging over Monte Carlo realizations (multiple analyses) as compared to larger ensembles is not completely understood. We believe that this is an artifact of poorly estimated analysis errors for a subset of the proxies, but fully exploring this issue is beyond the scope of the present paper. In revision we will highlight the issue and the hypothesis we have for it.

**How much is the result sensitive to the choice of this arbitrary number of sample perturbations? It is also worth noting the the NMC method used for 3D-Var uses singular value decomposition to make it full rank. Such a approach is different from (and likely more advantageous over) the current Kalman filter update using purely non-envolving static ensemble covariances.**

We have found little sensitivity to the ensemble size, provided it is at least 100 members and that covariance localization is used (effectively increases the rank of the covariance matrix). We find larger improvements by randomly subsampling the proxies through many Monte Carlo realizations.

**It is unclear what is the purpose of such as hastily done LMR reanalysis products with such ad-hoc DA approaches and the not-good-enough forecast models? The so derived climate trend is almost certainly depending too much on the climate models used as a prior and ensemble sampled perturbations (and maybe the assumed climate forcings used in these models), as well as the density of observations over different periods. It could do more harm if such a premature reanalysis product is used or misused and if it were publicly released through NOAA, unfortunately. A more careful vetting of the products, and a more concerned effort in refined DA methodology are warranted before NOAA sanctioned such a product as reanalysis, in this reviewer's opinion.**

If you wish to see more sensitivity analysis with respect to these issues, please carefully read Hakim et al (2016), where we not only considered the performance statistics of analyses using different priors and calibrations of proxy forward models, but also examples of the differences that result in the spatial fields for an individual year.

**References**

Hakim, G. J., Emile-Geay, J., Steig, E. J., Noone, D., Anderson, D. M., Tardif, R., Steiger, N. J. and Perkins, W. A., (2016) The Last Millennium Climate Reanalysis Project: Framework and First Results, *JGR: Atmospheres*, 121, 6745–6764, doi:10.1002/2016JD024751, 2016

Bhend, J., Franke, J., Folini, D., Wild, M. and Brönnimann, S., An ensemble-based approach to climate reconstruction, *Climate of the Past*, 8, 963–976, 2012.

Steiger, N. J., Hakim, G. J., Steig, E. J., Battisti, D. S. and Roe, G. H., Assimilation of time-averaged pseudoproxies for climate reconstruction, *J. of Climate*, 27, 426–441, doi: 10.1175/JCLI-D-12-00693.1, 2014

Matsikaris, A., Widmann, M. and Jungclaus, J., On-line and off-line data assimilation in palaeo-climatology: a case study, *Climate of the Past*, 11, 81–93, doi:10.5194/cp-11-81-2015, 2015

Acevedo, W., Fallah, B., Reich, S. and Cubasch, U., Assimilation of pseudo-tree-ring-width observations into an atmospheric general circulation model, *Climate of the Past*, 13, 545–557, doi:10.5194/cp-13-545-2017, 2017

Franke, J., Brönnimann, S., Bhend, J. and Brugnara, Y., A monthly global paleo-reanalysis of the atmosphere from 1600 to 2005 for studying past climatic variations, *Sci. Data*, 4, doi: 10.1038/sdata.2017.76, 2017

Okazaki, A. and Yoshimura, K., Development and evaluation of a system of proxy data assimilation for paleoclimate reconstruction, *Climate of the Past*, 13, 379–393,doi:10.5194/cp-13-379-2017, 2017

Steiger, N. J., Smerdon, J. E., Cook, E. R. and Cook, B. I., A reconstruction of global hydroclimate and dynamical variables over the Common Era, *Sci. Data*, 5, doi: 10.1086/sdata.2018.86, 2018

Oke, P. R., Allen, J. S., Miller, R. N., Egbert, G. D. and Kosro, P. M., Assimilation of surface velocity data into a primitive equation coastal ocean model, *JGR:Oceans*, 107, doi:10.1029/2000JC000511, 2002

Oke, P.R., Schiller, A. Griffin, D. A. and Brassington, G. B., Ensemble data assimilation for an eddy-resolving ocean model of the Australian region, *Quart. J. of the Royal Meteor Soc.*, 131, 3301–3311, 2005

Oke, P. R., Sakov, P. and Corney, S. P., Impacts of localisation in the EnKF and EnOI: experiments with a small model, *Ocean Dyn.*, 57, 32–45, 2007

Evensen, G., The ensemble Kalman filter: Theoretical formulation and practical implementation, *Ocean Dyn.*, 53, 343–367, 2003

Perkins, W. A. and Hakim, G. J., Reconstructing paleoclimate fields using online data assimilation with a linear inverse model, *Climate of the Past*, 13, 421–436, 2017

---

## Author Comment (AC3) · 18 Jan 2019

**Authors' Responses to Anonymous Reviewer 3**

We thank the referee for helpful comments on the manuscript.

**I have two main concerns about the paper, which together require that the manuscript undergo major revisions before it is acceptable for publication. The first is the character of the derived reconstruction and the unsatisfactory verification of the product using only observational data. The second is the use of**

[Figure]

**multiple ad hoc methodological choices, none of which are reasonably justified or widely tested.**

The first concern, or two concerns really, were also raised by other referees. We have undertaken a revision of the manuscript that will include a more complete discussion on the character of the temperature reconstruction, compared and contrasted with other reconstructions (see response to next comment). We also plan to expand the verification performed in proxy space and move the discussion from the supplementary material into the main body of the paper.

The referee's concern on the justification of our selection of the data assimilation parameters will also be addressed. A brief mention on how those choices were made will be included. We believe however that a lengthy discussion on these is not warranted in the present paper, as these choices have been discussed in prior publications. However we agree that some clarification on how these choices were made should be included to convince the reader that careful consideration was involved.

**I am struck by the comparison in Figure 2a and the little attention the authors give to the differences between the previous LMR product and the newer version (not to mention the complete lack of comparison between either of these results and other temperature reconstructions).**

We agree that our results should be better framed with respect to other reconstructions, as also echoed by other reviews. We have compared LMR results of Northern Hemisphere (NH) temperature and other reconstructions included in the IPCC AR4 and AR5 reports; see Figure 1 in this document as an example. A low-pass filter has been applied on all results to focus on the lower frequencies. Such comparisons will be the basis of a discussion contrasting our results with other efforts in temperature

reconstructions in a revised Section 3.

**The GMT from the newer product looks almost like white noise and has lost not only the multi-decadal to centennial variability in the first product, it is also likely at odds with the now large collection of global and hemispheric temperature reconstructions spanning the last millennium or more. The authors not only need to spend more time discussing this issue, they also need to compare their results to the collection of large-scale temperature and hydroclimate reconstructions currently available.**

The perspective gained from the comparison shown in Figure 1 suggests that indeed there are differences between the updated and prototype LMR reconstructions, as well as with other reconstructions, but also provides some evidence that a claim of "white noise" is an overstatement. The differences between the other reconstructions themselves serve as a testament of the uncertainties characterizing climate reconstructions. In that context, we should to point out that most other reconstructions are found within the bounds of the LMR ensemble most of the time, indicating some level of agreement between the different products, when framed within the context of the uncertainty defined by our reconstructions.

We do acknowledge that a loss of low frequency variability characterizes our updated reconstruction. As pointed out by the referee about global-mean temperature, also reflected in the NH-mean temperature results in Fig. 1, the perceived loss of variability is mainly the result of differences limited to three distinct periods. These are: a colder medieval period, most notably during the 875–1050CE period, and warmer temperatures during the 1600–1700CE and 1810–1920CE periods during the Little Ice Age (LIA). We further note that the updated LMR is among the warm outliers during these cold periods compared to the prototype and other reconstructions. An analysis of results from a large number of reconstruction experiments has allowed us to conclude

that colder temperatures during the medieval period, compared to the prototype LMR, are related to the change from an earlier proxy dataset PAGES 2k Consortium (2013) to a recently published collection PAGES 2k Consortium (2017). The global temperature composites presented in PAGES 2k Consortium (2017) shows that a distinctly warmer medieval period isn't a prominent feature of the new collection, and does not result from other updates to our data assimilation system. As this dataset reflects the community's most recent and rigorous identification of proxy records suitable for temperature reconstructions, we believe that the lack of a "classic" medieval warm period in our updated reconstructions of global mean surface temperature (GMST) and NH-mean temperature should not necessarily be considered an outstanding shortcoming of our updated reanalysis.

On the other end, beneficial further refinements to our DA system have been identified as a result of issues raised by the referees. A notable sensitivity of reconstructed LIA NH-mean temperatures to the set of assimilated tree ring width chronologies has been identified, underlining possible weaknesses in observation error variance estimation (i.e. the $R_k$ terms in equations 4 in the manuscript) for these records. Additional reconstruction experiments have shown that colder LIA temperatures are obtained when some of the large number of tree ring width records from the Breitenmoser et al (2014) collection are not assimilated (see Fig. 2). A more important low frequency variability is recovered, in better agreement with other reconstructions.

As suggested by the referee, a revised Section 3 will include a more complete discussion of these updated results, including comparisons with other available reconstructions.

**...it is essential for them to do more to verify their results beyond the comparisons they make to observational data. While the latter is important and useful, it is not enough. Incidentally, the authors do perform validation exercises on a withheld period of observational data and using withheld proxy series, but that**

**work is buried in the supplemental and not adequately discussed in this context. More should be made of those efforts, which strengthen the authors results with truly out-of-sample validation experiments.**

We agree. We will include an expanded discussion of proxy verification results in the main body of the text.

**I do not think the use of CE is the same as it is traditionally used in the paleo literature, given that the latter approach requires a true cross-validation period. The authors should clarify this point.**

The formulation has been used in numerous other published work. We have found that CE is a valuable complementary metric to correlation, due to its sensitivity to mean errors and representation of variance in the evaluated fields. We will contrast our implementation of the metric cmoapred to its traditional use in the literature.

**I think it is further important for the authors to derive validation experiments for the sparsely sampled periods early in the proxy network (e.g. deriving reconstructions using only subsets of the proxies that extend back to specific time intervals). This would go a long way toward helping to better understand the loss of proxy information back in time. This is partially addressed by the variance exercise the authors perform, but more can be done. Recons for temporal subsets of the proxy network would in fact be more useful than the MC sampling of the proxy network that the authors perform, given that it would be systematic and inform a direct question about the influence of the declining proxy network.**

This is a very good suggestion. Experiments similar to what is suggested here have

been conducted in support of another publication, currently under review. This issue can also be investigated through an assessment of reanalysis performance performed in proxy space, with a focus on different time intervals. Proxy verification results will be included in the revised manuscript. This characterization should also be framed in the context of the uncertainties characterizing reconstructions. This capability is enabled in LMR through the availability of ensemble-member information. This complementary perspective will also be incorporated in the revised version of the paper.

**Here is a ... list of choices the authors have adopted that are not accompanied by any justifications or sensitivity discussion: 1-Use of 100-member ensemble; 2-Use of the CCSM4 last millennium simulation as the prior; 3-Use of 51 MC realizations; 4-Use of a proxy sampling scheme based on 75% of the proxy records; 5-Degradation of the model resolution to a ~5x5 grid. All of these choices undoubtedly influence the derived LMR product. Some of them can be justified based on discussions in the literature. Some of them require empirical demonstrations. All of them come across as ad hoc. I would also venture to guess that the LMR results are more dependent on a couple of these choices than the other dependencies that the authors more systematically test. It is therefore essential that the authors do a better job of justifying these choices and convincing the reader that they are either reasonable choices or chosen based on some methodological/logistical rationale.**

Some of these choices have been discussed in prior publications. See Hakim et al (2016) and references therein for example. However we agree that text reminding the reader of how parameters were chosen should be included to convince that careful consideration was involved. To further clarify the context, we should point out that some parameter values used here were chosen to to maintain consistency with the configuration used in Hakim et al (2016), in order emphasize the impact of updates

specifically addressed in this work.

Some of the parameters were originally chosen based on practical considerations. For example, the ratio of assimilated versus withheld proxies and the number of Monte-Carlo realizations were chosen so that the random sample of proxies set aside for independent validation is representative of the overall dataset. Such considerations will be more clearly described in our revision of the manuscript.

**I should specifically mention the use of the CCSM4 as the prior. The authors say nothing about how their results might depend on the model prior and whether they have tested alternative last-millennium simulations in their analysis. This is an obvious question and the authors need to address it.**

This topic is addressed in Hakim et al (2016), with results from reconstructions using a wide range of calibration and prior data (see Fig. 12 of that paper). Work on this topic has since been expanded and explored in more detail. These results are, however, expected to be the subject of a separate publication.

**Page 2, lines 5-6: What does "synthesizing information" mean? This is vague and I am not even sure the statement is true. There are lots of central challenges of paleoclimate science, and it is arguable that what the authors are alluding to is one of them. This strikes me as an unsupported justification for what the authors subsequently say they are attempting to do.**

We will modify the text to more precisely reflect the goals of the work presented in our paper. In this part of the text, our goal is to emphasize that data assimilation is a powerful framework in which information from proxies and numerical model simulations is combined for the production of, hopefully, robust climate reconstructions.

**Page 2, lines 30-32: This is a much more mundane objective than the sense given in the abstract. Are the authors attempting to release a shiny new LMR product or should this be seen as an iterative verification step toward some improved effort down the line?**

We apologize if the text does not clearly convey the goals pursued here. As with any complex problem to be solved, as is the case here, improvements are generally incremental and modest. The main goal of efforts reported in this paper has been to improve our system over the LMR prototype, and to deliver an incrementally improved product, while being fully cognizant of the fact that room for improvement remains. In the revised manuscript, we will attempt to convey these goals in a more precise manner.

**Page 8, line 9: The use of precipitation is not justified and concerning. First, precipitation is almost never the variable associated with moisture sensitivity in trees - some measure of soil moisture is. It is therefore not clear why the authors used precipitation and how it influences their results. Why not use a more conventional variable like PDSI? Secondly, how do the characteristics of precipitation influence the results? Does it matter that precip is likely not Gaussian and that it has limited spatial and temporal covariance structure? Is the use of precip perhaps adding to the loss of low-frequency variance in this new LMR product? My guess is that this specific choice has a large impact on the derived reconstruction and the use of precip is not justified in any way.**

The referee is raising fair and important points here. We acknowledge that soil moisture is the preferred response variable for the modeling of tree-ring widths. However, given our approach, which relies on the availability of calibrated forward models, the absence, to our knowledge, of a reliable century-long soil moisture dataset is an important limiting factor. An alternative, as pointed out by the referee, would be PDSI (or other drought indices such as SPEI) instead of soil moisture. This option, is enabled within our framework with the use of the Dai et al (2004) dataset for forward model calibration. We intend to generate test reconstructions using this configuration and compare with results using precipitation. We believe these comparisons will allow the impact of the issues raised by the referee to come into greater focus.

To provide additional context on our efforts reported in the submitted manuscript, more particularly with the development of bilinear PSMs, our intent is to replicate VSlite's (Tolwinski-Ward et al, 2011) general approach of combining the influences of temperature and moisture in modeling tree-ring width variability, albeit in a simpler fashion, while avoiding the issues associated with VSlite's formulation involving thresholds (see Dee et al, 2016). We will perform a careful quantitative comparison with the more conventional approach using linear models calibrated on PDSI and discuss the results in our revised manuscript. Particular attention will be placed on the characterization of low frequency variability to assess whether the updated LMR's characteristic loss of variability is rooted in the use of precipitation to forward model tree-ring width proxies.

**Figures: In general, there is a lot of small text in the figures that is hard to read and also rather confusing and messy.**

We will simplify the design of the revised figures.

**Figures: The many colorbars are also unnecessary in many plots when one would do.**

We will streamline the design of the revised figures.

**References**

Hakim, G. J., Emile-Geay, J., Steig, E. J., Noone, D., Anderson, D. M., Tardif, R., Steiger, N. J. and Perkins, W. A., The Last Millennium Climate Reanalysis Project: Framework and First Results, *JGR: Atmospheres*, 121, 6745–6764, doi:10.1002/2016JD024751, 2016

PAGES 2k Consortium, Continental-scale temperature variability during the past two millennia, *Nature Geoscience*, 6(5), doi:10.1038/ngeo1797, 2013

PAGES 2k Consortium, A global multiproxy database for temperature reconstructions of the Common Era, *Scientific Data*, 4(170088), 1–33, doi:10.1038/sdata.2017.88, 2017

Tolwinski-Ward, S. E., Evans, M. N., Hughes, M. K. and Anchukaitis, K. J., An efficient forward model of the climate controls on interannnual variation in tree-ring width, *Climate Dyn.*, 36, 2419–2439, doi: 10.1007/s00382-011-1062-9, 2011

Dee, S., Steiger, N. J., Emile-Geay, J. and Hakim, G. J., The utility of proxy system modeling in estimating climate states over the Common Era, *J. Adv. Model. Earth Syst.*, 8, 1164–1179, doi: 10.1002/2016MS000677, 2016

Breitenmoser, P., Brönnimann, S. and Frank, D, Forward modelling of tree-ring width and comparison with a global network of tree-ring chronologies, *Climate of the Past*, 10, 437–449, doi:10.5194/cp-10-437-2014, 2014

Dai, A., Trenberth, K. E. and Qian, T., A global data set of Palmer Drought Severity Index for 1870-2002: Relationship with soil moisture and effects of surface warming, *J. Hydrometeorology*, 5, 1117–130, doi:10.1175/JHM-386.1, 2004

[Figure]

[Figure]

**Fig. 1.** Reconstructed Northern Hemisphere temperatures during the Common Era from LMR (prototype and updated reanalysis discussed in the original paper submission) and other reconstructions.

[Figure]

**Fig. 2.** As in Fig. 1, but with updated LMR results generated by applying some screening of the Breitenmoser et al. (2014) tree-ring chronologies.

---

## Author Response (AR1)

Last Millennium Reanalysis with an expanded proxy database and seasonal proxy modeling

Tardif et al.

Authors Responses to Reviewers

In the following, comments from the referees are shown in bold, with our responses shown in plain text. A marked-up version of the revised manuscript, showing the changes from the originally submitted version, is also included. Page and line numbers refer to the revised manuscript.

**Anonymous Reviewer 1**

We thank the referee for thorough and insightful comments on the manuscript. The comments have challenged us to take a more comprehensive look at the numerous reconstruction experiments we performed, and as a result, we have gained a more complete perspective on the results.

**The paleoclimatic discussion needs to go more into depth and the authors need to be more critical about their own results. The authors compare their new data set to the previous version (Hakim et al. 2016) and conclude this new version would be an overall improvement. This conclusion is based on validation statistics in the 20th century. However, the new data set lost all multi-decadal to centennial variability in the global mean temperature and does not show a warmer medieval period nor a cooler "little ice age" anymore. The authors do not discuss this issue at all. The paper suggests that this new reanalysis version would present the more likely global mean temperature evolution of the past 2000 years although it is in contrast to what most other reconstructions and paleodata records suggest.**

We agree with the referee's suggestion that results should be framed in the context of other reconstructions, and a discussion focused on the long-term perspective (not limited to the 20th century) of the updated reanalysis be included in the manuscript. To address this issue, a revised section 3 includes a more complete discussion of our reconstruction results, including a comparison of LMR reconstructions of Northern Hemisphere (NH) temperatures with other NH reconstructions found in the IPCC AR4 and AR5 reports (see bottom panel of Figure 2, section 3). A greater perspective on the long-term (i.e. across the Common Era) evolution of temperature in our reconstructions is gained. The main takeaways from this comparison can be found in the fourth paragraph of the revised section 3, page 10 line 23 to page 11 line 7.

We do acknowledge the updated reanalysis in our originally submitted manuscript was characterized by a significant loss of variability compared to our prototype reanalysis. We have revisited some of the choices made in the configuration of our system and conducted additional experiments to identify the source of this loss of variability. We have determined that it is preferable to eliminate the large number of unscreened tree-ring records from the Breitenmoser et al (2014) collection, as included in Anderson et al (2019) from our reanalysis. This is despite an increase in skill in temperature and hydroclimate reconstructions when this dataset is included, as determined from observational data. We believe this underlines issues with our estimates

of observation error variance (i.e. the $R_k$ terms in equations 4 in the manuscript) specifically for these records. We have found that the inclusion of these tree-ring chronologies is largely resposible for the warm bias which characterizes reconstructed temperatures in our previous updated reanalysis during the 1600–1700CE and 1810–1920CE periods of the Little Ice Age (LIA). Therefore, we now present a new updated reanalysis using a revised configuration of the assimilated proxies, limited to the proxies from the PAGES 2k Consortium (2017) collection. With this configuration, we find a greater level of temperature variability is recovered, as well as a much improved agreement between the LMR Northern Hemisphere temperature to the other reconstructions during the LIA. We note however that the absence of a notable warm medieval period continues to characterize our reanalysis. An analysis of results from a large number of reconstruction experiments has allowed us to conclude that colder temperatures during the medieval period, compared to the prototype LMR, are related to the change from the proxy dataset of PAGES 2k Consortium (2013) to the more recent PAGES 2k Consortium (2017) collection. The global temperature composites presented in PAGES 2k Consortium (2017) shows that a distinctly warmer medieval period isn't a prominent feature of the new collection, and is not the result of other updates to our data assimilation system. As this dataset reflects the community's most recent and rigorous identification of proxy records suitable for temperature reconstructions, we believe that the lack of a "classic" medieval warm period in our updated reconstructions of global mean surface temperature and NH-mean temperature should not necessarily be considered an outstanding shortcoming of our updated reanalysis. Discussions on the issues outlined above are included in section 3, page 10 lines 3 to 12, and in section 4.3, entirely revised due to the re-focus of the updated reanalysis.

**The loss of low-frequency variability is most likely a consequence of the proxy data sets used, because this is the major change in the new version. Many of the tree-ring chronologies in Breitenmoser et al. 2014 are not climate sensitive at all or moisture sensitive. As precipitation does not show any low-frequency variability in contrast to temperature, it is a logical consequence that using covariance information from moisture sensitive trees to correct temperature data leads to a loss of low frequency variability.**

The referee raises good points, confirmed following a careful review of our results, including those from additional test reconstructions (see response to previous comment). We note however that the data assimilation (DA) framework is different than other reconstruction methods. Key elements are the forward models (the proxy system models, or PSMs) used to estimate proxy observations from a model prior, and the observation error variance assigned to the proxies, i.e. the $R_k$ terms in equations 4 in the manuscript. Within a DA framework, there are no fundamental reasons why the inclusion of records with sensitivities other than temperature would lead to a deterioration to reconstructions of temperature, provided that PSMs properly account for the proxy sensitivities and the associated observation error variances representative are properly specified. However, it is acknowledged that these conditions may not easily achieved. Initial LMR results (i.e the prototype) suggested that our approach has the ability to delineate proxy records with weaker sensitivity to climate by assigning relatively larger observation error variances ($R_k$), resulting in such records only weakly influencing the reanalysis results even though they are assimilated. Also, the main motivation for developing the bilinear approach to model tree-ring width (TRW) data as been to gain an ability to seamlessly handle the more

complex sensitivities to temperature and/or moisture of these chronologies. With this approach, in principle, TRW records can be assimilated without having to make a binary decision whether each record is dominantly sensitive to temperature or moisture, or having to screen records out a priori. However, we acknowledge that relying on simple regression-based PSMs opens up the process to the influence of spurious correlations between noisy data. With a large number of proxies considered, some records will invariably be characterized by somewhat overestimated confidence, i.e. too-small error variance, and therefore overly weighted in the update. Closer examination of additional test reconstructions strongly suggest that these issues are responsible for the loss of variability observed when the entire set of Breitenmoser TRW records are assimilated. We conclude that the possible inclusion of proxies from this dataset, including an improved characterization of observation errors, requires further attention. Hence, the revised manuscript presents, throughout the paper, an updated LMR using a revised configuration, where assimilated proxies are limited to the PAGES 2k Consortium (2017) collection.

**A second reason may be the use of proxy data with dating uncertainties, such as ice cores, in an annual reconstruction. These proxies probably do not have age errors in the 20th century validation period but become just noise if they have an age offset of one or a few years further back in the past. The authors just conclude that using moisture sensitive data leads to improved reconstruction skill, although this is only true in the 20th century validation period but not in the pre-instrumental period, most user of this data set will be interested in.**

This is an interesting suggestion. We have tested this hypothesis by performing an additional experiment in which all ice cores records were withheld from assimilation. The results do not support the hypothesis put forward by the referee however. GMST and NH-mean temperatures exhibit similar multi-decadal variability compared to reconstructions which include ice core information. The main difference consists of a modified long-term trend, showing a flatter temporal evolution over most of the Common Era prior to the 20th century warming, worsening the agreement with the other reconstructions. The primary role of ice core proxy data seems to rather anchor the millennial-scale cooling characterizing the pre-industrial era. However, we reserve a discussion on this specific topic for a future publication which is expected to focus on the role of various proxies in the reanalysis.

**In the current version, the global mean temperature evolution is the reappearance of the famous "hockey stick" in climate science. After all the discussion, the hockey stick was rising 20 years ago, I would not publish this as a state-of-the-art temperature reconstruction, especially not without a discussion and not if it is an artefact of unscreened input data.**

See responses to the comments above. In the light of results outlined above, the revised manuscript presents an alternate updated LMR, showing a greater level of variability, including at lower frequencies, in better agreement with reconstructions from other authors as shown in panel (c) in a revised Figure 2 and discussed in the revised section 3, page 10 lines 3 to 12 and page 10 line 23 t page 11, line 7.

**I see two options, the first would involve minor revisions and the second major revisions: 1. It must be stated prominently (already in the abstract) that this reanalysis should not be used or considered to have the correct multi-decadal and centennial variability and that the global mean time series over the last 2000 year potentially has serious issues. The discussion needs to include all problems of data set, too and ideas how to overcome them in the future. In general, the paper should be put more into a context of methodological improvements to achieve better products in the future instead of claiming this would be nearly the prefect reanalysis for the past 2000 years. 2. A proper screening of the data needs to be introduced that prohibits the assimilation of non-climatic information, which has just spurious correction with observations in the short window of overlapping instrumental and proxy data (a minimum of 25 data points has been used in this study, page 5, line 2). These records will have little weight in the assimilation procedure due to large residual variance, but hundreds of little errors probably produce significant noise. Probably, precipitation limited proxies and proxies with age errors have to be removed or treated specifically, too. These are just some ideas and it will need many improvements and new experiments to find and solve the problem.**

We appreciate these comments and suggestions. As outlined in our response to previous comments, we have chosen to follow a path inspired by the second suggestion. A complete review of results has been undertaken and new reconstructions experiments were carried out. The newly gained perspective has led us to consider an alternate reanalysis configuration which addresses the issues identified by the referee. Suggestions by the referee were carefully considered and integrated in the design of our latest experiments.

However, we wish to underline our belief that an approach that seeks to simply remove the information from precipitation limited proxies, as suggested by the referee, is not the preferred framework in which to seek improvements in reconstructions. Rather, improved forward models (PSMs), describing more accurately the relationships between climate variables and proxies, in addition to improved characterization of observation errors, are the key aspects where improvements can be achieved. We believe that this is reflected in part in our results showing improved performance with the reanalysis using the seasonal bivariate temperature/precipitation PSMs on the screened PAGES2k TRW records.

**A difficulty in the review process is that the input data has not been published, yet. Hence, it is not possible to properly judge the input data base. However, it appears to be basically the Breitenmoser et al. 2014 data set with a few coral and ice core records added. Why do you not simply refer to this first publication and give citations for the additional records or wait until the Anderson et al. paper is published?**

The manuscript is now available online at:
https://datascience.codata.org/article/10.5334/dsj-2019-002/

**In general, the decrease of skill further back in time is not discussed sufficiently.**

This is a great comment. We address this issue using our framework enabling an assessment of reanalysis performance in proxy space. The scope of the verification of reconstruction

results presented in the revised paper has been expanded by including results from verification performed in proxy space on independent (withheld) proxies covering both calibration and pre-calibration periods. Results that were originally included in the supplementary material have been revised and moved to the main body of the paper. These results are found in section 4.1, Table 4, to complement the evaluation of the various PSM configurations considered, confirming with increased confidence the choices made in configuring the updated reanalysis system. Results from proxy–space verification are also included in section 4.3, Table 6, to gain an additional perspective on the accuracy of reconstructions performed with the addition of proxy records from the Anderson et al (2019) collection. The discussions on the proxy verification results are found on page 14, lines 1 to 7, and on page 16 lines 3 to 16.

**It should also be discussed why forcings are not important and what the consequences of unforced simulations ensembles are for the final product, especially further back in the past when the proxy network is sparse.**

We are not sure how the referee has come to believe that forcings are not important. In fact the model simulations from which we draw prior information do include forcings, such as pre-industrial greenhouse gas and aerosol variability, including the effects of volcanic eruptions. This point is mentioned in the manuscript. However, to bring the context into greater focus, we have found that an important characteristic of prior information within our DA framework is the amount of variance characterizing the simulations. We have found that the greater variance generally characterizing the "Last Millennium"-type simulations (which include the forcings listed above) provide for more accurate reconstructions, compared to using simulations performed without the influence of external forcings (as in the "pre-industrial" or piControl CMIP5 protocol). We also have generally refrained from using simulations which cover the 20th century warming to dispel the notion that we are "cooking the books" when reconstructing temperature trends. In our framework, temporal information (trends) come entirely from weighted information from the proxies.

**I suggest to evaluate the spatial skill of the reanalysis in the 20th century but with the spatial proxy network at multiple time slices, e.g. 0 AD, 500 AD, 1000 AD, 1500 AD.**

This is a great suggestion. We have generated proxy verification results over distinct periods of the Common Era: 1–499, 500–999, 1000–1499, 1500–1879, 1880-2000 (calibration period) in Table 2 and discuss the results from the perspective of reanalysis accuracy across time in a revised section 3. Even though the verification is not performed on an identical set of proxies we also compare the verification statistics from the prototype reanalysis, to further highlight the enhanced performance of the updated LMR. Additions to the text are found on Page 12, lines 16 to 22.

**Additionally, not using forced simulation offers the potential to use them in the validation procedure. It could be checked if temporal and spatial patterns of known past events or periods are well represented in the reanalysis, e.g. spatial moisture distribution after eruptions (Iles and Hegerl, 2015).**

This is also a good suggestion, however we believe that such efforts are outside the scope of the current work. The suggestion will be considered in future efforts.

**Finally, it would be interesting to see a map of the regression residuals to get an idea how many paleodata records have significant influence in the assimilation procedure and which are basically ignored because they have no climate information.**

We agree, however a concise presentation of this is challenging, due to the varied nature of the proxy data. We believe this would be addressed in a more informative way by a formal proxy impact study, which is intended to be the subject of another paper.

**Additionally, I would like to know how many records in the PAGES2Kv2 data base have expert information on seasonality? I would be interested to read how well the expert-based seasonality in the PAGES data base agrees with the objective assessment in this study. Probably, the experts did a similar search for highest correlation, maybe just including more possible combinations of growing season months.**

For the 2017 publication, the PAGES 2k consortium requested that each data certifier assess the seasonality of the temperature response and report its basis. In the LiPD format (McKay and EmileGeay, 2016), this information is encoded in the `climateInterpretation_seasonality` and `climateInterpretation_basis` metadata properties. All records in the database include seasonality information, either as letters (JJA) or numbers ([6 7 8]) indexing calendar months. When the basis is reported, it is either from "first principles" (e.g. trees are known to grow in local summer, which in most cases is synonymous with June July August), or from a search for the highest correlation. When the basis is not reported, the reader is referred to the publication documenting each record, which in most cases uses a mix of first principles and search for highest correlation, similar to the approach used in this paper. A key difference between the expert assessment of seasonality and the one done in our paper lies in the choice of target datasets. Studies focusing on individual series tend to be more careful about selecting an instrumental dataset appropriate for calibration (e.g. local GHCN station, rather than GISTEMP grid box average). These choices of target datasets are likely contributing to differences in the seasonal window determined via this process.

The choice of calibration datasets in LMR is driven by the need to uniformly process a large number of globally distributed records, hence the more general, likely not optimal, selection as is possible when one has to consider a single or few records.

We also wish to point out that the information requested by the referee on the differences between the expert-based seasonality in the PAGES data base and objective assessment in this study is already provided in the supplemental information accompanying the main manuscript (Figure S1).

**I was surprised to read that the authors use an extended fall period (JJASON)? Is there any reference for trees which are limited by climatic conditions in these autumn months.**

Consideration of this period has initially been motivated following D'Arrigo et al (2005), along with the fact that some seasonal responses found in the expert-derived PAGES2k metadata extend to fall months, as suggested by the data shown in Figure S1a in the supplemental material accompanying our submission. It is interesting to note however that the objectively-derived seasonal responses determined by the approach described in our paper leads to less emphasis on those fall months compared to the PAGES2k expert data.

**It would be favorable to store the data at a world data center and not at a personal homepage.**

We completely agree with the referee on this point. The LMR team has engaged interactions with the project's sponsor (NOAA) to identify a suitable storage location and access point, but these have yet to be identified, hence the reference to a personal homepage at this point.

Technical corrections

**Abstract: skill score increase in percent is misleading. It is easy to have a large relative increase if scores were very low in the comparison data set, e.g. in Z500 where CE improves from very negative to less negative the increase in percent is large but the skill is still negative!**

The point made by the referee is well taken. However, our emphasis has been about quantifying differences with respect to our main benchmark, the LMR prototype, to highlight improvements. Therefore we remain convinced that the formulation used is appropriate. The fact that some skill scores remain characterized by negative values is not hidden and becomes quite clear in the core of the text.

**Abstract: be more precise what is meant with "ensemble characteristics".**

We have eliminated the sentence containing this expression to streamline the abstract. We were referring to the preferred outcome of maintaining good correspondence between ensemble-mean errors and ensemble variance (i.e. uncertainty), a concept referred to as ensemble calibration in the data assimilation literature. We now more clearly define this in the last paragraph of Section 3 in the revised paper, page 12 lines 7 to 16.

**Introduction, Line 17: apart from paleoclimate with annual observations, the forecast model is a third important component (this is even written in the Methods section).**

We agree that this statement could be interpreted to mean that the prior data has less importance in the DA context. A revised statement, more accurately conveying the importance of the various components, is included at the beginning of the second paragraph of the Introduction, Page 2 lines 12 to 15.

**Methods, Page 7, line 6ff: I do not understand why the calibration is done with a different gridded data set than the validation. Both data sets a based on largely overlapping instrumental observations and therefore clearly not independent.**

The use of GISTEMP as the calibration dataset has largely been motivated by the fact that a larger number of proxy records could be calibrated (larger number of records with sufficient overlap with valid calibration data) compared to other datasets. Therefore, a larger number of proxies may participate to the reanalysis. We in fact perform validation using all datasets at our disposition, including the calibration dataset. Skill metrics show small differences among the various results, but remain in general agreement. Here we have chosen the Berkeley Earth dataset because it provides a greater spatial coverage, comparable to the calibration dataset, therefore providing a larger sample of spatial verification results.

**Methods, Page 4 line 14: How many ensemble members?**

The information is already provided in the manuscript, in Section 2.3, where details of the configuration are listed.

**Methods, Page 4 line 14 it is written that Hakim et al. 2016 worked "with the same randomly drawn ensemble members used for every year in the reconstruction", whereas on page page 5, line 28 it is written for this study: "each using a different randomly chosen 100member ensemble". Are both studies consistent and is each year build on different randomly chosen ensemble members?**

Both statements are consistent, in that the first statement describes how the ensemble members for a given reanalysis realization are used to populate prior information in the time domain, whereas the second statement points out that different sets of states are used to populate the prior ensemble for different Monte-Carlo realizations of reconstructions. A revised statement clarifies this in the last paragraph of section 2.1, page 5, line 10, with the addition of "for every year in the reconstruction of a given reanalysis realization".

**Page 5, line 1: "Only records for which a PSM can be established are shown ...". What do you mean by "shown"? There is no reference to a figure. Do you want to say that only records meeting these criteria are assimilated?**

We wish to point out that a reference to Figure 1 is included in the previous sentence. The text of the last paragraph of section 2.2 has been modified to ensure the reader is not confused about what the reference is. See page 5, lines 31-32.

**Methods, Page 5, line 8: Breitenmoser et al. 2014 is not screened for any climate sensitivity.**

This text has been moved to section 4.3, and a clearer statement about the absence of screening for climate sensitivity has been added, page 14, line 30.

**Methods, Page 6, Proxy modeling: It should be repeated here over which period the regression coefficients are calculated. As many data points as available in the overlapping period with instrumental data but minimum 25 pairs of x and y?**

This information is now briefly clarified in the revised manuscript. We have added the following statement: "a threshold of at least 25 overlapping data is imposed", page 7 lines 4-5.

**Methods, Page 7, line 30: Is there a reference that any tree-ring proxy responds to an extended fall period (JJASON)? The given references point to common growing seasons from May to August in the northern hemisphere. Why are not all combinations of growing season length tested and the optimum is chosen? In the PAGES data base there are also various different length of growing seasons defined.**

For the first part of the question, see the reply provided earlier in this document. About the second part, we concede that our objective seasonal PSM calibration methodology is a compromise between comprehensiveness and efficiency in the calculations, performed over more than 2000 proxy records. Nonetheless, we believe that while perhaps non-optimal, the resulting characterization provides realistic results, as suggested by the fact that more accurate reconstructions are obtained when this information is used in forward modeling tree-ring records. This outcome is now further supported in the revised manuscript with verification results performed in proxy space using independent records, with results presented in the new Table 4 and discussed in Page 14 lines 1 to 7.

**Methods, Page 8, line 26: "local" should better be "grid box".**

The word "local" has been changed to "grid cell" in the revised text, page 9, line 18.

**Updated reanalysis, Page 9, line 5: Can you explain the localization better? Does a cut-off radius of 25000 km mean that each proxy influences basically the entire globe?**

A clear reference to section 4.2 has been added in the first paragraph of section 3 , where the influence of covariance localization is characterized in more detail. Also, more details are now provided in section S5 in the supplementary material. See text modifications on Page 9, line 30 and Page 14, line 16.

**Updated reanalysis, Page 9, line 7: Mention that the reference for the skill score is climatology.**

We believe that the CE skill score has well-known characteristics among the intended readership of this paper. We also provide the reference which describes the metric in detail Therefore we decline to add further details, in order to avoid lengthening the text.

**Figures: some text is too small that it cannot be read in a print version.**

We have simplified the design of figures in the revised manuscript. Summary information included as text within figures has been minimized for greater clarity. The font size has also been increased when necessary to increase clarity.

**Figures: figures should have consistent font types.**

We have revised all figures so that a more uniform visual is achieved.

**Anonymous Reviewer 2**

**This reviewer believes the methodology, analysis and the final LMR product presented herein are too premature to be acceptable for formal publication, let alone for its stated purpose to serve as the basis for the first publicly released NOAA last millennium reanalysis.**

The data from the first release described in Hakim et al (2016) has been publically available for over two years. The basic method on which Hakim et al (2016) and the present paper are based has been evaluated and tested extensively in the literature (e.g. Bhend et al, 2012; Steiger et al , 2014; Matsikaris et al, 2015; Acevedo et al, 2017; Franke et al, 2017; Okazaki and Yoshimura, 2017; Steiger et al , 2018).

**It is misleading for this study (and its prototype in H16) to call the DA method used ... as an ensemble Kalman filter (EnKF). As in Evensen (1994) and subsequent studies, the primary promise of the EnKF is the use of flow dependent background error covariance represented by the forecasting ensemble. The current socalled "offline" DA method has none of that: the ensemble perturbations are randomly sampled from a past-millennium climate simulation that has no relation to the prior estimate, and the same set of sampled perturbations were used at all analysis times.**

The reviewer appears to have a view of data assimilation limited to operational weather forecasting. In fact, from a broader perspective, the prior may come from a wide variety of sources, and Monte Carlo sampling of that distribution using ensembles has proved to be a powerful solution method. The "offline" EnKF approach was originally described by Oke et al (2002) and in Evensen's review of the ensemble Kalman filter (Evensen, 2003), as well as subsequently used in ocean data assimilation applications Oke et al (e.g. 2005, 2007), and in various published paleoclimate data assimilation applications (e.g. Steiger et al , 2014; Matsikaris et al, 2015; Steiger et al , 2018). To make that point clearer to readers perhaps less familiar with this literature, we have added a brief discussion in the last paragraph of section 2.1 outlining the aforementioned literature on the origin and applicability of the technique. See Page 5, lines 13 to 20.

**This method used in this study is similar to the commonly used 3D-Var method for numerical weather prediction with static background error covariance, and is arguably less advanced than 3D-Var since 3D-Var in NWP used the dynamic model to propagate the previous cycles analysis as the prior before the analysis. The current so-called "offline" DA method neither cycles the analysis nor the ensemble perturbations, with the stated reason that the forecast model is not good enough to do either.**

We regret the reviewer's interpretation that the motivation for offline DA is because "the forecast model is not good enough," which is not the case. Peraps the original text was not clear enough. The choice is a result of a cost–benefit analysis: predictive skill of Earth System models on proxy timescales is small, but the cost of ensemble forecasts with these models is high. We now make that point clearer in the revised manuscript, in the last paragraph of section 2.1, Page 5 lines 16 to 20.

**If the forecast model is not good enough to cycle the mean analysis or the analysis uncertainties to provide the best estimate of the prior estimate and related prior uncertainties, why would this model(s) be good at all for use as the prior estimate that the LMR reanalysis depends critically on? In this regards, it is premature to state (line 10) that the "LMR employs the ensemble data assimilation to optimally blend the information from the proxies and the climate model data". The current method is more like an objective analysis method.**

Yes, the method is a form of "OI", although we believe that using such jargon is not helpful to the readership of this journal.

**It is not clear whether the authors are aware that the traditional static 3D-Var methods also derive the background covariance from an ensemble of perturbations, as is traditionally called "the NMC method" using the sampled forecast divergence between different lead times from many realizations. The Kalman filter update in this case is equivalent to the variational update using the 3D-Var algorithm, though again the 3DVar in NWP cycles the analysis and forecast during data assimilation, which is the most basic function in combining the model and data.**

We are indeed aware of the NMC method, which samples forecast differences on the timescale of the DA cycle. In our case that is one year, and the random sampling method we employ assumes that forecast differences on that timescale have converged on the climatological distribution; we lack analyses and forecasts over the Common Era to formally apply the NMC method.

**The validation performed in this study for the prototype and updated LMR "reanalysis" with several existing 20th-century reanalysis is misleading at best. The quality of the LMR reanalysis for the 20th century is the least issue given the availability of the modern much more advanced reanalysis and given the exponentially increased number of proxies or model instrumental observations. The validation currently focuses exclusively on the 20th century says little on the quality and performance of the LMR products, in particular over the early period when the proxy data are scarce. A more appropriate validation can potentially be done in two objective methods: (1) perform the 20th century "reanalysis" through thinning the observation density and maybe also degrading the observation accuracy to those representation of different periods of the past millennium; and/or (2) performing observing system experiments in which a certain number of observations are not assimilated but reserved for independent validation (or all of them in cross validation).**

Our validation efforts focusing on the 20th century were primarily motivated by the desire to validate against the most robust reference datasets, as well as being able to assess the spatial skill of our reconstructions with some confidence. However we agree this is not comprehensive, as information about performance over the pre-industrial period is not provided. Consistent with suggestion (2), part of the DA method we have developed involves withholding 25% of the proxies for independent validation, which is performed randomly for each of the 51 realizations in each experiment, so that all proxies participate in validation. Validation statistics are compiled both before and during the instrumental period. These results are now highlighted more clearly

in numerous different parts in the main body of the revised paper. See Page 12 lines 17 to 23, Page 14, lines 2 to 7, and Page 16 lines 3 to 20.

Regarding suggestion (1), we have performed experiments where we vary the percentage of withheld proxies and we have found little sensitivity to the 25% value.

**The use of a 2,5000-km covariance localization is highly questionable for the use of a 100 sets of fixed ensemble perturbations. At midlatitudes, this is amount to the observation impacts across the entire global latitude belt. The use of a fixed set of 100 sample perturbations also means a high rank deficiency over such a large area with this large localization distance.**

This reasoning is consistent with covariance lengthscales on weather timescales. Covariance lengthscales on annual timescales are much longer, and the effective dimension of the covariance matrix is comparatively smaller.

**The current final LMR reanalysis derives from the mean of 51 such 100-member analyses, should it be the same if the 5100 samples of perturbations are used simultaneously in the Kalman filter update given the Kalman filter used is largely a linear operation?**

We have tested this and have found that better results are obtained by averaging over Monte Carlo realizations (multiple analyses) as compared to a single large ensemble. The underlying reason is not completely understood and believe that fully exploring this issue is beyond the scope of the present paper. However, we believe that this finding is an artifact of averaging over analysis errors related to poorly estimated observation errors for some proxies. In the revised manuscript, section 2.3, we highlight this issue as an additional motivation for performing multiple Monte-Carlo realizations and outline the hypothesis we have for the behavior. See revised text Page 6, lines 19 to 21.

**How much is the result sensitive to the choice of this arbitrary number of sample perturbations? It is also worth noting the the NMC method used for 3D-Var uses singular value decomposition to make it full rank. Such a approach is different from (and likely more advantageous over) the current Kalman filter update using purely non-envolving static ensemble covariances.**

We have found little sensitivity to the ensemble size, provided it is at least 100 members and that covariance localization is used (effectively increases the rank of the covariance matrix). We find larger improvements by randomly subsampling the proxies through many Monte Carlo realizations.

**It is unclear what is the purpose of such as hastily done LMR reanalysis products with such ad-hoc DA approaches and the not-good-enough forecast models? The so derived climate trend is almost certainly depending too much on the climate models used as a prior and ensemble sampled perturbations (and maybe the assumed climate forcings used in these models), as well as the density of observations over different periods. It could do more harm if such a premature reanalysis product is used or misused and if it**

**were publicly released through NOAA, unfortunately. A more careful vetting of the products, and a more concerned effort in refined DA methodology are warranted before NOAA sanctioned such a product as reanalysis, in this reviewers opinion.**

If the reviewer wishes to see more sensitivity analysis with respect to these issues, please carefully read Hakim et al (2016), where we not only considered the performance statistics of analyses using different priors and calibrations of proxy forward models, but also examples of the differences that result in the spatial fields for an individual year.

**Anonymous Reviewer 3**

**I have two main concerns about the paper, which together require that the manuscript undergo major revisions before it is acceptable for publication. The first is the character of the derived reconstruction and the unsatisfactory verification of the product using only observational data. The second is the use of multiple ad hoc methodological choices, none of which are reasonably justified or widely tested.**

The revised manuscript includes a more complete discussion on the character of the temperature reconstruction, compared and contrasted with other reconstructions (see response to next comment).

The scope of the verification of reconstruction results presented in the paper has been expanded by including results from verification performed in proxy space on independent (withheld) proxies. Results that were originally included in the supplementary material have been revised and moved to the main body of the paper. These results are now found in section 4.1, Table 3, to evaluate in a more comprehensive fashion the various PSM configurations considered and to further confirm choices made for the configuration of the updated reanalysis. Results from proxy–space verification are also included in section 4.3, Table 5, to gain an additional perspective on the accuracy of reconstructions performed with the addition of proxy records from the Anderson et al (2019) collection. See revised text page 14, lines 1 to 7, Page 16 lines 3 to 19.

Regarding the referee's concern on the justification of our selection of data assimilation parameters is addressed with the addition of a brief mention on how those choices were made at the end of section 2.3. We believe however that a lengthy discussion on these is not warranted in the present paper, as these choices have been discussed in prior publications. The following statement has been added in the last paragraph os section 2.3: "Little sensitivity to the use of 75% of the proxies for each realization hase been found (not shown), while 100 members have been chosen to maintain consistency with H16. " , Page 6 lines 19 to 23.

**I am struck by the comparison in 2a and the little attention the authors give to the differences between the previous LMR product and the newer version (not to mention the complete lack of comparison between either of these results and other temperature reconstructions).**

We agree with this comment. As a result, we now compare LMR results of Northern Hemisphere (NH) temperature to other NH reconstructions found in the IPCC AR4 and AR5 reports. This comparison is included in the bottom panel of Figure 2, section 3, to obtain a greater perspective on the long-term (i.e. across the Common Era) evolution of temperature in our reconstructions. A discussion outlining the main takeaways from this comparison can be found in the fourth paragraph of a revised section 3. See revised manuscript, Page 10 line 23 to Page 11 line 7.

**The GMT from the newer product looks almost like white noise and has lost not only the multi-decadal to centennial variability in the first product, it is also likely at odds with the now large collection of global and hemispheric temperature reconstructions spanning the last millennium or more. The authors not only need to spend more time discussing**

**this issue, they also need to compare their results to the collection of large-scale temperature and hydroclimate reconstructions currently available.**

We do acknowledge the updated reanalysis in our originally submitted manuscript was characterized by a significant loss of variability compared to our prototype reanalysis. The comparison with other reconstructions (see previous response) has helped gain additional perspective on the long-term evolution of temperature in our reanalyses. As a result, we have revisited some of the choices made in the configuration of our system and conducted additional experiments to identify the source of this loss of variability. We have determined that it is preferable to eliminate from our reanalysis the large number of unscreened tree-ring records from the Breitenmoser et al (2014) collection, as included in Anderson et al (2019). This is despite increased skill in temperature and hydroclimate reconstructions when this dataset is included, as determined from observational data. We believe this underlines some issues with our estimates of observation error variance (i.e. the $R_k$ terms in equations 4 in the manuscript) for these records. We have found that the inclusion of these tree-ring chronologies lead to the warm bias characterizing the reconstructed temperatures in our previous updated reanalysis during the 1600–1700CE and 1810–1920CE periods of the Little Ice Age (LIA).

In the revised manuscript, we present a new updated reanalysis using a revised configuration of the assimilated proxies, limited to the proxies from the PAGES 2k Consortium (2017) collection. With this configuration, we find that greater level of temperature variability is recovered, as well as a much improved agreement between the LMR Northern Hemisphere temperature and other reconstructions during the LIA. We note however that the absence of a notable warm medieval period continues to characterize our reanalysis. An analysis of results from a large number of reconstruction experiments has allowed us to conclude that colder temperatures during the medieval period, compared to the prototype LMR, are related to the change from the proxy dataset of PAGES 2k Consortium (2013) to the more recent PAGES 2k Consortium (2017) collection. The global temperature composites presented in PAGES 2k Consortium (2017) shows that a distinctly warmer medieval period isn't a prominent feature of the new collection, and is not the result of other updates to our data assimilation system. As this dataset reflects the community's most recent and rigorous identification of proxy records suitable for temperature reconstructions, we believe that the lack of a "classic" medieval warm period in our updated reconstructions of global mean surface temperature and NH-mean temperature should not necessarily be considered an outstanding shortcoming of our updated reanalysis. Discussions in the issues outlined above are included in sections 3 and 4.3. See revised mansucript Page 10 lines 3 to 12, and Page 10 line 23 to Page 11 line 7.

**...it is essential for them to do more to verify their results beyond the comparisons they make to observational data. While the latter is important and useful, it is not enough. Incidentally, the authors do perform validation exercises on a withheld period of observational data and using withheld proxy series, but that work is buried in the supplemental and not adequately discussed in this context. More should be made of those efforts, which strengthen the authors results with truly out-of-sample validation experiments.**

We agree. As described in our response to a previous similar comment, additional verification results are presented in the revised manuscript. More specifically, verification performed

in proxy space on independent proxies are used to further evaluate the impact of the various PSM configurations considered and to further confirm choices made for the configuration of the updated reanalysis (section 4.1, Table 4), and to gain additional perspective on the accuracy of test reconstructions using added records from Anderson et al (2019) (section 4.3, Table 6). See revised mansucript, first paragraph on Page 14, and Page 16, lines 3 to 19.

**I do not think the use of CE is the same as it is traditionally used in the paleo literature, given that the latter approach requires a true cross-validation period. The authors should clarify this point.**

The formulation has been used in numerous other published work, including in a similar manner as we do in this work. We have found that CE is a valuable complementary metric to correlation, due to its sensitivity to mean errors and representation of variance in the evaluated fields. Furthermore, we now use the CE score using independent proxy data as the verification data.

**I think it is further important for the authors to derive validation experiments for the sparsely sampled periods early in the proxy network (e.g. deriving reconstructions using only subsets of the proxies that extend back to specific time intervals). This would go a long way toward helping to better understand the loss of proxy information back in time. This is partially addressed by the variance exercise the authors perform, but more can be done. Recons for temporal subsets of the proxy network would in fact be more useful than the MC sampling of the proxy network that the authors perform, given that it would be systematic and inform a direct question about the influence of the declining proxy network.**

This is a very good suggestion. Experiments similar to what is suggested here have been conducted in support of another publication, currently under review. This issue can also be investigated through an assessment of reanalysis performance performed in proxy space, with a focus on different time intervals. We have generated proxy verification results over distinct periods of the Common Era: 0–499, 500–999, 1000–1499, 1500–1879, 1880-2000 (calibration period) in Table 2 and discuss the results from the perspective of reanalysis accuracy across time in a revised section 3. Even though the verification is not performed on an identical set of proxies we also compare the verification statistics from the prototype reanalysis, to further highlight the enhanced performance of the updated LMR. See revised manuscript Page 11, line 34 to Page 12 line 22.

**Here is a ... list of choices the authors have adopted that are not accompanied by any justifications or sensitivity discussion: 1-Use of 100-member ensemble; 2-Use of the CCSM4 last millennium simulation as the prior; 3-Use of 51 MC realizations; 4-Use of a proxy sampling scheme based on 75% of the proxy records; 5-Degradation of the model resolution to a ~5x5 grid. All of these choices undoubtedly influence the derived LMR product. Some of them can be justified based on discussions in the literature. Some of them require empirical demonstrations. All of them come across as ad hoc. I would also venture to guess that the LMR results are more dependent on a couple of these choices**

**than the other dependencies that the authors more systematically test. It is therefore essential that the authors do a better job of justifying these choices and convincing the reader that they are either reasonable choices or chosen based on some methodological/logistical rationale.**

Some of these choices have been discussed in prior publications. See Hakim et al (2016) and references therein for example. However we agree that text reminding the reader of how parameters were chosen should be included to convince that careful consideration was involved. To further clarify the context, we should point out that some parameter values used here were chosen to maintain consistency with the configuration used in Hakim et al (2016), in order emphasize the impact of updates specifically addressed in this work.

Some of the parameters were originally chosen based on practical considerations. For example, the ratio of assimilated versus withheld proxies and the number of Monte-Carlo realizations were chosen so that the random sample of proxies set aside for independent validation is representative of the overall dataset, while maintaining minimal sensitivity on the accuracy of the the reanalysis. A revised statement clarifying these points has been added in the last paragraph of section 2.3., page 6, lines 17 to 23.

**I should specifically mention the use of the CCSM4 as the prior. The authors say nothing about how their results might depend on the model prior and whether they have tested alternative last-millennium simulations in their analysis. This is an obvious question and the authors need to address it.**

This topic is addressed in Hakim et al (2016), with results from reconstructions using a wide range of calibration and prior data (see Fig. 12 of that paper). Work on this topic has since been expanded and explored in more detail. These results are, however, expected to be the subject of a separate publication.

**Page 2, lines 5-6: What does "synthesizing information" mean? This is vague and I am not even sure the statement is true. There are lots of central challenges of paleoclimate science, and it is arguable that what the authors are alluding to is one of them. This strikes me as an unsupported justification for what the authors subsequently say they are attempting to do.**

In this part of the text, our goal is to emphasize that data assimilation is a powerful framework in which information from proxies and numerical model simulations is combined for the production of, hopefully, robust climate reconstructions. Therefore, we believe that this goal is appropriately described by the verb "synthesize", which may be defined as the act of combining (a number of things) into a coherent whole. We modified the statement as being "one of the challenges of paleoclmate science" in order to more accurately convey our belief that other challenges than this one exist. See revised manuscript Page 2 line 1.

**Page 2, lines 30-32: This is a much more mundane objective than the sense given in the abstract. Are the authors attempting to release a shiny new LMR product or should this be seen as an iterative verification step toward some improved effort down the line?**

We apologize if the text does not clearly convey the goals pursued here. As with any complex problem to be solved, as is the case here, improvements are generally incremental and modest. The main goal of efforts reported in this paper has been to improve our system over the LMR prototype, and to deliver an incrementally improved product, while being fully cognizant of the fact that room for improvement remains. A revised statement at the end of the third paragraph now includes "compared to the prototype LMR" to better convey the scope of our objective, while the remaining room for improvement is acknowledged in a clearer manner in the next to last paragraph of the conclusion. See Page 2 lines 29-30, and Page 17, lines 12 and 13.

**Page 8, line 9: The use of precipitation is not justified and concerning. First, precipitation is almost never the variable associated with moisture sensitivity in trees - some measure of soil moisture is. It is therefore not clear why the authors used precipitation and how it influences their results. Why not use a more conventional variable like PDSI? Secondly, how do the characteristics of precipitation influence the results? Does it matter that precip is likely not Gaussian and that it has limited spatial and temporal covariance structure? Is the use of precip perhaps adding to the loss of low-frequency variance in this new LMR product? My guess is that this specific choice has a large impact on the derived reconstruction and the use of precip is not justified in any way.**

The reviewer is raising fair and important points here. We acknowledge that soil moisture is the preferred response variable for the modeling of tree-ring widths. However, given our approach, which relies on the availability of calibrated forward models, the absence, to our knowledge, of a reliable century-long soil moisture dataset is an important limiting factor. An alternative, as pointed out by the referee, would be PDSI (or other drought indices such as SPEI) instead of soil moisture. This option, is enabled within our framework with the use of the Dai et al (2004) dataset for forward model calibration (see below). To provide additional context about our efforts, more particularly with the development of bivariate "temperature and moisture" PSMs, our intent is to replicate VSlite's (Tolwinski-Ward et al, 2011) general approach of combining the influences of temperature and moisture in modeling tree-ring width variability, albeit in a simpler fashion, while avoiding the issues associated with VSlite's formulation involving thresholds (see Dee et al, 2016). We use precipitation as the moisture source variable in bivariate models rather than PDSI, since the latter is itself a function of temperature through the potential evapotranspiration term involved in its calculation.

In response to the reviewer's concern, we compare two reconstruction experiments in which the univariate "temperature or moisture" ("TorM") models are used to forward model tree-ring width proxies. One experiment is performed with PSMs calibrated on PDSI using the dataset from Dai et al (2004), while the other uses models calibrated on precipitation with the GPCC dataset (Schneider et al, 2014). In each case, as described in section 2.4.2 in the main text, decisions are made whether each tree-ring record is moisture sensitive or temperature sensitive by comparing moisture calibrations (with PDSI or precipitation) and temperature calibrations (using GISTEMP as the calibration data). The regression providing the better fit is used to forward model the proxy record. In all experiments. all other proxies are modeled with univariate temperature PSMs. Other reconstruction parameters are the same as with experiments described in the paper (100 ensemble members, CCSM4 as the prior model, 51 Monte-Carlo realizations with 75% of proxies assimilated). Covariance localization is not applied here. Temperature reconstructions from both experiments are verified against instrumental-era analyses and against independent (withheld from assimilation) proxies and results are compared in Tables AR.1 and AR.2. Results obtained from a reconstruction using bivariate PSMs calibrated on temperature and precipitation are included for comparison. The skill of reconstructions using the univariate models are similar, whether PSMs are calibrated on PDSI or precipitation. But more importantly, the skill scores are generally inferior to those obtained with the use of bivariate models. This is particularly true with the CE score. Proxy-based verification results are less definitive due to the noisy nature of the proxies, but generally support to the conclusion drawn from instrumental-era verification. We have also calculated the spectra of Northern Hemisphere mean temperature from both reconstructions, shown in Fig. AR.1. The results are nearly identical (indistinguishable at the 95% confidence level) , suggesting that the use of precipitation-calibrated PSMs does not lead to a loss of variability in temperature reconstructions. Furthermore, the exercise has shown that bivariate temperature/precipitation PSMs provide superior results compared to univatiate "TorM" models, even when the latter are calibrated on PDSI. As part of the revision of the manuscript, the results discussed herein are now included in the supplementary material, while a statement referring to section S4 of the supplementary material is included in the last sentence of section 2.4. See revised manuscript Page 7, lines 33-34.

**Figures: In general, there is a lot of small text in the figures that is hard to read and also rather confusing and messy.**

We have simplified the design of figures in the revised manuscript. The summary information included as text on the figures has been minimized for greater clarity.

**Figures: The many colorbars are also unnecessary in many plots when one would do.**

We thank you for your recommendation. We acknowledge the presence of multiple colorbars, however we point out that the ranges shown in the various frames are not all identical. Therefore, we elect to keep showing the colorbars. However we made efforts in cleaning up and streamlining the design of the figures for greater clarity.

Table AR.1: Summary of instrumental–era verification results for reconstructions performed with various PSM configurations to model tree-ring width proxies (see main text, section 2.4.2 for PSM details). Verification scores shown are correlation (r) and coefficient of efficiency (CE) for the annual global mean temperature (GMT) and detrended GMT verified against the consensus of instrumental–era analyses, the global mean of gridpoint r and CE characterizing the spatially reconstructed temperature, verified against the Berkeley Earth analysis.

| PSM configuration | Annual GMT | | Detrended GMT | | Spatial temperature | |
|---|---|---|---|---|---|---|
| | r | CE | r | CE | r | CE |
| Univariate "TorM" (PDSI) | 0.92 | 0.79 | 0.72 | 0.46 | 0.52 | 0.17 |
| Univariate "TorM" (precip.) | 0.93 | 0.77 | 0.74 | 0.48 | 0.53 | 0.17 |
| Bivariate | 0.93 | 0.86 | 0.77 | 0.54 | 0.53 | 0.20 |

[revised manuscript text omitted]
 bottom of each panel, for the verification period shown in (c) original and (d)detrended time series.

**Instrumental-era verification for temperature**
**LMR vs. Berkeley Earth**

[Figure]

**Figure 4.** Verification of LMR 2m air temperature against the Berkeley Earth instrumental–era analysis over the 1880–2000 period. Shown are time series correlation (left column) and coefficient of efficiency (CE, right column), for (a) and (b) the prototype and, (c) and (d) the updated reanalysis. Differences in correlations and CE between the two experiments are shown in (e) and (f) respectively. Gray shading indicate regions with insufficient valid data for meaningful verification statistics.

[Figure]

**Instrumental-era verification for Z500**
**LMR vs. 20CRv2**

**(a)**
Correlation, Prototype, mean=0.41

**(b)**
CE, Prototype, mean=0.07

**(c)**
Correlation, Updated LMR, mean=0.45

**(d)**
CE, Updated LMR, mean=0.18

**(e)**
Correlation difference, mean=0.04

**(f)**
CE difference, mean=0.11

**Figure 5.** As in Fig. 4 except for the verification of LMR 500 hPa geopotential height anomalies against the 20CR-v2 reanalysis.

**Instrumental-era verification for temperature**
**LMR vs. Berkeley Earth**

**(a)**
r, Exp1:Annual, mean=0.55

**(b)**
CE, Exp1:Annual, mean=0.14

**(c)**
r, Exp2:Seasonal (metadata), mean=0.54

**(d)**
CE, Exp2:Seasonal (metadata), mean=0.19

**(e)**
r, Exp3:Seasonal (objective), mean=0.55

**(f)**
CE, Exp3:Seasonal (objective), mean=0.22

**(g)**
(Exp2-Exp1) r difference, mean=-0.01

**(h)**
(Exp2-Exp1) CE difference, mean=0.05

**(i)**
(Exp3-Exp1) r difference, mean=0.0

**(j)**
(Exp3-Exp1) CE difference, mean=0.08

[revised manuscript text omitted]

---

## Author Response (AR2)

Last Millennium Reanalysis with an expanded proxy database and seasonal proxy modeling

Tardif et al.

Authors Responses to Reviewers - Second revision

In the following, comments from the referees are shown in bold, with our responses shown in plain text. A marked-up version of the revised manuscript, showing the changes from the previously revised version, is also included. Page and line numbers refer to the most recently revised manuscript.

**Anonymous Reviewer 1**

**The manuscript has improved significantly due to the major revisions that have been conducted by the authors. Reconstruction skill is now evaluated before the 20th century. Additionally, multi-decal variability is discussed with regard to the selected input data set and a comparison with previous statistical reconstructions has been added. Therefore, I suggest publication after a few minor improvements and clarifications.**

We thank the referee for comments that have challenged us to present a more comprehensive set of results in the paper. We believe that the manuscript has improved as a result.

**What I would like to see most is, how spatial reconstruction skill changes over time, i.e. which regions are corrected by data assimilation with a sparse paleodata. This could be shown for instance with maps that indicate the reductions of ensemble spread at a few time slices, e.g. 250, 750, 1250 and 1750 CE. Something similar is presented in Fig. 2b but just for global mean temperature.**

We share the referee's interest in this aspect of paleoclimate reanalyses. In the prior revision of the manuscript, we have introduced results on the dependence of skill over time by showing proxy space verification statistics compiled over various time intervals within the Common Era. We agree however that these efforts do not provide a complete perspective on the issue. Nonetheless, we believe that a comprehensive discussion on the topic remains outside the scope of the current presentation. To maintain a reasonable length of the current manuscript and because we plan to characterize the impact of a temporally variable proxy network in a future publication, we leave the proposed analysis for future work.

**Looking at the time series in Fig. 2c it appears like the reconstruction is very close to climatology in the first millennium due to static offline assimilation approach and few paleodata, which very available at this time. This tendency towards climatology, inherent to this data assimilation approach, should be discussed.**

The referee is correct with his/her interpretation. In our data assimilation methodology, the posterior (the analysis) reverts to the prior in the absence of new information from proxies. This can be easily inferred from Eqn. 1 in the manuscript. Since the prior is defined here using the

same random draws of model states at every analysis times, the absence of proxy information leads to a posterior representing climatology expressed through the randomly selected states. This possible outcome is now described in the last paragraph of section 2.1, where the offline approach is discussed in more detail. See Page 5, lines 15–17.

**The main conclusion from this paper is the improvement compared to Hakim et al, 2016 mostly based on validation in the 20th century. In Fig. 2b it appears like the major improvement compared to the prototype version would be just in the 20th century, which is mostly used for evaluation in this study of not of scientific interest. This needs to be discussed, too.**

We share the view of the referee that the 20th century is not the primary interest within the context of paleoclimate reconstructions. However we respectfully disagree with the inference from our results that improvements over the prototype reconstruction are confined to the 20th century. Improvements are obtained over most of the reconstruction period. This is shown, for example, by proxy verification results (Table 2), discussed in the last paragraph of section 3. Moreover, Fig. 2b provides information on the uncertainty (ensemble spread) characterizing the reanalyses. From that perspective, we see a noticeable reduction of the uncertainty in the updated reanalysis over the prototype over the first 500 years and the over last 300 years of the reconstruction. This is discussed in the third paragraph of section 3. We acknowledge a significant emphasis on instrumental-era verification in our manuscript. Although narrower in scope, we believe that a comparison of our reanalyses against the most accurate gridded products available provide additional insights into the spatially-distributed skill of our reconstructions. Following the recommendations of the prior review, we have complemented these results with a more complete evaluation using proxy verification which cover periods prior to the instrumental era. We hope to convince the referee that our conclusions are more nuanced and provide a wider perspective than what is suggested here.

**Abstract, Line 5: Are citations allowed in the abstract?**

Thank you for raising this issue. Indeed the publication guidelines specify that reference citations should not be included in this section, unless urgently required. The citation in the abstract has been removed without loss of clarity.

**Abstract, I suggest adding one sentences about the important result that the choice of the input data set determines the amplitude of multi-decadal to centennial variability in the reconstruction.**

The following sentence has been added in the abstract: "The variability of temperature at multi-decadal to centennial scales is also shown to be significantly sensitive to the set of assimilated proxies, especially to the inclusion of primarily moisture-sensitive tree-ring width records." See Page 1, lines 13–15.

**Section 2.2. Climate proxies: Are possible duplicate records in the PAGES2k and Breitenmoser et al. 2014 collection somehow identified/treated?**

This is a great point. Indeed possible duplication of records has been addressed by comparing site locations and by a correlation analysis. For highly correlated co-located records, the available metadata has been examined in greater detail to confirm duplication and flag the records. In the event of duplication, prevalence is given to the record included in the PAGES2k dataset. This process is now briefly mentioned in section 4.3, instead of section 2.2 as suggested by the referee, and a reference to Anderson et al (2019) is provided for completeness. We believe the discussion in section 4.3 is a more appropriate context for this information. See Page 15, lines 13–15.

**Section 2.2. Climate proxies: Tree-ring density records discussed here do not appear in Fig. 1 but only in a separate Fig. 8. I suggest merging Fig. 8 into a third panel of Fig. 1 and having an overview of all three tested proxy networks this way.**

All tree-ring density records considered are included in the PAGES2k-2017 dataset and do appear in Fig. 1. Perhaps the referee refers to the tree-ring width records from Breitenmoser et al (2014) briefly mentioned in this section. However we chose to focus the discussion on the addition of these proxy records in section 4.3, and maintain a focus on proxies from the PAGES2k-2017 dataset in the part of the manuscript leading to the presentation of the updated reanalysis in section 3. We believe the flow of the material is clearer this way. Therefore, we thank you for your comment, but decline to make the proposed change to the manuscript.

**Section 2.3. Climate model prior information: Does the CCSM ensemble really only cover the period 850 to 1850 CE? If so, it would be worth discussing why an ensemble without 20th century greenhouse gas forcing performs so well in reconstructing 20th century climate.**

We thank you for this comment, as this is an interesting point. We confirm that the CCSM simulation used as the source of prior data does not include the 20th century greenhouse gas forcing (please see the reference provided in the text). This fact, along with the use of offline data assimilation, implies that the trends and temporal structure characterizing the reconstructions are entirely generated from the assimilated proxies, and good performance in reconstructing 20th century climate also implies that the proxy system models used in LMR are appropriate. This fact was discussed in Hakim et al (2016), but we also now point this out more clearly in section 2.3 by adding the sentence "Consequently, all trends and temporal structure in reconstructed fields result from information provided by the proxies." See Page 6, lines 20–21.

**Section 2.4.1 Seasonality: Line 21: in neither of the cited papers I find indications that the growing season could last until November as in the used season (JJASON).**

We agree that inclusion of November is a stretch for the growing season, but included it because it was indicated for some chronologies in the expert metadata in the PAGES2k dataset. As indicated in the supplementary material (Fig. S1), our objective results show fewer occurrences of autumn months in definitions of tree-ring seasonal responses compared to the expert metadata included in the PAGES2k dataset. We conclude that such autumn seasonal responses should have less influence on reconstructions performed with our objective seasonal responses

compared to those defined in the PAGES2k metadata.

**Section 2.4.2: Line 31: It would be helpful for the reader to know a little more details about how the withheld proxy data with low signal-to-noise ratios will be used to measure reconstruction error.**

Thank you for your comment. This information is presented in section 3. We have added a reference pointing the reader to this material in section 2.4.2, page 9, lines 10–11.

**Section 3. The updated analysis: Page 11, line 1: "isn't" should be "is not"**

Thank you. The correction has been made.

**Section 3. The updated analysis: Page 11, line 14: Is the abbreviation BE really needed? It is used very rarely.**

Thank you for pointing this out. The abbreviation has been eliminated.

**Section 3. The updated analysis: Page 11, line 17: define or explain CE briefly. Either with an equation or as a RMSE based skill score with climatology as a reference.**

The CE verification score is introduced earlier in section 3, with a reference to Nash and Sutcliffe (1970) already provided. However, to better inform the reader, we have added the following sentence "These skill scores are complementary since correlation measures signal timing while CE, based on mean square error with climatology as a reference, is sensitive to bias and errors in signal amplitude." at the end of the first paragraph of section 3. See Page 10, lines 10–12.

**Section 4.1 Proxy system models: Page 13, line 2: the so-called expert metadata is also mainly derived by identification of highest correlation coefficients and hence not really different from the so-called objective method. The difference is rather that more variations of growing season months are possible and only correlation with temperature was used.**

We believe that the expert metadata has been derived using various methods, among which is a highest correlation approach, similar to our "objective" method. However we cannot ascertain the proportion of records for which metadata seasonality has been derived using this method. Therefore, we cannot unequivocally identify the source of the differences observed in the reconstructions using metadata versus objective seasonalities. However, the main motivation for developing our own correlation-based method is rooted in properly forward modeling the Breitenmoser tree-ring records, for which metadata seasonality has been derived using a simple approach (see Anderson et al (2019)), and also to include moisture in the forward modeling of tree-rig widths, a variable for which metadata seasonality is not readily available in our datasets.

**Section 4.2 Covariance localization: Page 14, line 24: add a reference to the Fig. S4**

We agree. The sentence "See Figure S4 in the supplementary material for an example where the 25000 km localization function is applied to a proxy record located in California, United States" has been added to Section 4.2, page 15, lines 7–9.

**Anonymous Reviewer 3**

**The authors have addressed my comments comprehensively and the manuscript is ready for publication after attention to a few minor details.**

We thank the referee for comments which have helped improve the manuscript.

**Pg. 2, Ln. 31: There is little purpose for this kind of section summary. It should just be removed.**

Thank you for this suggestion. However, a quick survey of recently published papers reveals that the inclusion of such a paragraph remains a common practice. We conclude that some readers find this information useful. We therefore elect to keep the paragraph outlining the content of the paper.

**Pg. 7, Ln 4-7: I would specify that they are really talking about linear statistical PSMs here. As such, they should add that relationships may not be linear, and if univariate models are used, they may be subject to errors associated with multivariate influences.**

We agree with the referee. The issues raised are now listed in the first paragraph of Section 2.4, on page 7, line 17.

**Pg. 11, Ln 2-4: The focus on the Spörer minimum period seems a little narrow here. Yes the updated recon fits better, but there are lots of examples where other 100-yr periods go the other way. I would just focus on the more collective behavior and leave this odd specificity out.**

The paragraph in question highlights several periods where differences between the LMR prototype and updated temperature reconstructions are most noticeable, including, but not limited to, the period around the Spörer minimum. Differences observed during the Medieval Warm Period, as well as during the late nineteenth and early twentieth centuries are also discussed, including comparisons with reconstructions from other authors to bring a greater perspective on the results. Therefore we believe the discussion is broader in scope than suggested. However, to eliminate possible confusion related to the definition of the Spörer minimum, we have elected to designate the highlighted period with the more general "late fifteenth and early sixteenth centuries" in the text. See page 11, lines 10 and 17–18.

**Pg. 14, Ln 22: scores not cores.**

The typo has been fixed. Thank you for pointing it out.

**Pg. 15, Ln. 21: The authors discuss Dai (2004) not Dai (2011) in their response to my comments. Please confirm that Dai (2011) was used, which is the appropriate PM formulation of PDSI, which was updated from 2004 that used the Thornthwaite version.**

Thank you for identifying this discrepancy. The reference to Dai et al (2004) instead of Dai (2011) in our response to the referee was an oversight. We use data generated at the National Center for Atmospheric Sciences, obtained via the NOAA Earth System Research Laboratory Physical Sciences Division data portal. The metadata included in the netCDF data file confirms that PDSI calculations were performed using the Penman-Monteith PE formulation.

**Pg. 15, Ln 22: The independence of precipitation and PDSI is more nuanced than the authors suggest, given that PDSI of course includes precipitation in its formulation.**

We agree with the referee that the statement in the manuscript can be perceived as an overstatement. We have replaced "We note here that this verification is entirely independent..." with "We note here that the reconstruction is not directly elated to the PDSI product used for verification...". See page 16, line 4.

**Pg. 15, Ln 24-25: The authors should clarify that PDSI was not included in the original or prototype LMR product. They appear to be testing it here as a new variable included in the prototype calculation, but my understanding was that it was not originally reconstructed.**

The referee is correct that PDSI was not reconstructed in Hakim et al (2016). But as already indicated in footnote 2 on Page 9, PDSI has been generated using the reanalysis configuration used by Hakim et al (2016) as part of efforts reported in this paper, for the purpose of comparing with results obtained with the updated LMR configuration. An additional footnote has been included as a reminder to the reader. See Page 16, line 6 and bottom of page.

**Figure 11: It is hard to compare the two time series in this format. Why not include them on the same axis, as done for the two time series in Figure 2?**

Figure 11 has been revised following the referee's suggestion.

[revised manuscript text omitted]

---

## Author Response (AR3)

Last Millennium Reanalysis with an expanded proxy database and seasonal proxy modeling

Tardif et al.

Authors Responses to Reviewers - Final revision

In the following, comments are shown in bold, with our response shown in plain text.

**Editor**

**Before I can accept the manuscript, I would like you to include a Statement on the availability of underlying data as a section of its own.**

Following your suggestion, the statement on code and data availability originally included in the section *Code and data availability* has been divided in two separate statements in the *Code availability* and *Data availability* sections.